# Expressive Higher-Order Link Prediction through Hypergraph Symmetry Breaking

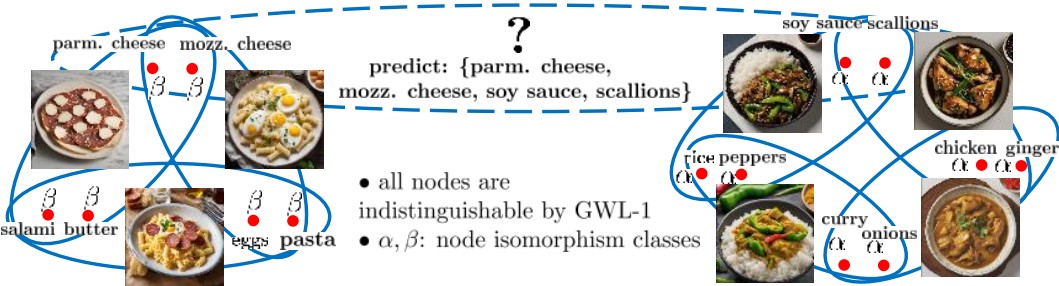

Figure 1: An illustration of a hypergraph of recipes. The nodes are the ingredients and the hyperedges are the recipes. The task of higher order link prediction is to predict hyperedges in the hypergraph. A negative hyperedge sample would be the dotted hyperedge. The Asian ingredient nodes ($\alpha$) and the European ingredient nodes ($\beta$) form two separate isomorphism classes. However, GWL-1 cannot distinguish between these classes and will predict a false positive for the negative sample.

## Abstract

A hypergraph consists of a set of nodes along with a collection of subsets of the nodes called hyperedges. Higher order link prediction is the task of predicting the existence of a missing hyperedge in a hypergraph. A hyperedge representation learned for higher order link prediction is fully expressive when it does not lose distinguishing power up to an isomorphism. Many existing hypergraph representation learners, are bounded in expressive power by the Generalized Weisfeiler Lehman-1 (GWL-1) algorithm, a generalization of the Weisfeiler Lehman-1 algorithm. However, GWL-1 has limited expressive power. In fact, induced subhypergraphs with identical GWL-1 valued nodes are indistinguishable. Furthermore, message passing on hypergraphs can already be computationally expensive, especially on GPU memory. To address these limitations, we devise a preprocessing algorithm that can identify certain regular subhypergraphs exhibiting symmetry. Our preprocessing algorithm runs once with complexity the size of the input hypergraph. During training, we randomly replace subhypergraphs identfied by the algorithm with covering hyperedges to break symmetry. We show that our method improves the expressivity of GWL-1. Our extensive experiments [1] also demonstrate the effectiveness of our approach for higher-order link prediction on both graph and hypergraph datasets with negligible change in computation.

# 1 Introduction

In real world networks, it is common for a relation amongst nodes to be defined beyond a pair of nodes. Hypergraphs are the most general examples of this. These have applications in recommender systems Lü et al.

---

[1] https://anonymous.4open.science/r/HypergraphSymmetryBreaking-B07F/

(2012), visual classification Feng et al. (2019), and social networks Li et al. (2013). Given an unattributed hypergraph, our goal is to perform higher order link prediction (finding missing hyperedges) with deep learning methods while also respecting symmetries of the hypergraph.

Current approaches towards hypergraph representation are based on the Generalized Weisfeiler Lehman 1 (GWL-1) algorithm Huang & Yang (2021), a hypergraph isomorphism testing approximation algorithm that generalizes the message passing algorithm called Weisfeiler Lehman 1 (WL-1) Weisfeiler & Leman (1968)to hypergraphs. Due to message passing locally within a node's neighborhood, Weisfeiler Lehman 1 views a graph as a collection of trees rooted at each node. However this can violate the true meaning of the graph, in particular its symmetry properties. GWL-1 also views the hypergraph as a collection of rooted trees. These hypergraph representation methods, called hyperGNNs, are parameterized versions of GWL-1.

To improve the expressivity of hypergraph/graph isomorphism approximators like GWL-1 or WL-1, it is common to augment the nodes with extra information You et al. (2021); Sato et al. (2021). We devise a method that, instead, selectively breaks the symmetry of the hypergraph topology itself coming from the limitations of the hyperGNN architecture. Since message passing on hypergraphs can be very computationally expensive, our method is designed as a preprocessing algorithm that can improve the expressive power of GWL-1 for higher order link prediction. Since the preprocessing only runs once with complexity linear in the input, we do not add to any computational complexity from training.

Similar to a substructure counting algorithm Bouritsas et al. (2022), we identify certain symmetries in induced subhypergraphs. However, unlike in existing work where node attributes are modified, we directly target and modify the symmetries in the topology. During training, we randomly replace the hyperedges of the identified symmetric regular induced subhypergraphs with single hyperedges that cover the nodes of each subhypergraph. We show that our method can increase the expressivity of existing hypergraph neural networks.

We summarize our contributions as follows:

- We characterize the expressive power and limitations of GWL-1.

- We devise an efficient hypergraph preprocessing algorithm to improve the expressivity of GWL-1 for higher order link prediction

- We perform extensive experiments on real world datasets to demonstrate the effectiveness of our approach.

## 2 Background

We go over what a hypergraph is and how these structures are represented as tensors. We then define what a hypergraph isomorphism is.

### 2.1 Isomorphisms on Higher Order Structures

A hypergraph is a generalization of a graph. Hypergraphs allow for all possible subsets over a set of vertices, called hyperedges. We can thus formally define a hypergraph as:

**Definition 2.1.** *An undirected hypergraph is a pair $\mathcal{H} = (\mathcal{V}, \mathcal{E})$ consisting of a set of vertices $\mathcal{V}$ and a set of hyperedges $\mathcal{E} \subset 2^{\mathcal{V}} \setminus (\{\emptyset\} \cup \{\{v\} \mid v \in \mathcal{V}\})$ where $2^{\mathcal{V}}$ is the power set of the vertex set $\mathcal{V}$.*

We will assume all hypergraphs are undirected as in Definition 2.1. For a given hypergraph $\mathcal{H} = (\mathcal{V}, \mathcal{E})$, a hypergraph $\mathcal{G} = (\mathcal{V}', \mathcal{E}')$ is a subhypergraph of $\mathcal{H}$ if $\mathcal{V}' \subseteq \mathcal{V}$ and $\mathcal{E}' \subseteq \mathcal{E}$. For a $\mathcal{W} \subseteq \mathcal{V}$, an induced hypergraph is a subhypergraph $(\mathcal{W}, \mathcal{F} = 2^{\mathcal{W}} \cap \mathcal{E})$.

For a given hypergraph $\mathcal{H}$, we also use $\mathcal{V}_{\mathcal{H}}$ and $\mathcal{E}_{\mathcal{H}}$ to denote the sets of vertices and hyperedges of $\mathcal{H}$ respectively. According to the definition, a hyperedge is a nonempty subset of the vertices. A hypergraph with all hyperedges the same size $d$ is called a $d$-uniform hypergraph. A 2-uniform hypergraph is an undirected graph, or just graph.

When viewed combinatorially, a hypergraph can include some symmetries, which are called isomorphisms. On a hypergraph, isomorphisms are defined by bijective structure preserving maps. Such maps are a pair maps that respect hyperedge structure.

**Definition 2.2.** *For two hypergraphs $\mathcal{H}$ and $\mathcal{D}$, a structure preserving map $\rho : \mathcal{H} \to \mathcal{D}$ is a pair of maps $\rho = (\rho_{\mathcal{V}} : \mathcal{V}_{\mathcal{H}} \to \mathcal{V}_{\mathcal{D}}, \rho_{\mathcal{E}} : \mathcal{E}_{\mathcal{H}} \to \mathcal{E}_{\mathcal{D}})$ such that $\forall e \in \mathcal{E}_{\mathcal{H}}, \rho_{\mathcal{E}}(e) \triangleq \{\rho_{\mathcal{V}}(v_i) \mid v_i \in e\} \in \mathcal{E}_{\mathcal{D}}$. A hypergraph isomorphism is a structure preserving map $\rho = (\rho_{\mathcal{V}}, \rho_{\mathcal{E}})$ such that both $\rho_{\mathcal{V}}$ and $\rho_{\mathcal{E}}$ are bijective. Two hypergraphs are said to be isomorphic, denoted as $\mathcal{H} \cong \mathcal{D}$, if there exists an isomorphism between them. When $\mathcal{H} = \mathcal{D}$, an isomorphism $\rho$ is called an automorphism on $\mathcal{H}$. All the automorphisms form a group, which we denote as $Aut(\mathcal{H})$.*

A graph isomorphism is the special case of a hypergraph isomorphism between 2-uniform hypergraphs according to Definition 2.2.

A *neighborhood* $N(v) \triangleq (\bigcup_{v \in e} e, \{e : v \in e\})$ of a node $v \in \mathcal{V}$ of a hypergraph $\mathcal{H} = (\mathcal{V}, \mathcal{E})$ is the subhypergraph of $\mathcal{H}$ induced by the set of all hyperedges incident to $v$. The *degree* of $v$ is denoted $deg(v) = |\mathcal{E}_{N(v)}|$. A simple but very common symmetric hypergraph is of importance to our task, namely the neighborhood-regular hypergraph, or just regular hypergraph.

**Definition 2.3.** *A neighborhood-regular hypergraph is a hypergraph where all neighborhoods of each node are isomorphic to each other.*

A *d*-uniform neighborhood of $v$ is the set of all hyperedges of size $d$ in the neighborhood of $v$. Thus, in a neighborhood-regular hypergraph, all nodes have their $d$-uniform neighborhoods of the same degree for all $d \in \mathbb{N}$.

**Representing Higher Order Structures as Tensors** : There are many data stuctures one can define on a higher order structure like a hypergraph. An *n*-order tensor Maron et al. (2018), as a generalization of an adjacency matrix on graphs can be used to characterize the higher order connectivities. For simplicial complexes, which are hypergraphs where all subsets of a hyperedge are also hyperedges, a Hasse diagram, which is a multipartite graph induced by the poset relation of subset amongst hyperedges, or simplices, differing in exactly one node, is a common data structure Birkhoff (1940). Similarly, the star expansion matrix Agarwal et al. (2006) can be used to characterize hypergraphs up to isomorphism.

In order to define the star expansion matrix, we define the star expansion bipartite graph.

**Definition 2.4** (star expansion bipartite graph)**.** *Given a hypergraph $\mathcal{H} = (\mathcal{V}, \mathcal{E})$, the* star expansion bipartite graph $\mathcal{B}_{\mathcal{V}, \mathcal{E}}$ *is the bipartite graph with vertices $\mathcal{V} \bigsqcup \mathcal{E}$ and edges $\{(v, e) \in \mathcal{V} \times \mathcal{E} \mid v \in e\}$.*

**Definition 2.5.** *The star expansion incidence matrix $H$ of a hypergraph $\mathcal{H} = (\mathcal{V}, \mathcal{E})$ is the $|\mathcal{V}| \times 2^{|\mathcal{V}|}$ 0-1 incidence matrix $H$ where $H_{v,e} = 1$ iff $v \in e$ for $(v, e) \in \mathcal{V} \times \mathcal{E}$ for some fixed orderings on both $\mathcal{V}$ and $2^{\mathcal{V}}$.*

In practice, as data to machine learning algorithms, the matrix $H$ is sparsely represented by its nonzeros.

To study the symmetries of a given hypergraph $\mathcal{H} = (\mathcal{V}, \mathcal{E})$, we consider the permutation group on the vertices $\mathcal{V}$, denoted as $Sym(\mathcal{V})$, which acts jointly on the rows and columns of star expansion adjacency matrices. For an introduction to group theory, see Dummit & Foote (2004). We assume the rows and columns of a star expansion adjacency matrix have some canonical ordering, say lexicographic ordering, given by some prefixed ordering of the vertices. Therefore, each hypergraph $\mathcal{H}$ has a unique canonical matrix representation $H$.

We define the action of a permutation $\pi \in Sym(\mathcal{V})$ on a star expansion adjacency matrix $H$:

$$(\pi \cdot H)_{v, e = (u_1 \ldots v \ldots u_k)} \triangleq H_{\pi^{-1}(v), \pi^{-1}(e) = (\pi^{-1}(u_1) \ldots \pi^{-1}(v) \ldots \pi^{-1}(u_k))} \tag{1}$$

Based on the group action, consider the stabilizer subgroup of $Sym(\mathcal{V})$ on an incidence matrix $H$:

$$Stab_{Sym(\mathcal{V})}(H) = \{\pi \in Sym(\mathcal{V}) \mid \pi \cdot H = H\} \tag{2}$$

For simplicity we omit the lower index of $Sym(\mathcal{V})$ when the permutation group is clear from context. It can be checked that $Stab(H) \subseteq Sym(\mathcal{V})$ is a subgroup. Intuitively, $Stab(H)$ consists of all permutations that fix $H$. These are equivalent to hypergraph automorphisms on the original hypergraph $\mathcal{H}$.

**Proposition 2.1.** $Aut(\mathcal{H}) \cong Stab(H)$ *are equivalent as isomorphic groups.*

We can also define a notion of isomorphism between $k$-node sets using the stabilizers on $H$.

**Definition 2.6.** *For a given hypergraph $\mathcal{H}$ with star expansion matrix $H$, two $k$-node sets $S, T \subseteq \mathcal{V}$ are called* isomorphic, *denoted as $S \simeq T$, if $\exists \pi \in Stab(H), \pi(S) = T$ and $\pi(T) = S$.*

Such isomorphism is an equivalance relation on $k$-node sets. When $k = 1$, we have isomorphic nodes, denoted $u \cong_{\mathcal{H}} v$ for $u, v \in \mathcal{V}$. Node isomorphism is also studied as the so-called structural equivalence in Lorrain & White (1971). Furthermore, when $S \simeq T$ we can then say that there is a matching amongst the nodes in sets $S$ and $T$ so that matched nodes are isomorphic.

## 2.2 Invariance and Expressivity

For a given hypergraph $\mathcal{H} = (\mathcal{V}, \mathcal{E})$, we want to do hyperedge prediction on $\mathcal{H}$, which is to predict missing hyperedges from $k$-node sets for $k \geq 2$. Let $|\mathcal{V}| = n$, $|\mathcal{E}| = m$, and $H \in \mathbb{Z}_2^{n \times 2^n}$ be the star expansion adjacency matrix of $\mathcal{H}$. To do hyperedge prediction, we study $k$-node representations $h : [\mathcal{V}]^k \times \mathbb{Z}_2^{n \times 2^n} \to \mathbb{R}^d$ that map $k$-node sets of hypergraphs to $d$-dimensional Euclidean space. Ideally, we want a most-expressive $k$-node representation for hyperedge prediction, which is intuitively a $k$-node representation that is injective on $k$-node set isomorphism classes from $\mathcal{H}$. We break up the definition of most-expressive $k$-node representation into possessing two properties, as follows:

**Definition 2.7.** *Let $h : [\mathcal{V}]^k \times \mathbb{Z}_2^{n \times 2^n} \to \mathbb{R}^d$ be a $k$-node representation on a hypergraph $\mathcal{H}$. Let $H \in \mathbb{Z}_2^{n \times 2^n}$ be the star expansion adjacency matrix of $\mathcal{H}$ for $n$ nodes. The representation $h$ is $k$-node most expressive if $\forall S, S' \subset \mathcal{V}, |S| = |S'| = k$, the following two conditions are satisfied:*

*1. $h$ is **$k$-node invariant**: $\exists \pi \in Stab(H), \pi(S) = S' \implies h(S, H) = h(S', H)$*

*2. $h$ is **$k$-node expressive** $\nexists \pi \in Stab(H), \pi(S) = S' \implies h(S, H) \neq h(S', H)$*

The first condition of a most expressive $k$-node representation states that the representation must be well defined on the $k$ nodes up to isomorphism. The second condition requires the injectivity of our representation. These two conditions mean that the representation does not lose any information when doing prediction for missing $k$-sized hyperedges on a set of $k$ nodes.

## 2.3 Generalized Weisfeiler-Lehman-1

We describe a generalized Weisfeiler-Lehman-1 (GWL-1) hypergraph isomorphism test similar to Huang & Yang (2021); Feng et al. (2023) based on the WL-1 algorithm for graph isomorphism testing. There have been many parameterized variants of the GWL-1 algorithm implemented as neural networks, see Section 3.

Let $H$ be the star expansion matrix for a hypergraph $\mathcal{H}$. We define the GWL-1 algorithm as the following two step procedure on $H$ at iteration number $i \geq 0$.

$$f_e^0 \leftarrow \{\}, h_v^0 \leftarrow \{\}$$
$$f_e^{i+1} \leftarrow \{\{(f_e^i, h_v^i)\}\}_{v \in e}, \forall e \in \mathcal{E}_{\mathcal{H}}(H) \tag{3}$$
$$h_v^{i+1} \leftarrow \{\{(h_v^i, f_e^{i+1})\}\}_{v \in e}, \forall v \in \mathcal{V}_{\mathcal{H}}(H)$$

This is slightly different from the algorithm presented in Huang & Yang (2021) at the $f_e^{i+1}$ update step. Our update step involves an edge representation $f_e^i$, which is not present in their version. Thus our version of GWL-1 is more expressive than that in Huang & Yang (2021). However, they both possess some of the same issues that we identify. We denote $f_e^i(H)$ and $h_v^i(H)$ as the hyperedge and node ith iteration GWL-1, called $i$-GWL-1, values on an unattributed hypergraph $\mathcal{H}$ with star expansion $H$. If GWL-1 is run to convergence then we omit the iteration number $i$. We also mean this when we say $i = \infty$.

For a hypergraph $\mathcal{H}$ with star expansion matrix $H$, GWL-1 is strictly more expressive than WL-1 on $A = H \cdot D_e^{-1} \cdot H^T$ with $D_e = diag(H^T \cdot \mathbf{1}_n)$, the node to node adjacency matrix, also called the clique

expansion of $\mathcal{H}$. This follows since a triangle with its 3-cycle boundary: $T$ and a 3-cycle $C_3$ have exactly the same clique expansions. Thus WL-1 will give the same node values for both $T$ and $C_3$. GWL-1 on the star expansions $H_T$ and $H_{C_3}$, on the other hand, will identify the triangle as different from its bounding edges.

Let $f^i(H) \triangleq [f^i_{e_1}(H), \cdots f^i_{e_m}(H)]$ and $h^i(H) \triangleq [h^i_{v_1}(H), \cdots h^i_{v_n}(H)]$ be two vectors whose entries are ordered by the column and row order of $H$, respectively.

**Proposition 2.2.** *The update steps $f^i(H)$ and $h^i(H)$ of GWL-1 are permutation equivariant; For any $\pi \in Sym(\mathcal{V})$, let: $\pi \cdot f^i(H) \triangleq [f^i_{\pi^{-1}(e_1)}(H), \cdots, f^i_{\pi^{-1}(e_m)}(H)]$ and $\pi \cdot h^i(H) \triangleq [h^i_{\pi^{-1}(v_1)}(H), \cdots h^i_{\pi^{-1}(v_n)}(H)]$:*

$$\forall i \in \mathbb{N}, \pi \cdot f^i(H) = f^i(\pi \cdot H), \ \pi \cdot h^i(H) = h^i(\pi \cdot H) \tag{4}$$

Define the operator $AGG$ as a $k$-set map to representation space $\mathbb{R}^d$. Define the following representation of a $k$-node subset $S \subset \mathcal{V}$ of hypergraph $\mathcal{H}$ with star expansion matrix $H$:

$$h(S, H) = AGG[\{h^i_v(H)\}_{v \in S}] \tag{5}$$

where $h^i_v(H)$ is the node value of $i$-GWL-1 on $H$ for node $v$. The representation $h(S, H)$ preserves hyperedge isomorphism classes as shown below:

**Proposition 2.3.** *Let $h(S, H) = AGG_{v \in S}[h^i_v(H)]$ with injective $AGG$ and $h^i_v$ permutation equivariant. The representation $h(S, H)$ is $k$-node invariant but not necessarily $k$-node expressive for $S$ a set of $k$ nodes.*

It follows that we can guarantee a $k$-node invariant representation by using GWL-1. For deep learning, we parameterize $AGG$ as a universal set learner. The node representations $h^i_v(H)$ are also parameterized and rewritten into a message passing hypergraph neural network with matrix equations Huang & Yang (2021).

## 3 Related Work and Existing Issues

There are many hyperlink prediction methods. Most message passing based methods for hypergraphs are based on the GWL-1 algorithm. These include Huang & Yang (2021); Yadati et al. (2019); Feng et al. (2019); Gao et al. (2022); Dong et al. (2020); Srinivasan et al. (2021); Chien et al. (2022); Zhang et al. (2018). Examples of message passing based approaches that incorporate positional encodings on hypergraphs include SNALS Wan et al. (2021). The paper Zhang et al. (2019) uses a pair-wise node attention mechanism to do higher order link prediction. For a survey on hyperlink prediction, see Chen & Liu (2022).

There have been methods to improve the expressive power due to symmetries in graphs. In Papp & Wattenhofer (2022), substructure labeling is formally analyzed. One of the methods analyzed includes labeling fixed radius ego-graphs as in You et al. (2021); Zhang & Li (2021). Other methods include appending random node features Sato et al. (2021), labeling breadth-first and depth-first search trees Li et al. (2023b) and encoding substructures Zeng et al. (2023); Wijesinghe & Wang (2021). All of the previously mentioned methods depend on a fixed subgraph radius size. This prevents capturing symmetries that span long ranges across the graph. Zhang et al. (2023) proposes to add metric information of each node relative to all other nodes to improve WL-1. This would be very computationally expensive on hypergraphs.

Cycles are a common symmetric substructure. There are many methods that identify this symmetry. Cy2C Choi et al. is a method that encodes cycles to cliques. It has the issue that if the the cycle-basis algorithm is not permutation invariant, isomorphic graphs could get different cycle bases and thus get encoded by Cy2C differently, violating the invariance of WL-1. Similarly, the CW Network Bodnar et al. (2021) is a method that attaches cells to cycles to improve upon the distinguishing power of WL-1 for graph classification. However, inflating the input topology with cells as in Bodnar et al. (2021) would not work for link predicting since it will shift the hyperedge distribution to become much denser. Other works include cell attention networks Giusti et al. (2022) and cycle basis based methods Zhang et al. (2022). For more related work, see the Appendix.

# 4 A Characterization of GWL-1

A hypergraph can be represented by a bipartite graph $\mathcal{B}_{\mathcal{V},\mathcal{E}}$ from $\mathcal{V}$ to $\mathcal{E}$ where there is an edge $(v,e)$ in the bipartite graph iff node $v$ is incident to hyperedge $e$. This bipartite graph is called the star expansion bipartite graph.

We introduce a more structured version of graph isomorphism called a 2-color isomorphism to characterize hypergraphs. It is a map on 2-colored graphs, which are graphs that can be colored with two colors so that no two nodes in any graph with the same color are connected by an edge. We define a 2-colored isomorphism formally here:

**Definition 4.1.** *A 2-colored isomorphism is a graph isomorphism on two 2-colored graphs that preserves node colors. It is denoted by $\cong_c$.*

A bipartite graph always has a 2-coloring. In this paper, we canonically fix a 2-coloring on all star expansion bipartite graphs by assigning red to all the nodes in the node partition and and blue to all the nodes in the hyperedge partition. See Figure 2(a) as an example. We let $\mathcal{B}_{\mathcal{V}}, \mathcal{B}_{\mathcal{E}}$ be the red and blue colored nodes in $\mathcal{B}_{\mathcal{V},\mathcal{E}}$ respectively.

**Proposition 4.1.** *We have two hypergraphs $(\mathcal{V}_1, \mathcal{E}_1) \cong (\mathcal{V}_2, \mathcal{E}_2)$ iff $\mathcal{B}_{\mathcal{V}_1,\mathcal{E}_1} \cong_c \mathcal{B}_{\mathcal{V}_2,\mathcal{E}_2}$ where $\mathcal{B}_{\mathcal{V}_i,\mathcal{E}_i}$ is the star expansion bipartite graph of $(\mathcal{V}_i, \mathcal{E}_i)$*

We define a topological object for a graph originally from algebraic topology called a universal cover:

**Definition 4.2** (Hatcher (2005))**.** *The universal covering of a connected graph $G$ is a (potentially infinite) graph $\tilde{G}$ together with a map $p_G : \tilde{G} \to G$ such that:*

1. *$\forall x \in \mathcal{V}(\tilde{G})$, $p_G|_{N(x)}$ is an isomorphism onto $N(p_G(x))$.*

2. *$\tilde{G}$ is simply connected (a tree)*

We call such $p_G$ the *universal covering map* and $\tilde{G}$ the *universal cover* of $G$. A covering graph is a graph that satisfies property 1 but not necessarily 2 in Definition 4.2. The universal covering $\tilde{G}$ is essentially unique Hatcher (2005) in the sense that it can cover all connected covering graphs of $G$. Furthermore, define a rooted isomorphism $G_x \cong H_y$ as an isomorphism between graphs $G$ and $H$ that maps $x$ to $y$ and vice versa. It is a known result that:

**Theorem 4.2.** *[Krebs & Verbitsky (2015)] Let $G$ and $H$ be two connected graphs. Let $p_G : \tilde{G} \to G, p_H : \tilde{H} \to H$ be the universal covering maps of $G$ and $H$ respectively. For any $i \in \mathbb{N}$, for any two nodes $x \in G$ and $y \in H$: $\tilde{G}^i_{\tilde{x}} \cong \tilde{G}^i_{\tilde{y}}$ iff the WL-1 algorithm assigns the same value to nodes $x = p_G(\tilde{x})$ and $y = p_H(\tilde{y})$.*

We generalize the second result stated above about a topological characterization of WL-1 for GWL-1 for hypergraphs. In order to do this, we need to generalize the definition of a universal covering to suite the requirements of a bipartite star expansion graph. To do this, we lift $\mathcal{B}_{\mathcal{V},\mathcal{E}}$ to a 2-colored tree universal cover $\tilde{\mathcal{B}}_{\mathcal{V},\mathcal{E}}$ where the red/blue nodes of $\mathcal{B}_{\mathcal{V},\mathcal{E}}$ are lifted to red/blue nodes in $\tilde{\mathcal{B}}_{\mathcal{V},\mathcal{E}}$. Furthermore, the labels {} are placed on the blue nodes corresponding to the hyperedges in the lift and the labels $X_v$ are placed on all its corresponding red nodes in the lift. Let $(\tilde{\mathcal{B}}^k_{\mathcal{V},\mathcal{E}})_{\tilde{x}}$ denote the $k$-hop rooted 2-colored subtree with root $\tilde{x}$ and $p_{\mathcal{B}_{\mathcal{V},\mathcal{E}}}(\tilde{x}) = x$ for any $x \in \mathcal{V}(\mathcal{B}_{\mathcal{V},\mathcal{E}})$.

**Theorem 4.3.** *Let $\mathcal{H}_1 = (\mathcal{V}_1, \mathcal{E}_1)$ and $\mathcal{H}_2 = (\mathcal{V}_2, \mathcal{E}_2)$ be two connected hypergraphs. Let $\mathcal{B}_{\mathcal{V}_1,\mathcal{E}_1}$ and $\mathcal{B}_{\mathcal{V}_2,\mathcal{E}_2}$ be two canonically colored bipartite graphs for $\mathcal{H}_1$ and $\mathcal{H}_2$ (vertices colored red and hyperedges colored blue). Let $p_{\mathcal{B}_{\mathcal{V}_1,\mathcal{E}_1}} : \tilde{\mathcal{B}}_{\mathcal{V}_1,\mathcal{E}_1} \to \mathcal{B}_{\mathcal{V}_1,\mathcal{E}_1}, p_{\mathcal{B}_{\mathcal{V}_2,\mathcal{E}_2}} : \tilde{\mathcal{B}}_{\mathcal{V}_2,\mathcal{E}_2} \to \mathcal{B}_{\mathcal{V}_2,\mathcal{E}_2}$ be the universal coverings of $\mathcal{B}_{\mathcal{V}_1,\mathcal{E}_1}$ and $\mathcal{B}_{\mathcal{V}_2,\mathcal{E}_2}$ respectively. For any $i \in \mathbb{N}^+$, for any of the nodes $x_1 \in \mathcal{B}_{\mathcal{V}_1}, e_1 \in \mathcal{B}_{\mathcal{E}_1}$ and $x_2 \in \mathcal{B}_{\mathcal{V}_2}, e_2 \in \mathcal{B}_{\mathcal{E}_2}$:*

*$(\tilde{\mathcal{B}}^{2i-1}_{\mathcal{V}_1,\mathcal{E}_1})_{\tilde{e}_1} \cong_c (\tilde{\mathcal{B}}^{2i-1}_{\mathcal{V}_2,\mathcal{E}_2})_{\tilde{e}_2}$ iff $f^i_{e_1} = f^i_{e_2}$*

*$(\tilde{\mathcal{B}}^{2i}_{\mathcal{V}_1,\mathcal{E}_1})_{\tilde{x}_1} \cong_c (\tilde{\mathcal{B}}^{2i}_{\mathcal{V}_2,\mathcal{E}_2})_{\tilde{x}_2}$ iff $h^i_{x_1} = h^i_{x_2}$ , with $f^i_\bullet, h^i_\bullet$ the ith GWL-1 values for the hyperedges and nodes respectively where $e_1 = p_{\mathcal{B}_{\mathcal{V}_1,\mathcal{E}_1}}(\tilde{e}_1)$, $x_1 = p_{\mathcal{B}_{\mathcal{V}_1,\mathcal{E}_1}}(\tilde{x}_1)$, $e_2 = p_{\mathcal{B}_{\mathcal{V}_2,\mathcal{E}_2}}(\tilde{e}_2)$, $x_2 = p_{\mathcal{B}_{\mathcal{V}_2,\mathcal{E}_2}}(\tilde{x}_2)$.*

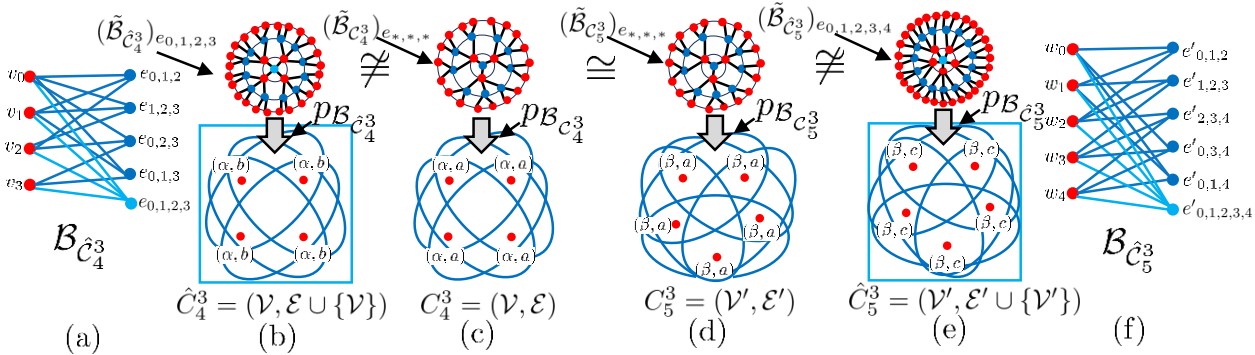

Figure 2: An illustration of hypergraph symmetry breaking. (c,d) 3-regular hypergraphs $C_4^3$, $C_5^3$ with 4 and 5 nodes respectively and their corresponding universal covers centered at any hyperedge $(\tilde{\mathcal{B}}_{C_4^3})_{e_{*,*,*}}, (\tilde{\mathcal{B}}_{C_5^3})_{e_{*,*,*}}$ with universal covering maps $p_{\mathcal{B}_{C_4^3}}, p_{\mathcal{B}_{C_5^3}}$. (b,e) the hypergraphs $\hat{C}_4^3, \hat{C}_5^3$, which are $C_4^3, C_5^3$ with $4, 5$-sized hyperedges attached to them and their corresponding universal covers and universal covering maps. (a,f) are the corresponding bipartite graphs of $\hat{C}_4^3, \hat{C}_5^3$. (c,d) are indistinguishable by GWL-1 and thus will give identical node values by Theorem 4.3. On the other hand, (b,e) gives node values which are now sensitive to the the order of the hypergraphs $4, 5$, also by Theorem 4.3.

See Figure 2 for an illustration of the universal covering of two 3-uniform neighborhood regular hypergraphs and their corresponding bipartite graphs. Notice that by Theorems 4.3, 4.2 GWL-1 reduces to computing WL-1 on the bipartite graph up to the 2-colored isomorphism.

## 4.1 A Limitation of GWL-1

For two neighborhood-regular hypergraphs $C_1$ and $C_2$, the red/blue colored universal covers $\tilde{B}_{C_1}, \tilde{B}_{C_2}$ of the star expansions of $C_1$ and $C_2$ are isomorphic, with the same GWL-1 values on all nodes. However, two neighborhood-regular hypergraphs of different order become distinguishable if a single hyperedge covering all the nodes of each neighborhood-regular hypergraph is added. Furthermore, deleting the original hyperedges, does not change the node isomorphism classes of each hypergraph. Referring to Figure 2, consider the hypergraph $\mathcal{C} = C_4^3 \sqcup C_5^3$, the hypergraph with two 3-regular hypergraphs $C_4^3$ and $C_5^3$ acting as two connected components of $\mathcal{C}$. As shown in Figure 2, the node representations of the two hypergraphs are identical due to Theorem 4.3.

Given a hypergraph $\mathcal{H}$, we define a special induced subhypergraph $\mathcal{R} \subset \mathcal{H}$ whose node set GWL-1 cannot distinguish from other such special induced subhypergraphs.

**Definition 4.3.** *A $L$-GWL-1 symmetric* induced subhypergraph $\mathcal{R} \subset \mathcal{H}$ *of $\mathcal{H}$ is a connected induced subhypergraph determined by $\mathcal{V}_\mathcal{R} \subset \mathcal{V}_\mathcal{H}$, some subset of nodes that are all indistinguishable amongst each other by $L$-GWL-1:*

$$h_u^L(H) = h_v^L(H), \forall u, v \in \mathcal{V}_\mathcal{R} \tag{6}$$

*When $L = \infty$, we call such $\mathcal{R}$ a GWL-1 symmetric induced subhypergraph. Furthermore, if $\mathcal{R} = \mathcal{H}$, then we say $\mathcal{H}$ is* GWL-1 symmetric.

This definition is similar to that of a symmetric graph from graph theory Godsil & Royle (2001), except that isomorphic nodes are determined by the GWL-1 approximator instead of an automorphism. The following observation follows from the definitions.

**Observation 1.** *A hypergraph $\mathcal{H}$ is GWL-1 symmetric if and only if it is $L$-GWL-1 symmetric for all $L \geq 1$ if and only if $\mathcal{H}$ is neighborhood regular.*

Our goal is to find GWL-1 symmetric induced subhypergraphs in a given hypergraph and break their symmetry without affecting any other nodes.

## 5 Method

Our goal is to predict higher order links in a hypergraph transductively. This can be formulated as follows:

**Problem 1.** *Given a hypergraph $\mathcal{H} = (\mathcal{V}, \mathcal{E})$ and ground truth hypergraph $\mathcal{H}_{gt} = (\mathcal{V}, \mathcal{E}_{gt}), \mathcal{E} \subset \mathcal{E}_{gt}$, where $\mathcal{E}$ is observable: Predict the existence of the unobserved hyperedges $\mathcal{E}_{gt} \setminus \mathcal{E}$.*

We will assume that the unobservable hyperedges are of the same size $k$ so that we only need to predict on $k$-node sets. In order to preserve the most information while still respecting topological structure, we aim to start with an invariant multi-node representation to predict hyperedges and increase its expressiveness, as defined in Definition 2.7. For input hypergraph $\mathcal{H}$ and its matrix representation $H$, to do the prediction of a missing hyperedge on node subsets, we use a multi-node representation $h(S, H)$ for $S \subset \mathcal{V}(H)$ as in Equation 5 due to its simplicity, guaranteed invariance, and improve its expressivity. We aim to not affect the computational complexity since message passing on hypergraphs is already quite expensive, especially on GPU memory.

Our method is a preprocessing algorithm that operates on the input hypergraph. In order to increase expressivity, we search for potentially indistinguishable regular induced subhypergraphs so that they can be replaced with hyperedges that span the subhypergraph to break the symmetries that prevent GWL-1 from being more expressive. We devise an algorithm, which is shown in Algorithm 1. It takes as input a hypergraph $\mathcal{H}$ with star expansion matrix $H$. The idea of the algorithm is to identify nodes of the same GWL-1 value that are maximally connected and use this collection of node subsets to break the symmetry of $\mathcal{H}$.

First we introduce some combinatorial definitions for hypergraph data that we will use in our algorithm:

**Definition 5.1.** *A hypergraph $\mathcal{H} = (\mathcal{V}, \mathcal{E})$ is **connected** if $\mathcal{B}_{\mathcal{V},\mathcal{E}}$ is a connected graph.*

*A **connected component** of $\mathcal{H}$ is a connected induced subhypergraph which is not properly contained in any connected subhypergraph of $\mathcal{H}$.*

**Definition 5.2.** *Chitra & Raphael (2019) A **random walk** on a hypergraph $\mathcal{H} = (\mathcal{V}, \mathcal{E})$ is a Markov chain with state space $\mathcal{V}$ with transition probabilities $P_{u,v} \triangleq \sum_{e \supset \{u,v\}: e \in \mathcal{E}} \frac{\omega(e)}{deg(u)|e|}$, where $\omega(e) : \mathcal{E} \to [0, 1]$ is some discrete probability distribution on the hyperedges. When not specified, this is the uniform distribution.*

**Definition 5.3.** *A **stationary distribution** $\pi : \mathcal{V} \to [0, 1]$ for a Markov chain with transition probabilities $P_{u,v}$ is defined by the relationship $\sum_{u \in \mathcal{V}} P_{u,v}\pi(u) = \pi(v)$.*

*For a hypergraph random walk we have the closed form: $\pi(v) = \frac{deg(v)}{\sum_{u \in \mathcal{V}} deg(u)}$ for $v \in \mathcal{V}$ assuming $\mathcal{H}$ is a connected hypergraph.*

**Algorithm:** Our method is explicitly given in Algorithm 1. For a given $L \in \mathbb{N}^+$ and any $L$-GWL-1 node value $c_L$, we construct the induced subhypergraph $\mathcal{H}_{c_L}$ from the $L$-GWL-1 class of nodes:

$$\mathcal{V}_{c_L} \triangleq \{v \in \mathcal{V} : c_L = h_v^L(H)\}, \tag{7}$$

where $h_v^L$ denotes the $L$-GWL-1 class of node $v$. We then compute the connected components of $\mathcal{H}_{c_L}$. Denote $\mathcal{C}_{c_L}$ as the set of all connected components of $\mathcal{H}_{c_L}$. If $L = \infty$, then drop $L$. Each of these connected components is a subhypergraph of $\mathcal{H}$, denoted $\mathcal{R}_{c_L,i}$ where $\mathcal{R}_{c_L,i} \subset \mathcal{H}_{c_L} \subset \mathcal{H}$ for $i = 1...|\mathcal{C}_{c_L}|$.

**Downstream Training:** After executing Algorithm 1, we collect its output $(\mathcal{R}_V, \mathcal{R}_E)$. During training, for each $i = 1...|\mathcal{C}_{c_L}|$ we randomly perturb $\mathcal{R}_{c_L,i}$ by:

- Attaching a single hyperedge that covers $\mathcal{V}_{\mathcal{R}_{c_L,i}}$ with probability $q_i$ and not attaching with probability $1 - q_i$.

- All the hyperedges in $\mathcal{R}_{c_L,i}$ are dropped or kept with probability $p$ and $1 - p$ respectively.

Let $\hat{\mathcal{H}}$ be the estimator of the input hypergraph $\mathcal{H}$ as determined by the random drop and attaching operations. Since $\hat{\mathcal{H}}$ is random, each sample of $\hat{\mathcal{H}}$ has a stationary distribution $\hat{\pi}$. The expected stationary

distribution, denoted $\mathbb{E}[\hat{\pi}]$, is the expectation of $\hat{\pi}$ over the distribution determined by $\hat{\mathcal{H}}$. We show in Proposition 5.6 that the probabilities $p, q_i, i = 1...|\mathcal{C}_{c_L}|$ can be chosen so that $\hat{\pi}$ is unbiased.

Our method is similar to the concept of adding virtual nodes Hwang et al. (2022) in graph representation learning. This is due to the equivalence between virtual nodes and hyperedges by Proposition 4.1. For a guarantee of improving expressivity, see Lemma 5.2 and Theorems 5.3, 5.4. For an illustration of the data augmentation, see Figure 2.

Alternatively, downstream training using the output of Algorithm 1 can be done. Similar to subgraph NNs, this is done by applying an ensemble of models Alsentzer et al. (2020); Papp et al. (2021); Tan et al. (2023), with each model trained on transformations of $\mathcal{H}$ with its symmetric subhypergraphs randomly replaced. This, however, is computationally expensive.

---

**Algorithm 1:** A Symmetry Finding Algorithm

    **Data:** Hypergraph $\mathcal{H} = (\mathcal{V}, \mathcal{E})$, represented by its star expansion matrix $H$. $L \in \mathbb{N}^+$ is the number of iterations to run GWL-1.

    **Result:** A pair of collections: $(\mathcal{R}_V = \{\mathcal{V}_{R_j}\}, \mathcal{R}_E = \cup_j \{\mathcal{E}_{R_j}\})$ where $R_j$ are disconnected subhypergraphs exhibiting symmetry in $\mathcal{H}$ that are indistinguishable by $L$-GWL-1.

**1** $U_L \leftarrow h_v^L(H); \mathcal{G}_L \leftarrow \{U_L[v] : \forall v \in \mathcal{V}\}$ ;         /* $U_L[v]$ is the $L$-GWL-1 value of node $v \in \mathcal{V}$. */

**2** $\mathcal{B}_{\mathcal{V}_{\mathcal{H}}, \mathcal{E}_{\mathcal{H}}} \leftarrow Bipartite(\mathcal{H})$ /* Construct the bipartite graph from $\mathcal{H}$. */

**3** $\mathcal{R}_V \leftarrow \{\}; \mathcal{R}_E \leftarrow \{\}$

**4 for** $c_L \in \mathcal{G}_L$ **do**

**5**      $\mathcal{V}_{c_L} \leftarrow \{v \in \mathcal{V} : U_L[v] = c_L\}, \mathcal{E}_{c_L} \leftarrow \{e \in \mathcal{E} : u \in \mathcal{V}_{c_L}, \forall u \in e\}$

**6**      $\mathcal{C}_{c_L} \leftarrow \text{ConnectedComponents}(\mathcal{H}_{c_L} = (\mathcal{V}_{c_L}, \mathcal{E}_{c_L}))$

**7**      **for** $\mathcal{R}_{c_L,i} \in \mathcal{C}_{c_L}$ **do**

           /* There should be at least 3 nodes to form a nontrivial hyperedge */

**8**          **if** $|\mathcal{V}_{\mathcal{R}_{c_L,i}}| \geq 3$ **then**

**9**              $\mathcal{R}_V \leftarrow \mathcal{R}_V \cup \{\mathcal{V}_{\mathcal{R}_{c_L,i}}\}; \mathcal{R}_E \leftarrow \mathcal{R}_E \cup \mathcal{E}_{\mathcal{R}_{c_L,i}}$

**10**          **end**

**11**      **end**

**12 end**

**13 return** $(\mathcal{R}_V, \mathcal{R}_E)$

---

**Algorithm Guarantees:** We show some guarantees for the output of Algorithm 1.

**Notation:** Let $\mathcal{H} = (\mathcal{V}, \mathcal{E})$ be a hypergraph with star expansion matrix $H$ and let $(\mathcal{R}_{\mathcal{V}}, \mathcal{R}_{\mathcal{E}})$ be the output of Algorithm 1 on $H$ for $L \in \mathbb{N}^+$. Let $\hat{\mathcal{H}}_L \triangleq (\mathcal{V}, \mathcal{E} \cup \mathcal{R}_V)$ be $\mathcal{H}$ after adding all the hyperedges from $\mathcal{R}_{\mathcal{V}}$ and let $\hat{H}_L$ be the star expansion matrix of the resulting hypergraph $\hat{\mathcal{H}}_L$. Let $V_{c_L,s} \triangleq \{v \in \mathcal{V}_{c_L} : v \in R, R \in \mathcal{C}_{c_L}, |\mathcal{V}_R| = s\}$ be the set of all nodes of $L$-GWL-1 class $c_L$ belonging to a connected component in $\mathcal{C}_{c_L}$ of $s \geq 1$ nodes in $\mathcal{H}_{c_L}$, the induced subhypergraph of $L$-GWL-1. Let $\mathcal{G}_L \triangleq \{h_v^L(H) : v \in \mathcal{V}\}$ be the set of all $L$-GWL-1 values on $H$. Let $\mathcal{S}_{c_L} \triangleq \{|\mathcal{V}_{\mathcal{R}_{c_L,i}}| : \mathcal{R}_{c_L,i} \in \mathcal{C}_{c_L}\}$ be the set of node set sizes of the connected components in $\mathcal{H}_{c_L}$.

**Proposition 5.1.** *If $L = \infty$, for any GWL-1 node value $c$ computed on $\mathcal{H}$, all connected component subhypergraphs $\mathcal{R}_{c,i} \in \mathcal{C}_c$ are GWL-1 symmetric as hypergraphs.*

**Lemma 5.2.** *If $L \in \mathbb{N}^+$ is small enough so that after running Algorithm 1 on $L$, for any $L$-GWL-1 node class $c_L$ on $\mathcal{V}$ none of the discovered $\mathcal{V}_{\mathcal{R}_{c_L,i}}$ are within $L$ hyperedges away from any $\mathcal{V}_{\mathcal{R}_{c_L,j}}$ for all $i, j \in 1...|\mathcal{C}_{c_L}|, i \neq j$, then after forming $\hat{\mathcal{H}}_L$, the new $L$-GWL-1 node classes of $\mathcal{V}_{\mathcal{R}_{c_L,i}}$ for $i = 1...\mathcal{C}_{c_L}$ in $\hat{\mathcal{H}}_L$ are all the same class $c'_L$ but are distinguishable from $c_L$ depending on $|\mathcal{V}_{\mathcal{R}_{c_L,i}}|$.*

We also have the following guarantee on the number of pairs of distinguishable $k$-node sets on $\hat{\mathcal{H}}$:

**Theorem 5.3.** *Let $|\mathcal{V}| = n, L \in \mathbb{N}^+$. If $vol(v) \triangleq \sum_{e \in \mathcal{E}: e \ni v} |e| = O(\log^{\frac{1-\epsilon}{4L}} n), \forall v \in \mathcal{V}$ for any constant $\epsilon > 0$; $|\mathcal{S}_{c_L}| \leq S, \forall c_L \in \mathcal{C}_L$, $S$ constant, and $|V_{c_L,s}^L| = O(\frac{n^\epsilon}{\log^{\frac{1}{2k}}(n)}), \forall s \in \mathcal{C}_{c_L}$ , then for $k \in \mathbb{N}^+$ and $k$-tuple $C = (c_{L,1}...c_{L,k}), c_{L,i} \in \mathcal{G}_L, i = 1..k$ there exists $\omega(n^{2k\epsilon})$ many pairs of $k$-node sets $S_1 \neq S_2$ such*

*that $(h_u^L(H))_{u \in S_1} = (h_{v \in S_2}^L(H)) = C$, as ordered k-tuples, while $h(S_1, \hat{H}_L) \neq h(S_2, \hat{H}_L)$ also by $L$ steps of GWL-1.*

We show that our algorithm increases expressivity (Definition 2.7) for $h(S, H)$ of Equation 5.

**Theorem 5.4** (Invariance and Expressivity). *If $L = \infty$, GWL-1 enhanced by Algorithm 1 is still invariant to node isomorphism classes of $\mathcal{H}$ and can be strictly more expressive than GWL-1 to determine node isomorphism classes.*

Proposition 5.5 provides the time complexity of our algorithm.

**Proposition 5.5** (Complexity). *Algorithm 1 runs in time $O(nnz(H)L + (n + m))$, which is order linear in the size of the input star expansion matrix $H$ for hypergraph $\mathcal{H} = (\mathcal{V}, \mathcal{E})$, if $L$ is independent of $nnz(H)$, where $n = |\mathcal{V}|$, $nnz(H) = vol(\mathcal{V}) \triangleq \sum_{v \in \mathcal{V}} deg(v)$ and $m = |\mathcal{E}|$.*

Since Algorithm 1 runs in time linear in the size of the input when $L$ is constant, in practice it only takes a small fraction of the training time for hypergraph neural networks.

For the downstream training, we show that there are Bernoulli hyperedge drop/attachment probabilities $p, q_i$ respectively for each $\mathcal{R}_{c_L,i}$ so that the stationary distribution doesn't change. This shows that our data augmentation can still preserve the low frequency random walk signal.

**Proposition 5.6.** *For a connected hypergraph $\mathcal{H} = (\mathcal{V}, \mathcal{E})$, let $(\mathcal{R}_V, \mathcal{R}_E)$ be the output of Algorithm 1 on $\mathcal{H}$. Then there are Bernoulli probabilities $p, q_i$ for $i = 1...|\mathcal{R}_V|$ for attaching a covering hyperedge so that $\hat{\pi}$ is an unbiased estimator of $\pi$.*

## 6 Evaluation

| PR-AUC ↑ | Baseline | Ours | Baseln.+edrop |
|---|---|---|---|
| HGNN | 0.98 ± 0.03 | 0.99 ± 0.08 | 0.96 ± 0.02 |
| HGNNP | 0.98 ± 0.02 | 0.98 ± 0.09 | 0.96 ± 0.10 |
| HNHN | 0.98 ± 0.01 | 0.96 ± 0.07 | 0.97 ± 0.04 |
| HyperGCN | 0.98 ± 0.07 | 0.98 ± 0.11 | 0.98 ± 0.03 |
| UniGAT | 0.99 ± 0.06 | 0.99 ± 0.03 | 0.99 ± 0.07 |
| UniGCN | 0.99 ± 0.00 | 0.99 ± 0.03 | 0.99 ± 0.08 |
| UniGIN | 0.87 ± 0.02 | 0.86 ± 0.10 | 0.85 ± 0.08 |
| UniSAGE | 0.86 ± 0.04 | 0.86 ± 0.05 | 0.84 ± 0.09 |

(a) CAT-EDGE-DAWN

| PR-AUC ↑ | Baseline | Ours | Baseln.+edrop |
|---|---|---|---|
| HGNN | 0.90 ± 0.13 | 1.00 ± 0.00 | 0.90 ± 0.13 |
| HGNNP | 0.90 ± 0.09 | 1.00 ± 0.07 | 1.00 ± 0.03 |
| HNHN | 0.90 ± 0.09 | 0.91 ± 0.02 | 0.90 ± 0.08 |
| HyperGCN | 1.00 ± 0.00 | 1.00 ± 0.03 | 1.00 ± 0.02 |
| UniGAT | 0.90 ± 0.06 | 1.00 ± 0.03 | 1.00 ± 0.06 |
| UniGCN | 1.00 ± 0.01 | 0.91 ± 0.01 | 0.82 ± 0.09 |
| UniGIN | 0.90 ± 0.12 | 0.95 ± 0.06 | 0.90 ± 0.11 |
| UniSAGE | 0.90 ± 0.16 | 1.00 ± 0.08 | 0.90 ± 0.17 |

(b) CAT-EDGE-MUSIC-BLUES-REVIEWS

| PR-AUC ↑ | Baseline | Ours | Baseln.+ edrop |
|---|---|---|---|
| HGNN | 0.96 ± 0.10 | 0.98 ± 0.05 | 0.96 ± 0.04 |
| HGNNP | 0.96 ± 0.05 | 0.98 ± 0.09 | 0.97 ± 0.07 |
| HNHN | 0.96 ± 0.02 | 0.97 ± 0.08 | 0.97 ± 0.06 |
| HyperGCN | 0.93 ± 0.05 | 0.98 ± 0.07 | 0.96 ± 0.09 |
| UniGAT | 0.96 ± 0.01 | 0.98 ± 0.14 | 0.97 ± 0.04 |
| UniGCN | 0.96 ± 0.04 | 0.96 ± 0.11 | 0.96 ± 0.09 |
| UniGIN | 0.97 ± 0.03 | 0.97 ± 0.11 | 0.96 ± 0.05 |
| UniSAGE | 0.96 ± 0.10 | 0.96 ± 0.10 | 0.96 ± 0.02 |

(c) CONTACT-HIGH-SCHOOL

| PR-AUC ↑ | Baseline | Ours | Baseln.+edrop |
|---|---|---|---|
| HGNN | 0.95 ± 0.03 | 0.96 ± 0.01 | 0.95 ± 0.03 |
| HGNNP | 0.95 ± 0.02 | 0.96 ± 0.09 | 0.96 ± 0.07 |
| HNHN | 0.94 ± 0.07 | 0.97 ± 0.10 | 0.95 ± 0.05 |
| HyperGCN | 0.97 ± 0.01 | 0.97 ± 0.05 | 0.96 ± 0.08 |
| UniGAT | 0.95 ± 0.02 | 0.98 ± 0.14 | 0.98 ± 0.02 |
| UniGCN | 0.96 ± 0.00 | 0.97 ± 0.14 | 0.97 ± 0.10 |
| UniGIN | 0.95 ± 0.09 | 0.97 ± 0.02 | 0.95 ± 0.05 |
| UniSAGE | 0.96 ± 0.08 | 0.95 ± 0.05 | 0.96 ± 0.02 |

(d) CONTACT-PRIMARY-SCHOOL

| PR-AUC ↑ | Baseline | Ours | Baseln.+edrop |
|---|---|---|---|
| HGNN | 0.95 ± 0.07 | 0.97 ± 0.08 | 0.96 ± 0.07 |
| HGNNP | 0.95 ± 0.07 | 0.96 ± 0.02 | 0.96 ± 0.01 |
| HNHN | 0.94 ± 0.01 | 0.97 ± 0.02 | 0.95 ± 0.06 |
| HyperGCN | 0.92 ± 0.01 | 0.94 ± 0.06 | 0.94 ± 0.08 |
| UniGAT | 0.94 ± 0.08 | 0.98 ± 0.14 | 0.97 ± 0.08 |
| UniGCN | 0.97 ± 0.08 | 0.97 ± 0.14 | 0.97 ± 0.06 |
| UniGIN | 0.93 ± 0.07 | 0.94 ± 0.11 | 0.93 ± 0.09 |
| UniSAGE | 0.93 ± 0.07 | 0.93 ± 0.08 | 0.92 ± 0.04 |

(e) EMAIL-EU

| PR-AUC ↑ | Baseline | Ours | Baseln.+edrop |
|---|---|---|---|
| HGNN | 0.75 ± 0.09 | 0.85 ± 0.09 | 0.71 ± 0.14 |
| HGNNP | 0.83 ± 0.09 | 0.85 ± 0.08 | 0.85 ± 0.04 |
| HNHN | 0.72 ± 0.09 | 0.82 ± 0.03 | 0.74 ± 0.09 |
| HyperGCN | 0.87 ± 0.08 | 0.83 ± 0.05 | 1.00 ± 0.07 |
| UniGAT | 0.80 ± 0.09 | 0.83 ± 0.03 | 0.78 ± 0.05 |
| UniGCN | 0.84 ± 0.08 | 0.89 ± 0.10 | 0.71 ± 0.07 |
| UniGIN | 0.69 ± 0.14 | 0.76 ± 0.05 | 0.61 ± 0.11 |
| UniSAGE | 0.72 ± 0.11 | 0.71 ± 0.10 | 0.64 ± 0.10 |

(f) CAT-EDGE-MADISON-RESTAURANTS

Table 1: Transductive hyperedge prediction PR-AUC scores on six different hypergraph datasets. The highest scores per HyperGNN architecture (row) is colored. Red text denotes the highest average scoring method. Orange text denotes a two-way tie and brown text denotes a three-way tie. All datasets involve predicting hyperedges of size 3.

We evaluate our method on higher order link prediction with many of the standard hypergraph neural network methods. Due to potential class imbalance, we measure the PR-AUC of higher order link prediction on the hypergraph datasets. These datasets are: CAT-EDGE-DAWN, CAT-EDGE-MUSIC-BLUES-REVIEWS, CONTACT-HIGH-SCHOOL, CONTACT-PRIMARY-SCHOOL, EMAIL-EU, CAT-EDGE-MADISON-RESTAURANTS. These datasets range from representing social interactions as they develop over time to collections of reviews to drug combinations before overdose. We also evaluate on the AMHERST41 dataset, which is a graph dataset. All of our datasets are unattributed hypergraphs/graphs.

**Data Splitting:** For the hypergraph datasets, each hyperedge in it is paired with a timestamp (a real number). These timestamps are a physical time for which a higher order interaction, represented by a

| PR-AUC ↑ | HGNN | HGNNP | HNHN | HyperGCN | UniGAT | UniGCN | UniGIN | UniSAGE |
|---|---|---|---|---|---|---|---|---|
| Ours | 0.73 ± 0.10 | 0.61 ± 0.05 | 0.64 ± 0.06 | 0.71 ± 0.09 | 0.72 ± 0.08 | 0.70 ± 0.08 | 0.73 ± 0.03 | 0.73 ± 0.06 |
| HyperGNN Baseline | 0.62 ± 0.09 | 0.62 ± 0.10 | 0.63 ± 0.04 | 0.71 ± 0.07 | 0.70 ± 0.06 | 0.69 ± 0.07 | 0.73 ± 0.06 | 0.73 ± 0.09 |
| HyperGNN Baseln.+edrop | 0.61 ± 0.03 | 0.61 ± 0.03 | 0.61 ± 0.09 | 0.71 ± 0.06 | 0.71 ± 0.02 | 0.69 ± 0.05 | 0.73 ± 0.09 | 0.73 ± 0.04 |
| APPNP | 0.42 ± 0.07 | 0.42 ± 0.07 | 0.42 ± 0.07 | 0.42 ± 0.07 | 0.42 ± 0.07 | 0.42 ± 0.07 | 0.42 ± 0.07 | 0.42 ± 0.07 |
| APPNP+edrop | 0.42 ± 0.03 | 0.42 ± 0.03 | 0.42 ± 0.03 | 0.42 ± 0.03 | 0.42 ± 0.03 | 0.42 ± 0.03 | 0.42 ± 0.03 | 0.42 ± 0.03 |
| GAT | 0.49 ± 0.06 | 0.49 ± 0.06 | 0.49 ± 0.06 | 0.49 ± 0.06 | 0.49 ± 0.06 | 0.49 ± 0.06 | 0.49 ± 0.06 | 0.49 ± 0.06 |
| GAT+edrop | 0.49 ± 0.06 | 0.49 ± 0.06 | 0.49 ± 0.06 | 0.49 ± 0.06 | 0.49 ± 0.06 | 0.49 ± 0.06 | 0.49 ± 0.06 | 0.49 ± 0.06 |
| GCN2 | 0.56 ± 0.12 | 0.56 ± 0.12 | 0.56 ± 0.12 | 0.56 ± 0.12 | 0.56 ± 0.12 | 0.56 ± 0.12 | 0.56 ± 0.12 | 0.56 ± 0.12 |
| GCN2+edrop | 0.54 ± 0.02 | 0.54 ± 0.02 | 0.54 ± 0.02 | 0.54 ± 0.02 | 0.54 ± 0.02 | 0.54 ± 0.02 | 0.54 ± 0.02 | 0.54 ± 0.02 |
| GCN | 0.40 ± 0.03 | 0.40 ± 0.03 | 0.40 ± 0.03 | 0.40 ± 0.03 | 0.40 ± 0.03 | 0.40 ± 0.03 | 0.40 ± 0.03 | 0.40 ± 0.03 |
| GCN+edrop | 0.65 ± 0.04 | 0.65 ± 0.04 | 0.65 ± 0.04 | 0.65 ± 0.04 | 0.65 ± 0.04 | 0.65 ± 0.04 | 0.65 ± 0.04 | 0.65 ± 0.04 |
| GIN | 0.73 ± 0.10 | 0.73 ± 0.10 | 0.73 ± 0.10 | 0.73 ± 0.10 | 0.73 ± 0.10 | 0.73 ± 0.10 | 0.73 ± 0.10 | 0.73 ± 0.10 |
| GIN+edrop | 0.73 ± 0.10 | 0.73 ± 0.10 | 0.73 ± 0.10 | 0.73 ± 0.10 | 0.73 ± 0.10 | 0.73 ± 0.10 | 0.73 ± 0.10 | 0.73 ± 0.10 |
| GraphSAGE | 0.44 ± 0.01 | 0.44 ± 0.01 | 0.44 ± 0.01 | 0.44 ± 0.01 | 0.44 ± 0.01 | 0.44 ± 0.01 | 0.44 ± 0.01 | 0.44 ± 0.01 |
| GraphSAGE+edrop | 0.44 ± 0.10 | 0.44 ± 0.10 | 0.44 ± 0.10 | 0.44 ± 0.10 | 0.44 ± 0.10 | 0.44 ± 0.10 | 0.44 ± 0.10 | 0.44 ± 0.10 |

Table 2: PR-AUC on graph dataset AMHERST41. Each column is a comparison of the baseline PR-AUC scores against the PR-AUC score for our method (first row) applied to a standard HyperGNN architecture. The coloring scheme is the same as in Table 1.

hyperedge, occurs. We form a train-val-test split by letting the train be the hyperedges associated with the 80th percentile of timestamps, the validation be the hyperedges associated with the timestamps in between the 80th and 85th percentiles. The test hyperedges are the remaining hyperedges. The train validation and test datasets thus form a partition of the nodes. We do the task of hyperedge prediction for sets of nodes of size 3, also known as triangle prediction. Half of the size 3 hyperedges in each of train, validation and test are used as positive examples. For each split, we select random subsets of nodes of size 3 that do not form hyperedges for negative sampling. We maintain positive/negative class balance by sampling the same number of negative samples as positive samples. Since the test distribution comes from later time stamps than those in training, there is a possibility that certain datasets are out-of-distribution if the hyperedge distribution changes.

For the graph dataset, the single graph is deterministically split into 80/5/15 for train/val/test. We remove 10% of the edges in training and let them be positive examples $P_{tr}$ to predict. For validation and test, we remove 50% of the edges from both validation and test to set as the positive examples $P_{val}, P_{te}$ to predict. For train, validation, and test, we sample $|P_{tr}|, |P_{val}|, |P_{te}|$ negative link samples from the links of train, validation and test.

## 6.1 Architecture and Training

Our algorithm serves as a preprocessing step for selective data augmentation. Given a single training hypergraph $\mathcal{H}$, the Algorithm 1 is applied and during training, the identified hyperedges of the symmetric induced subhypergraphs of $\mathcal{H}$ are randomly replaced with single hyperedges that cover all the nodes of each induced subhypergraph. Each symmetric subhypergraph has a $p = 0.5$ probability of being selected. To get a large set of symmetric subhypergraphs, we run 2 iterations of GWL-1.

We implement $h(S, H)$ from Equation 5 as follows. Upon extracting the node representations from the hypergraph neural network, we use a multi-layer-perceptron (MLP) on each node representation, sum across such compositions, then apply a final MLP layer after the aggregation. We use the binary cross entropy loss on this multi-node representation for training. We always use 5 layers of hyperGNN convolutions, a hidden dimension of 1024, and a learning rate of 0.01.

## 6.2 Higher Order Link Prediction Results

We show in Table 1 the comparison of PR-AUC scores amongst the baseline methods of HGNN, HGNNP, HNHN, HyperGCN, UniGIN, UniGAT, UniSAGE, their hyperedge dropped versions, and "Our" method, which preprocesses the hypergraph to break symmetry during training. For the hyperedge drop baselines, there is a uniform 50% chance of dropping any hyperedge. We use the Laplacian eigenmap Belkin & Niyogi (2003) positional encoding on the clique expansion of the input hypergraph. This is common practice in (hyper)link prediction and required for using a hypergraph neural network on an unattributed hypergraph.

We show in Table 2 the PR-AUC scores on the AMHREST41. Along with HyperGNN architectures we use for the hypergraph experiments, we also compare with standard GNN architectures: APPNP Gasteiger et al. (2018), GAT Veličković et al. (2017), GCN2 Chen et al. (2020), GCN Kipf & Welling (2016a), GIN Xu et al. (2018), and GraphSAGE Hamilton et al. (2017). For every HyperGNN/GNN architecture, we also apply drop-edge Rong et al. (2019) to the input graph and use this also as baseline. The number of layers of each GNN is set to 5 and the hidden dimension at 1024. For APPNP and GCN2, one MLP is used on the initial node positional encodings.

Overall, our method performs well across a diverse range of higher order network datasets. We observe that our method can often outperform the baseline of not performing any data perturbations as well as the same baseline with uniformly random hyperedge dropping. Our method has an added advantage of being explainable since our algorithm works at the data level. There was also not much of a concern for computational time since our algorithm runs in time $O(nnz(H) + n + m)$, which is optimal since it is the size of the input.

### 6.3 Empirical Observations on the Components Discovered by the Algorithm

As we are primarily concerned with symmetries in a hypergraph, we empirically measure the size and frequency of the components found by the Algorithm for real-world datasets. For the real-world datasets listed in Appendix D, in Figure 3a, we plot the fraction of connected components of the same $L$-GWL-1 value ($L = 2$) that are large enough to be used by Algorithm 1 as a function of the number of nodes of the hypergraph. We notice that the fraction of connected components is not large, however every dataset has a nonzero fraction. On the right, in Figure 3b we show the distribution of the sizes of the connected components found by Algorithm 1. We see that, on average, the connected components are at least an order of magnitude smaller compared to the total number of nodes. Common to both plots, the graph datasets appear to have more nodes and a consistent fraction and size of components, while the hypergraph datasets have higher variance in the fraction of components, which is expected since there are more possibilities for the connections in a hypergraph. In terms of the number of identified connected components, there are at least

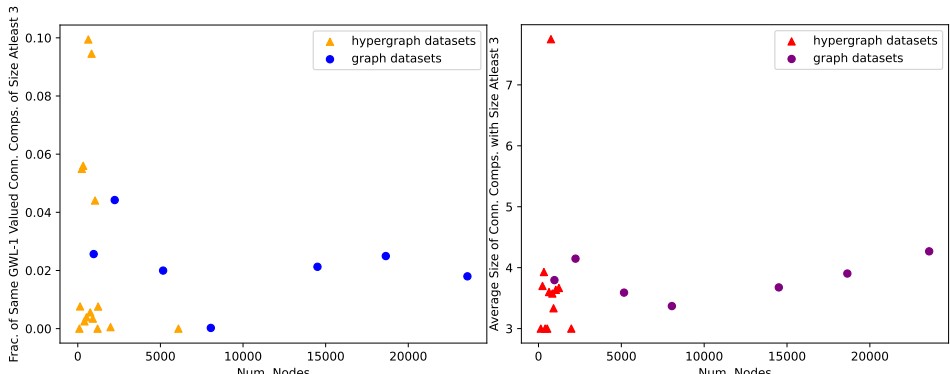

(a) Fraction of components of size atleast 3 selected by Algorithm 1.
(b) Average size of components of size atleast 3 from Algorithm 1.

Figure 3

exponentially many interventions that can be imposed on the hypergraph from simply dropping components. Thus, even finding just 10 components result in at least $2^{10} \approx 10^3$ many counterfactual hypergraphs. It is known, that a large set of data augmentations during learning improves learner generalization.

## 7 Conclusion

We have characterized and identified the limitations of GWL-1, a hypergraph isomorphism testing algorithm that underlies many existing HyperGNN architectures. A common issue with distinguishing regular hypergraphs exists. In fact more generally, maximally connected subsets of nodes that share the same value of

GWL-1, which act like regular hypergraphs, are indistinguishable. To address this issue while respecting the structure of a hypergraph, we have devised a preprocessing algorithm that improves the expressivity of any GWL-1 based learner. The algorithm searches for indistinguishable regular subhypergraphs and simplifies them by a single hyperedge that covers the nodes of the subhypergraph. We perform extensive experiments to evaluate the effectiveness of our approach and make empirical observations about the output of the algorithm on hypergraph data.

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

# Appendix

## A   More Background

We discuss in this section about the basics of graph representation learning and link prediction. Graphs are hypergraphs with all hyperedges of size 2. Simplicial complexes and hypergraphs are generalizations of graphs. We also discuss more related work.

### A.1   Graph Neural Networks and Weisfeiler-Lehman 1

The Weisfeiler-Lehman (WL-1) algorithm is an isomorphism testing approximation algorithm. It involves repeatedly message passing all nodes with their neighbors, a step called node label refinement. The WL-1 algorithm never gives false negatives when predicting whether two graphs are isomorphic. In other words, two isomorphic graphs are always indistinguishable by WL-1.

The WL-1 algorithm is the following successive vertex relabeling applied until convergence on a graph $G = (X, A)$ (a pair of the set of node attributes and the graph's adjacency structure):

$$
\begin{aligned}
h_v^0 &\leftarrow X_v, \forall v \in \mathcal{V}_G \\
h_v^{i+1} &\leftarrow \{\{(h_v^i, h_u^i)\}\}_{u \in Nbr_A(v)}, \forall v \in \mathcal{V}_G
\end{aligned}
\tag{8}
$$

The algorithm terminates after the vertex labels converge. For graph isomorphism testing, the concatenation of the histograms of vertex labels for each iteration is output as the graph representation. Since we are only concerned with node isomorphism classes, we ignore this step and just consider the node labels $h_v^i$ for every $v \in \mathcal{V}_C$.

The WL-1 isomorphism test can be characterized in terms of rooted tree isomorphisms between the universal covers for connected graphs Krebs & Verbitsky (2015). There have also been characterizations of WL-1 in terms of counting homomorphisms Knill (2013) as well as the Wasserstein Distance Chen et al. (2022) and Markov chains Chen et al. (2023).

A graph neural network (GNN) is a message passing based node representation learner modeled after the WL-1 algorithm. It has the important inductive bias of being equivariant to node indices. As a neural model of the WL-1 algorithm, it learns neural weights common across all nodes in order to obtain a vector representation for each node. A GNN must use some initial node attributes in order to update its neural weights. There are many variations on GNNs, including those that improve the distinguishing power beyond WL-1. For two surveys on the GNNs and their applications, see Zhou et al. (2020); Wu et al. (2020).

### A.2   Link Prediction

The task of link prediction on graphs involves the prediction of the existence of links. There are two kinds of link prediction. There is transductive link prediction where the same nodes are used for all of train validation and testing. There is also inductive link prediction where the test validation and training nodes can all be disjoint. Some existing works on link prediction include Zhang & Chen (2017). Higher order link prediction is a generalization of link prediction to hypergraph data.

A common way to do link prediction is to compute a node-based GNN and for a pair of nodes, aggregate, similar to in graph auto encoders Kipf & Welling (2016b), the node representations in any target pair in order to obtain a 2-node representation. Such aggregations are of the form:

$$
h(S = \{u, v\}) = \sigma(h_u \cdot h_v)
\tag{9}
$$

where $S$ is a pair of nodes. As shown in Proposition B.4, this guaranteems an equivariant 2-node representation but can often give false predictions even with a fully expressive node-based GNN Wang et al. (2023). A common remedy for this problem is to introduce positional encodings such as SEAL Wang et al. (2022) and DistanceEncoding Li et al. (2020). Positional encodings encode the relative distances amongst nodes via a low distortion embedding for example. In the related work section we have gone over many of these embeddings. We have also used these in our evaluation since they are common practice and must exist to compute a hypergraph neural network if there are no ground truth node attributes. According to Srinivasan & Ribeiro (2019), fully expressive pairwise node representations, as defined by 2-node invariance and expressivity, can be represented by some fully expressive positional embedding, which is a positional embedding that is injective on the node pair isomorphism classes. It is not clear how one would achieve this in practice, however. Another remedy is to increase the expressive power of WL-1 to WL-2 for link prediction Hu et al. (2022).

### A.3 More Related Work

The work of Wei et al. (2022) also does a data augmentation scheme. It considers randomly dropping edges and generating data through a generative model on hypergraphs. The work of Lee & Shin (2022) also performs data augmentation on a hypergraph so that homophilic relationships are maintained. It does this through contrastive losses at the node to node, hyperedge to hyperedge and intra hyperedge level. Neither of these methods provide guarantees for their data augmentations.

As mentioned in the main text, an ensemble of neural networks can be used with a drop-out Baldi & Sadowski (2014) like method on the output of the Algorithm. Subgraph neural networks Alsentzer et al. (2020); Tan et al. (2023) are ensembles of models on subgraphs of the input graph.

Some more of the many existing hypergraph neural network architectures include: Kim et al. (2022); Cai et al. (2022); Chien et al. (2021); Bai et al. (2021); Li et al. (2023a); Arya et al. (2020).

# B   Proofs

In this section we provide the proofs for all of the results in the main paper along with some additional theory.

## B.1   Hypergraph Isomorphism

We first repeat the definition of a hypergraph and its corresponding matrix representation called the star expansion matrix::

**Definition B.1.** *An undirected hypergraph is a pair $\mathcal{H} = (\mathcal{V}, \mathcal{E})$ consisting of a set of vertices $\mathcal{V}$ and a set of hyperedges $\mathcal{E} \subset 2^{\mathcal{V}} \setminus (\{\emptyset\} \cup \{\{v\} \mid v \in \mathcal{V}\})$ where $2^{\mathcal{V}}$ is the power set of the vertex set $\mathcal{V}$.*

**Definition B.2.** *The star expansion incidence matrix $H$ of a hypergraph $\mathcal{H} = (\mathcal{V}, \mathcal{E})$ is the $|\mathcal{V}| \times 2^{|\mathcal{V}|}$ 0-1 incidence matrix $H$ where $H_{v,e} = 1$ iff $v \in e$ for $(v, e) \in \mathcal{V} \times \mathcal{E}$ for some fixed orderings on both $\mathcal{V}$ and $2^{\mathcal{V}}$.*

We recall the definition of an isomorphism between hypergraphs:

**Definition B.3.** *For two hypergraphs $\mathcal{H}$ and $\mathcal{D}$, a structure preserving map $\rho : \mathcal{H} \to \mathcal{D}$ is a pair of maps $\rho = (\rho_{\mathcal{V}} : \mathcal{V}_{\mathcal{H}} \to \mathcal{V}_{\mathcal{D}}, \rho_{\mathcal{E}} : \mathcal{E}_{\mathcal{H}} \to \mathcal{E}_{\mathcal{D}})$ such that $\forall e \in \mathcal{E}_{\mathcal{H}}, \rho_{\mathcal{E}}(e) \triangleq \{\rho_{\mathcal{V}}(v_i) \mid v_i \in e\} \in \mathcal{E}_{\mathcal{D}}$. A hypergraph isomorphism is a structure preserving map $\rho = (\rho_{\mathcal{V}}, \rho_{\mathcal{E}})$ such that both $\rho_{\mathcal{V}}$ and $\rho_{\mathcal{E}}$ are bijective. Two hypergraphs are said to be isomorphic, denoted as $\mathcal{H} \cong \mathcal{D}$, if there exists an isomorphism between them. When $\mathcal{H} = \mathcal{D}$, an isomorphism $\rho$ is called an automorphism on $\mathcal{H}$. All the automorphisms form a group, which we denote as $Aut(\mathcal{H})$.*

The action of $\pi \in Sym(\mathcal{V})$ on the star expansion adjacency matrix $H$ is repeated here for convenience:

$$(\pi \cdot H)_{v,e=(u_1 \ldots v \ldots u_k)} \triangleq H_{\pi^{-1}(v), \pi^{-1}(e) = (\pi^{-1}(u_1) \ldots \pi^{-1}(v) \ldots \pi^{-1}(u_k))} \tag{10}$$

Based on the group action, consider the stabilizer subgroup of $Sym(\mathcal{V})$ on the star expansion adjacency matrix $H$ defined as follows:

$$Stab_{Sym(\mathcal{V})}(H) = \{\pi \in Sym(\mathcal{V}) \mid \pi \cdot H = H\} \tag{11}$$

For simplicity we omit the lower index when the permutation group is clear from the context. It can be checked that $Stab(H) \leq Sym(\mathcal{V})$ is a subgroup. Intuitively, $Stab(H)$ consists of all permuations that leave $H$ fixed.

For a given hypergraph $\mathcal{H} = (\mathcal{V}, \mathcal{E})$, there is a relationship between the group of hypergraph automorphisms $Aut(\mathcal{H})$ and the stabilizer group $Stab(H)$ on the star expansion adjacency matrix.

**Proposition B.1.** *$Aut(\mathcal{H}) \cong Stab(H)$ are equivalent as isomorphic groups.*

*Proof.* Consider $\rho \in Aut(\mathcal{H})$, define the map $\Phi : \rho \mapsto \pi := \rho|_{\mathcal{V}(\mathcal{H})}$. The group element $\pi \in Sym(\mathcal{V})$ acts as a stabilizer of $H$ since for any entry $(v, e)$ in $H$, $H_{\pi^{-1}(v), \pi^{-1}(e)} = (\pi \cdot H)_{v,e} = 1$ iff $\pi^{-1}(e) \in \mathcal{E}_{\mathcal{H}}$ iff $e \in \mathcal{E}_{\mathcal{H}}$ iff $H_{v,e} = 1 = H_{\pi \circ \pi^{-1}(v), \pi \circ \pi^{-1}(e)}$. Since $(v, e)$ was arbitrary, $\pi$ preserves the positions of the nonzeros.

We can check that $\Phi$ is a well defined injective homorphism as a restriction map. Furthermore it is surjective since for any $\pi \in Stab(H)$, we must have $H_{v,e} = 1$ iff $(\pi \cdot H)_{v,e} = H_{\pi^{-1}(v), \pi^{-1}(e)} = 1$ which is equivalent to $v \in e \in \mathcal{E}$ iff $\pi(v) \in \pi(e) \in \mathcal{E}$ which implies $e \in \mathcal{E}$ iff $\pi(e) \in \mathcal{E}$. Thus $\Phi$ is a group isomorphism from $Aut(\mathcal{H})$ to $Stab(H)$ □

In other words, to study the symmetries of a given hypergraph $\mathcal{H}$, we can equivalently study the automorphisms $Aut(\mathcal{H})$ and the stabilizer permutations $Stab(H)$ on its star expansion adjacency matrix $H$. Intuitively, the stabilizer group $0 \leq Stab(H) \leq Sym(\mathcal{V})$ characterizes the symmetries in a graph. When the graph has rich symmetries, say a complete graph, $Stab(H) = Sym(\mathcal{V})$ can be as large as the whole permutaion group.

Nontrivial symmetries can be represented by isomorphic node sets which we define as follow:

**Definition B.4.** *For a given hypergraph $\mathcal{H}$ with star expansion matrix $H$, two $k$-node sets $S, T \subseteq \mathcal{V}$ are called* isomorphic*, denoted as $S \simeq T$, if $\exists \pi \in Stab(H), \pi(S) = T$ and $\pi(T) = S$.*

When $k = 1$, we have isomorphic nodes, denoted $u \cong_{\mathcal{H}} v$ for $u, v \in \mathcal{V}$. Node isomorphism is also studied as the so-called structural equivalence in Lorrain & White (1971). Furthermore, if $S \simeq T$ we can then say that there is a matching amongst the nodes in the two node subsets so that matched nodes are isomorphic.

**Definition B.5.** *A $k$-node representation $h$ is **$k$-permutation equivariant** if:*

*for all $\pi \in Sym(\mathcal{V})$, $S \in 2^{\mathcal{V}}$ with $|S| = k$: $h(\pi \cdot S, H) = h(S, \pi \cdot H)$*

**Proposition B.2.** *If $k$-node representation $h$ is $k$-permutation equivariant, then $h$ is $k$-node invariant.*

*Proof.* given $S, S' \in \mathcal{C}$ with $|S| = |S'| = k$,

if there exists a $\pi \in Stab(H)$ (meaning $\pi \cdot H = H$) and $\pi(S) = S'$ then

$$
\begin{aligned}
h(S', H) &= h(S', \pi \cdot H) \text{ (by } \pi \cdot H = H) \\
&= h(S, H) \text{ (by } k\text{-permutation equivariance of } h \text{ and } \pi(S) = S')
\end{aligned}
\tag{12}
$$

$\square$

## B.2 Properties of GWL-1

Here are the steps of the GWL-1 algorithm on the star expansion matrix $H$ with node attributes $X$ is repeated here for convenience:

$$f_e^0 \leftarrow \{\}, h_v^0 \leftarrow X_v$$
$$f_e^{i+1} \leftarrow \{\{(f_e^i, h_v^i)\}\}_{v \in e}, \forall e \in \mathcal{E}(H)$$
$$h_v^{i+1} \leftarrow \{\{(h_v^i, f_e^{i+1})\}\}_{v \in e}, \forall v \in \mathcal{V}(H) \tag{13}$$

Where $\mathcal{E}(H)$ denotes the nonzero columns of $H$ and $\mathcal{V}(H)$ denotes the rows of $H$.

We make the following observations about each of the two steps of the GWL-1 algorithm:

**Observation 2.**

$$\{\{(f_e^i, h_v^i)\}\}_{v \in e} = \{\{(f'^i_e, h'^i_v)\}\}_{v \in e} \text{ iff } (f_e^i, \{\{h_v^i\}\}_{v \in e}) = (f'^i_e, \{\{h'^i_v\}\}_{v \in e}) \forall e \in \mathcal{E}(H) \text{ and} \tag{14a}$$

$$\{\{(h_v^i, f_e^{i+1})\}\}_{v \in e} = \{\{(h'^i_v, f'^{i+1}_e)\}\}_{v \in e} \text{ iff } (h_v^i, \{\{f_e^{i+1}\}\}_{v \in e}) = (h'^i_v, \{\{f'^{i+1}_e\}\}_{v \in e}) \forall v \in \mathcal{V}(H) \tag{14b}$$

*Proof.* Equation 14a follows since

$$\{\{(f_e^i, h_v^i)\}\}_{v \in e} = \{\{(f'^i_e, h'^i_v)\}\}_{v \in e} \forall e \in \mathcal{E}(H) \tag{15a}$$

$$\text{iff } f_e^i = f'^i_e \text{ and } \{\{h_v^i\}\}_{v \in e} = \{\{h'^i_v\}\}_{v \in e} \forall e \in \mathcal{E}(H) \tag{15b}$$

$$\text{iff } (f_e^i, \{\{h_v^i\}\}_{v \in e}) = (f'^i_e, \{\{h'^i_v\}\}_{v \in e}) \forall e \in \mathcal{E}(H) \tag{15c}$$

For Equation 14b, we have:

$$\{\{(h_v^i, f_e^{i+1})\}\}_{v \in e} = \{\{(h'^i_v, f'^{i+1}_e)\}\}_{v \in e} \forall v \in \mathcal{V}(H) \tag{16a}$$

$$\text{iff } \{\{(h_v^i, \{\{(f_e^i, h_u^i)\}\}_{u \in e})\}\}_{v \in e} = \{\{(h'^i_v, \{\{(f'^i_e, h'^i_u)\}\}_{u \in e})\}\}_{v \in e} \forall v \in \mathcal{V}(H) \tag{16b}$$

$$\text{iff } h_v^i = h'^i_v \text{ and } \{\{(f_e^i, h_u^i)\}\}_{u \in e, v \in e} = \{\{(f'^i_e, h'^i_u)\}\}_{u \in e, v \in e} \forall v \in \mathcal{V}(H) \tag{16c}$$

$$\text{iff } h_v^i = h'^i_v \text{ and } \{\{f_e^{i+1}\}\} = \{\{f'^{i+1}_e\}\} \forall v \in \mathcal{V}(H) \tag{16d}$$

These follow by the definition of multiset equality and since there is no loss of information upon factoring out a constant tuple entry of each pair in the multisets. $\square$

**Proposition B.3.** *The update steps of GWL-1:* $f^i(H) \triangleq [f_{e_1}^i(H), \cdots f_{e_m}^i(H)]$ *and* $h^i(H) \triangleq [h_{v_1}^i(H), \cdots h_{v_n}^i(H)]$, *are permutation equivariant; in other words, For any* $\pi \in Sym(\mathcal{V})$, *let* $\pi \cdot f^i(H) \triangleq [f_{\pi^{-1}(e_1)}^i(H), \cdots, f_{\pi^{-1}(e_m)}^i(H)]$ *and* $\pi \cdot h^i(H) \triangleq [h_{\pi^{-1}(v_1)}^i(H), \cdots h_{\pi^{-1}(v_n)}^i(H)]$, *we have* $\forall i \in \mathbb{N}, \pi \cdot f^i(H) = f^i(\pi \cdot H)$ *and* $\pi \cdot h^i(H) = h^i(\pi \cdot H)$

*Proof.* We prove by induction on $i$:

Base case, $i = 0$:

$[\pi \cdot f^0(H)]_{e=\{v_1...v_k\}} = \{\} = f^0_{\pi^{-1}(e)=\{\pi^{-1}(v_1)...\pi^{-1}(v_k)\}}(H) = f_e^0(\pi \cdot H)$ since the $\pi$ cannot affect a list of empty sets and the definition of the action of $\pi$ on $H$ as defined in Equation 10.

$[\pi \cdot h^0(H)]_v = [\pi \cdot X]_v = X_{\pi^{-1}(v)} = h^0_{\pi^{-1}(v)}(H) = h_v^0(\pi \cdot H)$ by definition of the group action $Sym(\mathcal{V})$ acting on the node indices of a node attribute tensor as defined in Equation 10.

Induction Hypothesis:

$$[\pi \cdot f^i(H)]_e = f^i_{\pi^{-1}(e)}(H) = f_e^i(\pi \cdot H) \text{ and } [\pi \cdot h^i(H)]_v = h^i_{\pi^{-1}(v)}(H) = h_v^i(\pi \cdot H) \tag{17}$$

Induction Step:

$$[\pi \cdot h^{i+1}(H)]_v = \{\{([\pi \cdot h^i(H)]_v, [\pi \cdot f^{i+1}(H)]_e)\}\}_{v \in e}$$
$$= \{\{([\pi \cdot h^i(H)]_v, \{\{([\pi \cdot f^i(H)]_e, [\pi \cdot h^i(H)]_u)\}\}_{u \in e})\}\}_{v \in e}$$
$$= \{\{(h^i_v(\pi \cdot H), \{\{(f^i_e(\pi \cdot H), h^i_u(\pi \cdot H))\}\}_{u \in e}\}\}_{v \in e}$$
$$= h^{i+1}_v(\pi \cdot H) \tag{18}$$

$$[\pi \cdot f^{i+1}(H)]_e = \{\{([\pi \cdot f^i(H)]_e, [\pi \cdot h^i(H)]_v)\}\}_{v \in e}$$
$$= \{\{(f^i_e(\pi \cdot H), h^i_v(\pi \cdot H)\}\}_{v \in e}$$
$$= f^{i+1}_e(\pi \cdot H) \tag{19}$$

$\square$

**Definition B.6.** *Let $h : [\mathcal{V}]^k \times \mathbb{Z}_2^{n \times 2^n} \to \mathbb{R}^d$ be a $k$-node representation on a hypergraph $\mathcal{H}$. Let $H \in \mathbb{Z}_2^{n \times 2^n}$ be the star expansion adjacency matrix of $\mathcal{H}$ for $n$ nodes. The representation $h$ is $k$-node most expressive if $\forall S, S' \subset \mathcal{V}, |S| = |S'| = k$, the following two conditions are satisfied:*

1. *$h$ is **$k$-node invariant**: $\exists \pi \in Stab(H), \pi(S) = S' \implies h(S, H) = h(S', H)$*

2. *$h$ is **$k$-node expressive** $\nexists \pi \in Stab(H), \pi(S) = S' \implies h(S, H) \neq h(S', H)$*

Let $AGG$ be a permutation invariant map from a set of node representations to $\mathbb{R}^d$.

**Proposition B.4.** *Let $h(S, H) = AGG_{v \in S}[h^i_v(H)]$ with injective $AGG$ and $h^i_v$ permutation equivariant. The representation $h(S, H)$ is $k$-node invariant but not necessarily $k$-node expressive for $S$ a set of $k$ nodes.*

*Proof.* $\exists \pi \in Stab(H)$ s.t. $\pi(S) = S', \pi \cdot H = H$

$\Rightarrow \pi(v_i) = v'_i$ for $i = 1...|S|, \pi \cdot H = H$

$\Rightarrow h^i_{\pi(v)}(H) = h^i_v(\pi \cdot H) = h^i_v(H)$ (By permutation equivariance of $h^i_v$ and $\pi \cdot H = H$)

$\Rightarrow AGG_{v \in S}[h^i_v(H)] = AGG_{v' \in S'}[h^i_{v'}(H)]$ (By Proposition B.2 and AGG being permutation invariant)

The converse, that $h(S, H)$ is $k$-node expressive, is not necessarily true since we cannot guarantee $h(S, H) = h(S', H)$ implies the existence of a permutation that maps $S$ to $S'$ (see Zhang et al. (2021)). $\square$

A hypergraph can be represented by a bipartite graph $\mathcal{B}_{\mathcal{V}, \mathcal{E}}$ from $\mathcal{V}$ to $\mathcal{E}$ where there is an edge $(v, e)$ in the bipartite graph iff node $v$ is incident to hyperedge $e$. This bipartite graph $\mathcal{B}_{\mathcal{V}, \mathcal{E}}$ is called the star expansion bipartite graph.

We introduce a more structured version of graph isomorphism called a 2-color isomorphism to characterize hypergraphs. It is a map on 2-colored graphs, which are graphs that can be colored with two colors so that no two nodes in any graph with the same color are connected by an edge. We define a 2-colored isomorphism formally here:

**Definition B.7.** *A 2-colored isomorphism is a graph isomorphism on two 2-colored graphs that preserves node colors. In particular, between two graphs $G_1$ and $G_2$ the vertices of one color in $G_1$ must map to vertices of the same color in $G_2$. It is denoted by $\cong_c$.*

A bipartite graph must always have a 2-coloring. In fact, the 2-coloring with all the nodes in the node bipartition colored red and all the nodes in the hyperedge bipartition colored blue forms a canonical 2-coloring of $\mathcal{B}_{\mathcal{V}, \mathcal{E}}$. Assume that all star expansion bipartite graphs are canonically 2-colored.

**Proposition B.5.** *We have two hypergraphs $(\mathcal{V}_1, \mathcal{E}_1) \cong (\mathcal{V}_2, \mathcal{E}_2)$ iff $\mathcal{B}_{\mathcal{V}_1, \mathcal{E}_1} \cong_c \mathcal{B}_{\mathcal{V}_2, \mathcal{E}_2}$ where $\mathcal{B}_{\mathcal{V}, \mathcal{E}}$ is the star expansion bipartite graph of $(\mathcal{V}, \mathcal{E})$*

*Proof.* Denote $L(\mathcal{B}_{\mathcal{V}_i,\mathcal{E}_i})$ as the left hand (red) bipartition of $\mathcal{B}_{\mathcal{V}_i,\mathcal{E}_i}$ to represent the nodes $\mathcal{V}_i$ of $(\mathcal{V}_i,\mathcal{E}_i)$ and $R(\mathcal{B}_{\mathcal{V}_i,\mathcal{E}_i})$ as the right hand (blue) bipartition of $\mathcal{B}_{\mathcal{V}_i,\mathcal{E}_i}$ to represent the hyperedges $\mathcal{E}_i$ of $(\mathcal{V}_i,\mathcal{E}_i)$. We use the left/right bipartition and $\mathcal{V}_i/\mathcal{E}_i$ interchangeably since they are in bijection.

$\Rightarrow$ If there is an isomorphism $\pi : \mathcal{V}_1 \to \mathcal{V}_2$, this means

- $\pi$ is a bijection and

- has the structure preserving property that $(u_1...u_k) \in \mathcal{E}_1$ iff $(\pi(u_1)...\pi(u_k)) \in \mathcal{E}_2$.

We may induce a 2-colored isomorphism $\pi^* : \mathcal{V}(\mathcal{B}_{\mathcal{V}_1,\mathcal{E}_1}) \to \mathcal{V}(\mathcal{B}_{\mathcal{V}_1,\mathcal{E}_1})$ so that $\pi^*|_{L(\mathcal{B}_{\mathcal{V}_1,\mathcal{E}_1})} = \pi$ where equality here means that $\pi^*|_{L(\mathcal{B}_{\mathcal{V}_1,\mathcal{E}_1})}$ acts on $L(\mathcal{B}_{\mathcal{V}_1,\mathcal{E}_1})$ the same way that $\pi$ does on $\mathcal{V}_1$. Furthermore $\pi^*$ has the property that $\pi^*|_{R(\mathcal{B}_{\mathcal{V}_1,\mathcal{E}_1})}(u_1...u_k) = (\pi(u_1)...\pi(u_k)), \forall(u_1...u_k) \in \mathcal{E}_1$, following the structure preserving property of isomorphism $\pi$.

The map $\pi^*$ is a bijection by definition of being an extension of a bijection.

The map $\pi^*$ is also a 2-colored map since it maps $L(\mathcal{B}_{\mathcal{V}_1,\mathcal{E}_1})$ to $L(\mathcal{B}_{\mathcal{V}_2,\mathcal{E}_2})$ and $R(\mathcal{B}_{\mathcal{V}_1,\mathcal{E}_1})$ to $R(\mathcal{B}_{\mathcal{V}_2,\mathcal{E}_2})$.

We can also check that the map is structure preserving and thus a 2-colored isomorphism since $(u_i, (u_1...u_i...u_k)) \in \mathcal{E}(\mathcal{B}_{\mathcal{V}_1,\mathcal{E}_1}), \forall i = 1...k$ iff $(u_i \in \mathcal{V}_1$ and $(u_1...u_i...u_k) \in \mathcal{E}_1)$ iff $\pi(u_i) \in \mathcal{V}_2$ and $(\pi(u_1)...\pi(u_i)...\pi(u_k)) \in \mathcal{E}_2$ iff $(\pi^*(u_i), (\pi^*(u_1,...u_i,...u_k)) \in \mathcal{E}(\mathcal{B}_{\mathcal{V}_2,\mathcal{E}_2}), \forall i = 1...k$. This follows from $\pi$ being structure preserving and the definition of $\pi^*$.

$\Leftarrow$ If there is a 2-colored isomorphism $\pi^* : \mathcal{B}_{\mathcal{V}_1,\mathcal{E}_1} \to \mathcal{B}_{\mathcal{V}_2,\mathcal{E}_2}$ then it has the properties that

- $\pi^*$ is a bijection,

- (is 2-colored): $\pi^*|_{L(\mathcal{B}_{\mathcal{V}_1,\mathcal{E}_1})} : L(\mathcal{B}_{\mathcal{V}_1,\mathcal{E}_1}) \to L(\mathcal{B}_{\mathcal{V}_2,\mathcal{E}_2})$ and $\pi^*|_{R(\mathcal{B}_{\mathcal{V}_1,\mathcal{E}_1})} : R(\mathcal{B}_{\mathcal{V}_1,\mathcal{E}_1}) \to R(\mathcal{B}_{\mathcal{V}_2,\mathcal{E}_2})$

- (it is structure preserving): $(u_i, (u_1...u_i...u_k)) \in \mathcal{E}(\mathcal{B}_{\mathcal{V}_1,\mathcal{E}_1}), \forall i = 1...k$ iff $(\pi^*(u_i), \pi^*(u_1...u_i...u_k)) \in \mathcal{E}(\mathcal{B}_{\mathcal{V}_2,\mathcal{E}_2}), \forall i = 1...k$ .

This then means that we may induce a $\pi : \mathcal{V}_1 \to \mathcal{V}_2$ so that $\pi = \pi^*|_{L(\mathcal{B}_{\mathcal{V}_1,\mathcal{E}_1})}$.

We can check that $\pi$ is a bijection since $\pi$ is the 2-colored bijection $\pi^*$ restricted to $L(\mathcal{B}_{\mathcal{V}_1,\mathcal{E}_1})$, thus remaining a bijection.

We can also check that $\pi$ is structure preserving. This means that $(u_1...u_k) \in \mathcal{E}_1$ iff $(u_i, (u_1...u_i...u_k)) \in \mathcal{E}(\mathcal{B}_{\mathcal{V}_1,\mathcal{E}_1}) \forall i = 1...k$ iff $(\pi^*(u_i), (\pi^*(u_1...u_i...u_k))) \in \mathcal{E}(\mathcal{B}_{\mathcal{V}_2,\mathcal{E}_2}) \forall i = 1...k$ iff $(\pi^*(u_1...u_k)) \in R(\mathcal{B}_{\mathcal{V}_2,\mathcal{E}_2})$ iff $(\pi(u_1)...\pi(u_k)) \in \mathcal{E}_2$ $\qquad \square$

We define a topological object for a graph originally from algebraic topology called a universal cover:

**Definition B.8.** *(Hatcher (2005)) A universal covering of a connected graph $G$ is a (potentially infinite) graph $\tilde{G}$, s.t. there is a map $p_G : \tilde{G} \to G$ called the universal covering map where:*

*1. $\forall x \in \mathcal{V}(\tilde{G})$, $p_G|_{N(x)}$ is an isomorphism onto $N(p_G(x))$.*

*2. $\tilde{G}$ is simply connected (a tree)*

A covering graph is a graph that satisfies property 1 but not necessarily property 2 in Definition B.8. It is known that a universal covering $\tilde{G}$ covers all the graph covers of the graph $G$. Let $T_x^G$ denote a tree with root $x$. Furthermore, define a rooted isomorphism $G_x \cong H_y$ as an isomorphism between graphs $G$ and $H$ that maps $x$ to $y$ and vice versa. We will use the following result to prove a characterization of GWL-1:

**Lemma B.6** (Krebs & Verbitsky (2015)). *Let $T$ and $S$ be trees and $x \in V(T)$ and $y \in V(S)$ be their vertices of the same degree with neighborhoods $N(x) = \{x_1, ..., x_k\}$ and $N(y) = \{y_1, ..., y_k\}$. Let $r \geq 1$. Suppose that $T_x^{r-1} \cong S_y^{r-1}$ and $T_{x_i}^r \cong S_{y_i}^r$ for all $i \leq k$. Then $T_x^{r+1} \cong S_y^{r+1}$.*

A universal cover of a 2-colored bipartite graph is still 2 colored. When we lift nodes $v$ and hyperedge nodes $e$ to their universal cover, we keep their respective red and blue colors.

Define a rooted colored isomorphism $T^k_{\tilde{e}_1} \cong_c T^k_{\tilde{e}_2}$ as a colored tree isomorphism where blue/red node $\tilde{e}_1/\tilde{v}_1$ maps to blue/red node $\tilde{e}_2/\tilde{v}_2$ and vice versa.

In fact, Lemma B.6 holds for 2-colored isomorphisms, which we show below:

**Lemma B.7.** *Let $T$ and $S$ be 2-colored trees and $x \in V(T)$ and $y \in V(S)$ be their vertices of the same degree with neighborhoods $N(x) = \{x_1, ..., x_k\}$ and $N(y) = \{y_1, ..., y_k\}$. Let $r \geq 1$. Suppose that $T^{r-1}_x \cong_c S^{r-1}_y$ and $T^r_{x_i} \cong_c S^r_{y_i}$ for all $i \leq k$. Then $T^{r+1}_x \cong_c S^{r+1}_y$.*

*Proof.* Certainly 2-colored isomorphisms are rooted isomorphisms on 2-colored trees. The converse is true if the roots match in color since recursively all descendants of the root must match in color.

If $T^{r-1}_x \cong_c S^{r-1}_y$ and $T^r_{x_i} \cong_c S^r_{y_i}$ for all $i \leq k$ and $N(x) = \{x_1...x_k\}, N(y) = \{y_1..y_k\}$, the roots $x$ and $y$ must match in color. The neighborhoods $N(x)$ and $N(y)$ then must both be of the opposing color. Since rooted colored isomorphisms are rooted isomorphisms, we must have $T^{r-1}_x \cong S^{r-1}_y$ and $T^r_{x_i} \cong S^r_{y_i}$ for all $i \leq k$. By Lemma B.6, we have $T^{r+1}_x \cong S^{r+1}_y$. Once the roots match in color, a rooted tree isomorphism is the same as a rooted 2-colored tree isomorphism. Thus, since $x$ and $y$ share the same color, $T^{r+1}_x \cong_c S^{r+1}_y$ □

**Theorem B.8.** *Let $\mathcal{H}_1 = (\mathcal{V}_1, \mathcal{E}_1)$ and $\mathcal{H}_2 = (\mathcal{V}_2, \mathcal{E}_2)$ be two connected hypergraphs. Let $\mathcal{B}_{\mathcal{V}_1, \mathcal{E}_1}$ and $\mathcal{B}_{\mathcal{V}_2, \mathcal{E}_2}$ be two canonically colored bipartite graphs for $\mathcal{H}_1$ and $\mathcal{H}_2$ (vertices colored red and hyperedges colored blue)*

*For any $i \in \mathbb{N}^+$, for any of the nodes $x_1 \in \mathcal{B}_{\mathcal{V}_1}, e_1 \in \mathcal{B}_{\mathcal{V}_1, \mathcal{E}_1}$ and $x_2 \in \mathcal{B}_{\mathcal{V}_1}, e_2 \in \mathcal{B}_{\mathcal{V}_2, \mathcal{E}_2}$:*

*$(\tilde{\mathcal{B}}^{2i-1}_{\mathcal{V}_1, \mathcal{E}_1})_{\tilde{e}_1} \cong_c (\tilde{\mathcal{B}}^{2i-1}_{\mathcal{V}_2, \mathcal{E}_2})_{\tilde{e}_2}$ iff $f^i_{e_1} = f^i_{e_2}$*

*$(\tilde{\mathcal{B}}^{2i}_{\mathcal{V}_1, \mathcal{E}_1})_{\tilde{x}_1} \cong_c (\tilde{\mathcal{B}}^{2i}_{\mathcal{V}_2, \mathcal{E}_2})_{\tilde{x}_2}$ iff $h^i_{x_1} = h^i_{x_2}$, with $f^i_\bullet, h^i_\bullet$ the ith GWL-1 values for the hyperedges and nodes respectively where $e_1 = p_{\mathcal{B}_{\mathcal{V}_1, \mathcal{E}_1}}(\tilde{e}_1)$, $x_1 = p_{\mathcal{B}_{\mathcal{V}_1, \mathcal{E}_1}}(\tilde{x}_1)$, $e_2 = p_{\mathcal{B}_{\mathcal{V}_1, \mathcal{E}_1}}(\tilde{e}_2)$, $x_2 = p_{\mathcal{B}_{\mathcal{V}_1, \mathcal{E}_1}}(\tilde{x}_2)$. The maps $p_{\mathcal{B}_{\mathcal{V}_1, \mathcal{E}_1}} : \tilde{\mathcal{B}}_{\mathcal{V}_1, \mathcal{E}_1} \to \mathcal{B}_{\mathcal{V}_1, \mathcal{E}_1}, p_{\mathcal{B}_{\mathcal{V}_2, \mathcal{E}_2}} : \tilde{\mathcal{B}}_{\mathcal{V}_2, \mathcal{E}_2} \to \mathcal{B}_{\mathcal{V}_2, \mathcal{E}_2}$ are the universal covering maps of $\mathcal{B}_{\mathcal{V}_1, \mathcal{E}_1}$ and $\mathcal{B}_{\mathcal{V}_2, \mathcal{E}_2}$ respectively.*

*Proof.* We prove by induction:

Let $T^k_{\tilde{e}_1} := (\tilde{\mathcal{B}}^k_{\mathcal{V}_1, \mathcal{E}_1})_{\tilde{e}_1}$ where $\tilde{e}_1$ is a pullback of a hyperedge, meaning $p_{\mathcal{B}_{\mathcal{V}_1, \mathcal{E}_2}}(\tilde{e}_1) = e_1$. Similarly, let $T^k_{\tilde{e}_2} := (\tilde{\mathcal{B}}^k_{\mathcal{V}_2, \mathcal{E}_2})_{\tilde{e}_2}$, $T^k_{\tilde{x}_1} := (\tilde{\mathcal{B}}^k_{\mathcal{V}_1, \mathcal{E}_1})_{\tilde{x}_1}$, $T^k_{\tilde{x}_2} := (\tilde{\mathcal{B}}^k_{\mathcal{V}_2, \mathcal{E}_2})_{\tilde{x}_2}$, $\forall k \in \mathbb{N}$, where $\tilde{e}_1, \tilde{e}_2, \tilde{x}_1, \tilde{x}_2$ are the respective pullbacks of $e_1, e_2, x_1, x_2$.

Define an (2-colored) isomorphism of multisets to mean that there exists a bijection between the two multisets so that each element in one multiset is (2-colored) isomorphic with exactly one other element in the other multiset.

By Observation 3 we can rewrite GWL-1 as:

$$f^0_e \leftarrow \{\}, h^0_v \leftarrow X_v \tag{20}$$

$$f^{i+1}_e \leftarrow (f^i_e, \{\{h^i_v\}\}_{v \in e}) \forall e \in \mathcal{E}_\mathcal{H} \tag{21}$$

$$h^{i+1}_v \leftarrow (h^i_v, \{\{f^{i+1}_e\}\}_{v \in e}) \forall v \in \mathcal{V}_\mathcal{H} \tag{22}$$

Base Case $i = 1$:

$$T^1_{\tilde{e}_1} \cong_c T^1_{\tilde{e}_2} \text{ iff } (T^0_{\tilde{e}_1} \cong_c T^0_{\tilde{e}_2} \text{ and } \{\{T^0_{\tilde{x}_1}\}\}_{\tilde{x}_1 \in N(\tilde{e}_1)} \cong_c \{\{T^0_{\tilde{x}_2}\}\}_{\tilde{x}_2 \in N(\tilde{e}_2)}) \text{ (By Lemma B.7)} \tag{23a}$$

$$\text{iff } (f^0_{e_1} = f^0_{e_2} \text{ and } \{\{h^0_{x_1}\}\} = \{\{h^0_{x_2}\}\}) \text{ (By Equation 20)} \tag{23b}$$

$$\text{iff } f^1_{e_1} = f^1_{e_2} \text{ (By Equation 21)} \tag{23c}$$

$$T^2_{\tilde{x}_1} \cong_c T^2_{\tilde{x}_2} \text{ iff } (T^0_{\tilde{x}_1} \cong_c T^0_{\tilde{x}_2} \text{ and } \{\{T^1_{\tilde{e}_1}\}\}_{\tilde{e}_1 \in N(\tilde{x}_1)} \cong_c \{\{T^1_{\tilde{e}_2}\}\}_{\tilde{e}_2 \in N(\tilde{x}_2)}) \text{ (By Lemma B.7)} \tag{24a}$$

$$\text{iff } (h_{e_1}^0 = h_{e_2}^0 \text{ and } \{\{f_{x_1}^1\}\} = \{\{f_{x_2}^1\}\}) \text{ (By Equation 20)} \tag{24b}$$

$$\text{iff } f_{e_1}^1 = f_{e_2}^1 \text{ (By Equation 22)} \tag{24c}$$

Induction Hypothesis: For $i \geq 1$, $T_{\tilde{e}_1}^{2i-1} \cong_c T_{\tilde{e}_2}^{2i-1}$ iff $f_{e_1}^i = f_{e_2}^i$ and $T_{\tilde{x}_1}^{2i} \cong_c T_{\tilde{x}_2}^{2i}$ iff $h_{x_1}^i = h_{x_2}^i$

Induction Step:

$$T_{\tilde{e}_1}^{2i+1} \cong_c T_{\tilde{e}_2}^{2i+1} \text{ iff } (T_{\tilde{e}_1}^{2i-1} \cong_c T_{\tilde{e}_2}^{2i-1} \text{ and } \{\{T_{\tilde{x}_1}^{2i}\}\}_{\tilde{x}_1 \in N(\tilde{e}_1)} \cong_c \{\{T_{\tilde{x}_2}^{2i}\}\}_{\tilde{x}_2 \in N(\tilde{e}_2)}) \text{ (By Lemma B.7)} \tag{25a}$$

$$\text{iff } (f_{e_1}^i = f_{e_2}^i \text{ and } \{\{h_{x_1}^i\}\} = \{\{h_{x_2}^i\}\}) \text{ (By Induction Hypothesis)} \tag{25b}$$

$$\text{iff } f_{e_1}^{i+1} = f_{e_2}^{i+1} \text{ (By Equation 21)} \tag{25c}$$

$$T_{\tilde{x}_1}^{2i} \cong_c T_{\tilde{x}_2}^{2i} \text{ iff } (T_{\tilde{x}_1}^{2i-2} \cong_c T_{\tilde{x}_2}^{2i-2} \text{ and } \{\{T_{\tilde{e}_1}^{2i-1}\}\}_{\tilde{e}_1 \in N(\tilde{x}_1)} \cong_c \{\{T_{\tilde{e}_2}^{2i-1}\}\}_{\tilde{e}_2 \in N(\tilde{x}_2)}) \text{ (By Lemma B.7)} \tag{26a}$$

$$\text{iff } (h_{e_1}^i = h_{e_2}^i \text{ and } \{\{f_{x_1}^i\}\} = \{\{f_{x_2}^i\}\}) \text{ (By Equation 20)} \tag{26b}$$

$$\text{iff } h_{x_1}^i = h_{x_2}^i \text{ (By Equation 22)} \tag{26c}$$

$\square$

**Observation 3.** *If the node values for nodes $x$ and $y$ from GWL-1 for $i$ iterations on two hypergraphs $\mathcal{H}_1$ and $\mathcal{H}_2$ are the same, then for all $j$ with $0 \leq j \leq i$, the node values for GWL-1 for $j$ iterations on $x$ and $y$ also agree. In particular $\deg(x) = \deg(y)$.*

*Proof.* There is a 2-color isomorphism on subtrees $(\tilde{\mathcal{B}}_{\mathcal{V}_1,\mathcal{E}_1}^j)_{\tilde{x}}$ and $(\tilde{\mathcal{B}}_{\mathcal{V}_2,\mathcal{E}_2}^j)_{\tilde{y}}$ of the $i$-hop subtrees of the universal covers rooted about nodes $x \in \mathcal{V}_1$ and $y \in \mathcal{V}_2$ for $0 \leq j \leq i$ since $(\tilde{\mathcal{B}}_{\mathcal{V}_1,\mathcal{E}_1}^i)_{\tilde{x}} \cong_c (\tilde{\mathcal{B}}_{\mathcal{V}_2,\mathcal{E}_2}^i)_{\tilde{y}}$. By Theorem B.8, we have that GWL-1 returns the same value for $x$ and $y$ for each $0 \leq j \leq i$. $\square$

**Proposition B.9.** *If GWL-1 cannot distinguish two connected hypergraphs $H_1$ and $H_2$ then HyperPageRank will not either.*

*Proof.* HyperPageRank is defined on a hypergraph with star expansion matrix $H$ as the following stationary distribution $\Pi$:

$$\lim_{n \to \infty} (D_v^{-1} \cdot H \cdot D_e^{-1} \cdot H^T)^n = \Pi \tag{27}$$

If $H$ is a connected bipartite graph, $\Pi$ must be the eigenvector of $(D_v^{-1} \cdot H \cdot D_e^{-1} \cdot H^T)$ for eigenvalue 1. In other words, $\Pi$ must satisfy

$$(D_v^{-1} \cdot H \cdot D_e^{-1} \cdot H^T) \cdot \Pi = \Pi \tag{28}$$

By Theorem 1 of Huang & Yang (2021), we know that the UniGCN defined by:

$$h_e^{i+1} \leftarrow \phi_2(h_e^i, h_v^i) = W_e \cdot H^T \cdot h_v^i \tag{29a}$$

$$h_v^{i+1} \leftarrow \phi_1(h_v^i, h_e^{i+1}) = W_v \cdot H \cdot h_e^{i+1} \tag{29b}$$

for constant $W_e$ and $W_v$ weight matrices, is equivalent to GWL-1 provided that $\phi_1$ and $\phi_2$ are both injective as functions. Without injectivity, we can only guarantee that if UniGCN distinguishes $H_1, H_2$ then GWL-1 distinguishes $H_1, H_2$. In fact, each matrix power of order $n$ in Equation 27 corresponds to $h_v^n$ so long as we satisfy the following constraints:

$$W_e \leftarrow D_e^{-1}, W_v \leftarrow D_v^{-1} \text{ and } h_v^0 \leftarrow I \tag{30}$$

We show that the matrix powers are UniGCN under the constraints of Equation 30 by induction:

Base Case: $n = 0$: $h_v^0 = I$

Induction Hypothesis: $n > 0$:

$$(D_v^{-1} \cdot H \cdot D_e^{-1} \cdot H^T)^n = h_v^n \tag{31}$$

Induction Step:

$$(D_v^{-1} \cdot H \cdot h_e^n) \tag{32a}$$

$$= (D_v^{-1} \cdot H \cdot ((D_e^{-1} \cdot H^T) \cdot h_v^n)) \tag{32b}$$

$$= (D_v^{-1} \cdot H \cdot D_e^{-1} \cdot H^T) \cdot (D_v^{-1} \cdot H \cdot D_e^{-1} \cdot H^T)^n \tag{32c}$$

$$= (D_v^{-1} \cdot H \cdot D_e^{-1} \cdot H^T)^{n+1} = h_v^{n+1} \tag{32d}$$

Since we cannot guarantee that the maps $\phi_1$ and $\phi_2$ are injective in Equation 32b, it must be that the output $h_v^n$, coming from UniGCN with the constraints of Equation 30, is at most as powerful as GWL-1.

In general, injectivity preserves more information. For example, if $\phi_1$ is injective and if $\phi_1'$ is an arbitrary map (not guaranteed to be injective) then:

$$\phi_1(h_1) = \phi_1(h_2) \Rightarrow h_1 = h_2 \Rightarrow \phi_1'(h_1) = \phi_1'(h_2) \tag{33}$$

HyperpageRank is exactly as powerful as UniGCN under the constraints of Equation 30. Thus HyperPageRank is at most as powerful as GWL-1 in distinguishing power. $\qquad\square$

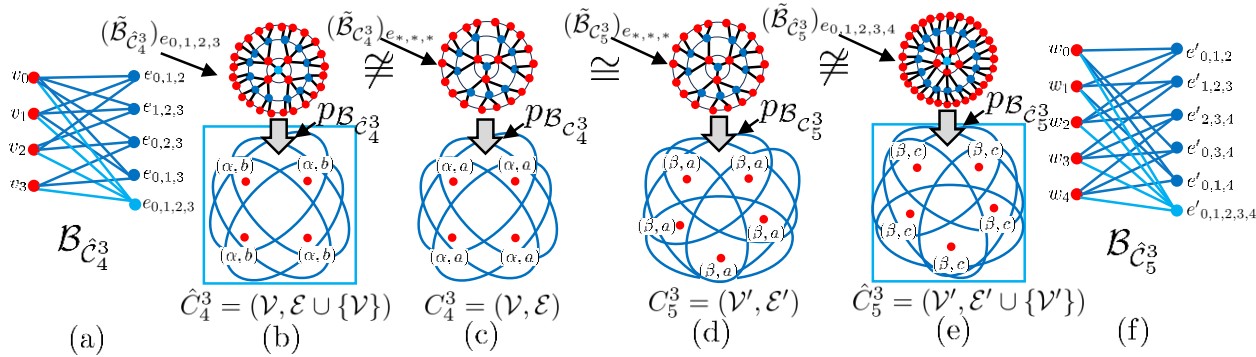

Figure 4: An illustration of hypergraph symmetry breaking. (c,d) 3-regular hypergraphs $C_4^3, C_5^3$ with 4 and 5 nodes respectively and their corresponding universal covers centered at any hyperedge $(\tilde{\mathcal{B}}_{C_4^3})_{e_{*,*,*}}, (\tilde{\mathcal{B}}_{C_5^3})_{e_{*,*,*}}$ with universal covering maps $p_{\mathcal{B}_{C_4^3}}, p_{\mathcal{B}_{C_5^3}}$. (b,e) the hypergraphs $\hat{C}_4^3, \hat{C}_5^3$, which are $C_4^3, C_5^3$ with $4, 5$-sized hyperedges attached to them and their corresponding universal covers and universal covering maps. (a,f) are the corresponding bipartite graphs of $\hat{C}_4^3, \hat{C}_5^3$. (c,d) are indistinguishable by GWL-1 and thus will give identical node values by Theorem B.8. On the other hand, (b,e) gives node values which are now sensitive to the the order of the hypergraphs $4, 5$, also by Theorem B.8.

## B.3    Method

We repeat here from the main text the symmetry finding algorithm:

---

**Algorithm 2:** A Symmetry Finding Algorithm

---

    **Data:** Hypergraph $\mathcal{H} = (\mathcal{V}, \mathcal{E})$, represented by its star expansion matrix $H$. $L \in \mathbb{N}^+$ is the number of iterations to run GWL-1.

    **Result:** A pair of collections: $(\mathcal{R}_V = \{\mathcal{V}_{R_j}\}, \mathcal{R}_E = \cup_j \{\mathcal{E}_{R_j}\})$ where $R_j$ are disconnected subhypergraphs exhibiting symmetry in $\mathcal{H}$ that are indistinguishable by $L$-GWL-1.

**1** $E_{deg} \leftarrow \{\{deg(v) : v \in e\} : \forall e \in \mathcal{E}\}$

**2** $U_L \leftarrow h_v^L(H); \mathcal{G}_L \leftarrow \{U_L[v] : \forall v \in \mathcal{V}\}$ ;             /* $U_L[v]$ is the $L$-GWL-1 value of node $v \in \mathcal{V}$. */

**3** $\mathcal{B}_{\mathcal{V}_\mathcal{H}, \mathcal{E}_\mathcal{H}} \leftarrow Bipartite(\mathcal{H})$ /* Construct the bipartite graph from $\mathcal{H}$. */

**4** $\mathcal{R}_V \leftarrow \{\}; \mathcal{R}_E \leftarrow \{\}$

**5** **for** $c_L \in \mathcal{G}_L$ **do**

**6**      $\mathcal{V}_{c_L} \leftarrow \{v \in \mathcal{V} : U_L[v] = c_L\}, \mathcal{E}_{c_L} \leftarrow \{e \in \mathcal{E} : u \in \mathcal{V}_{c_L}, \forall u \in e\}$

**7**      $\mathcal{C}_{c_L} \leftarrow \text{ConnectedComponents}(\mathcal{H}_{c_L} = (\mathcal{V}_{c_L}, \mathcal{E}_{c_L}))$

**8**      **for** $\mathcal{R}_{c_L,i} \in \mathcal{C}_{c_L}$ **do**

**9**          **if** $|\mathcal{V}_{\mathcal{R}_{c_L,i}}| \geq 3$ *and* $\{deg(v) : v \in (\mathcal{V}_{\mathcal{R}_{c_L,i}})\} \notin E_{deg}$ **then**

**10**             $\mathcal{R}_V \leftarrow \mathcal{R}_V \cup \{\mathcal{V}_{\mathcal{R}_{c_L,i}}\}; \mathcal{R}_E \leftarrow \mathcal{R}_E \cup \mathcal{E}_{\mathcal{R}_{c_L,i}}$

**11**          **end**

**12**      **end**

**13** **end**

**14** **return** $(\mathcal{R}_V, \mathcal{R}_E)$

---

We also repeat here for convenience some definitions used in the proofs. Given a hypergraph $\mathcal{H} = (\mathcal{V}, \mathcal{E})$, let

$$\mathcal{V}_{c_L} := \{v \in \mathcal{V} : c_L = h_v^L(H)\} \tag{34}$$

be the set of nodes of the same class $c_L$ as determined by $L$-GWL-1. Let $\mathcal{H}_{c_L}$ be an induced subgraph of $\mathcal{H}$ by $\mathcal{V}_{c_L}$.

**Definition B.9.** *A $L$-GWL-1 symmetric induced subhypergraph $\mathcal{R} \subset \mathcal{H}$ of $\mathcal{H}$ is a connected induced subhypergraph determined by $\mathcal{V}_\mathcal{R} \subset \mathcal{V}_\mathcal{H}$, some subset of nodes that are all indistinguishable amongst each other by*

*L-GWL-1:*

$$h_u^L(H) = h_v^L(H), \forall u, v \in \mathcal{V}_{\mathcal{R}} \tag{35}$$

*When $L = \infty$, we call such $\mathcal{R}$ a GWL-1 symmetric induced subhypergraph. Furthermore, if $\mathcal{R} = \mathcal{H}$, then we say $\mathcal{H}$ is GWL-1 symmetric.*

**Definition B.10.** *A neighborhood-regular hypergraph is a hypergraph where all neighborhoods of each node are isomorphic to each other.*

**Observation 4.** *A hypergraph $\mathcal{H}$ is GWL-1 symmetric if and only if it is L-GWL-1 symmetric for all $L \geq 1$ if and only if $\mathcal{H}$ is neighborhood regular.*

*Proof.*
**1. First if and only if** :

By Theorem B.8, GWL-1 symmetric hypergraph $\mathcal{H} = (\mathcal{V}, \mathcal{E})$ means that for every pair of nodes $u, v \in \mathcal{V}$, $(\tilde{\mathcal{B}}_{\mathcal{V}, \mathcal{E}})_{\tilde{u}} \cong_c (\tilde{\mathcal{B}}_{\mathcal{V}, \mathcal{E}})_{\tilde{v}}$. This implies that for any $L \geq 1$, $(\tilde{\mathcal{B}}_{\mathcal{V}, \mathcal{E}}^{2L})_{\tilde{u}} \cong_c (\tilde{\mathcal{B}}_{\mathcal{V}, \mathcal{E}}^{2L})_{\tilde{v}}$ by restricting the rooted isomorphism to $2L$-hop rooted subtrees, which means that $h_u^L(H) = h_v^L(H)$. The converse is true since $L$ is arbitrary. If there are no cycles, we can just take the isomorphism for the largest. Otherwise, an isomorphism can be constructed for $L = \infty$ by infinite extension.

**2. Second if and only if** :

Let $p_{\mathcal{B}_{\mathcal{V}, \mathcal{E}}}$ be the universal covering map for $\mathcal{B}_{\mathcal{V}, \mathcal{E}}$. Denote $\tilde{v}, \tilde{u}$ by the lift of some nodes $v, u \in \mathcal{V}$ by $p_{\mathcal{B}_{\mathcal{V}, \mathcal{E}}}$.

Let $(\tilde{N}(\tilde{u}))_{\tilde{u}}$ be the rooted bipartite lift of $(N(u))_u$. If $\mathcal{H}$ is $L$-GWL-1 symmetric for all $L \geq 1$ then with $L = 1$, $(\tilde{\mathcal{B}}_{\mathcal{V}, \mathcal{E}}^2)_{\tilde{u}} \cong_c (\tilde{N}(u))_{\tilde{u}} \cong_c (\tilde{N}(v))_{\tilde{v}} \cong_c (\tilde{\mathcal{B}}_{\mathcal{V}, \mathcal{E}}^2)_{\tilde{v}}$, iff $(N(u))_u \cong (N(v))_{\tilde{v}}, \forall u, v \in \mathcal{V}$ since $N(u)$ and $N(v)$ are cycle-less for any $u, v \in \mathcal{V}$. For the converse, assume all nodes $v \in \mathcal{V}$ have $(N(v))_v \cong (N^1)_x$ for some 1-hop rooted tree $(N^1)_x$ rooted at node $x$, independent of any $v \in \mathcal{V}$. We prove by induction that for all $L \geq 1$ and for all $v \in \mathcal{V}$, $(\tilde{\mathcal{B}}_{\mathcal{V}, \mathcal{E}}^{2L})_{\tilde{v}} \cong_c (\tilde{N}^{2L})_x$ for a $2L$-hop tree $(\tilde{N}^{2L})_x$ rooted at node $x$.

Base case: $L = 1$ is by assumption.

Inductive step: If $(\tilde{\mathcal{B}}_{\mathcal{V}, \mathcal{E}}^{2L})_{\tilde{v}} \cong_c (N^{2L})_x$, we can form $(\tilde{\mathcal{B}}_{\mathcal{V}, \mathcal{E}}^{2L+2})_{\tilde{v}}$ by attaching $(\tilde{N}(\tilde{u}))_{\tilde{u}}$ to each node $\tilde{u}$ in the $2L$-th layer of $(\tilde{\mathcal{B}}_{\mathcal{V}, \mathcal{E}}^{2L})_{\tilde{v}} \cong_c (\tilde{N}^{2L})_x$. Each $(\tilde{N}(u))_{\tilde{u}}$ is independent of the root $\tilde{v}$ since every $u \in \mathcal{V}$ has $(\tilde{N}(\tilde{u}))_{\tilde{u}} \cong_c (\tilde{N}^2)_x$ iff $(N(\tilde{u}))_{\tilde{u}} \cong (N^1)_x$ for an $x$ independent of $u \in \mathcal{V}$. This means $(\tilde{\mathcal{B}}_{\mathcal{V}, \mathcal{E}}^{2L+2})_{\tilde{v}} \cong_c (\tilde{N}^{2L+2})_x$ for the same root node $x$ where $(\tilde{N}^{2L+2})_x$ is constructed in the same manner as $(\tilde{\mathcal{B}}_{\mathcal{V}, \mathcal{E}}^{2L})_{\tilde{v}}, \forall v \in \mathcal{V}$.

$\square$

### B.3.1 Algorithm Guarantees

Continuing with the notation, as before, let $\mathcal{H} = (\mathcal{V}, \mathcal{E})$ be a hypergraph with star expansion matrix $H$ and let $(\mathcal{R}_{\mathcal{V}}, \mathcal{R}_{\mathcal{E}})$ be the output of Algorithm 1 on $H$ for $L \in \mathbb{N}^+$. Denote $\mathcal{C}_{c_L}$ as the set of all connected components of $\mathcal{H}_{c_L}$:

$$\mathcal{C}_{c_L} \triangleq \{C_{c_L} : \text{conn. comp. } C_{c_L} \text{ of } \mathcal{H}_{c_L}\} \tag{36}$$

If $L = \infty$, then drop the $L$. Thus, the hypergraphs represented by $(\mathcal{R}_V, \mathcal{R}_E)$ come from $\mathcal{C}_{c_L}$ for each $c_L$. Let:

$$\hat{\mathcal{H}}_L \triangleq (\mathcal{V}, \mathcal{E} \cup \mathcal{R}_V) \tag{37}$$

be $\mathcal{H}$ after adding all the hyperedges from $\mathcal{R}_{\mathcal{V}}$ and let $\hat{H}_L$ be the star expansion matrix of the resulting hypergraph $\hat{\mathcal{H}}_L$. Let:

$$\mathcal{G}_L \triangleq \{h_v^L(H) : v \in \mathcal{V}\} \tag{38}$$

be the set of all $L$-GWL-1 values on $H$. Let:

$$V_{c_L, s} \triangleq \{v \in \mathcal{V}_{c_L} : v \in R, R \in \mathcal{C}_{c_L}, |\mathcal{V}_R| = s\} \tag{39}$$

be the set of all nodes of $L$-GWL-1 class $c_L$ belonging to a connected component in $\mathcal{C}_{c_L}$ of $s \geq 1$ nodes in $\mathcal{H}_{c_L}$, the induced subhypergraph of $L$-GWL-1. Let:

$$\mathcal{S}_{c_L} \triangleq \{|\mathcal{V}_{\mathcal{R}_{c_L, i}}| : \mathcal{R}_{c_L, i} \in \mathcal{C}_{c_L}\} \tag{40}$$

be the set of node set sizes of the connected components in $\mathcal{H}_{c_L}$.

**Proposition B.10.** *If $L = \infty$, for any GWL-1 node value $c$ for $\mathcal{H}$, the connected component induced subhypergraphs $\mathcal{R}_{c,i}$, for $i = 1...|\mathcal{C}_c|$ are GWL-1 symmetric and neighborhood-regular.*

*Proof.* Let $p_{\mathcal{B}_{\mathcal{V},\mathcal{E}}}$ be the universal covering map for $\mathcal{B}_{\mathcal{V},\mathcal{E}}$. Denote $\tilde{v}, \tilde{u}, \tilde{v}', \tilde{u}'$ by the lift of some nodes $v, u, v', u' \in \mathcal{V}$ by $p_{\mathcal{B}_{\mathcal{V},\mathcal{E}}}$.

Let $L = \infty$ and let $\mathcal{H}_c = (\mathcal{V}_c, \mathcal{E}_c)$. For any $i$, since $u, v \in \mathcal{V}_c$, $(\tilde{\mathcal{B}}_{\mathcal{V},\mathcal{E}})_u \cong_c (\tilde{\mathcal{B}}_{\mathcal{V},\mathcal{E}})_v$ for all $u, v \in \mathcal{V}_{\mathcal{R}_{c,i}}$. Since $\mathcal{R}_{c,i}$ is maximally connected we know that every neighborhood $N_{\mathcal{H}_c}(u)$ for $u \in \mathcal{V}_c$ induced by $\mathcal{H}_c$ has $N_{\mathcal{H}_c}(u) \cong N(u) \cap \mathcal{H}_c$. Since $L = \infty$ we have that $N_{\mathcal{H}_c}(u) \cong N_{\mathcal{H}_c}(v), \forall u, v \in \mathcal{V}_{R_{c,i}}$ since otherwise WLOG there are $u', v' \in \mathcal{V}_{R_{c,i}}$ with $N_{\mathcal{H}_c}(u') \not\cong N_{\mathcal{H}_c}(v')$ then WLOG there is some hyperedge $e \in \mathcal{E}_{N_{\mathcal{H}_c}(u')}$ with some $w \in e$, $w \neq u'$ where $e$ cannot be in isomorphism with any $e' \in \mathcal{E}_{N_{\mathcal{H}_c}(v')}$. For two hyperedges to be in isomorphism means that their constituent nodes can be bijectively mapped to each other by a restriction of an isomorphism $\phi$ between $N_{\mathcal{H}_c}(u'), N_{\mathcal{H}_c}(v')$ to one of the hyperedges. This means that $(\tilde{\mathcal{B}}_{\mathcal{V}\setminus\{u'\},\mathcal{E}})_w$ is the rooted universal covering subtree centered about $w$ not passing through $u'$ that is connected to $u' \in (\tilde{\mathcal{B}}_{\mathcal{V},\mathcal{E}})_{u'}$ by $e$. However, $v'$ has no $e$ and thus cannot have a $T_x$ for $x \in \mathcal{V}_{(\tilde{N}(v'))_v}$ satisfying $T_x \cong_c (\tilde{\mathcal{B}}_{\mathcal{V}\setminus\{u'\},\mathcal{E}})_w$ with $x$ connected to $v'$ by a hyperedge $e'$ isomorphic to $e$ in its neighborhood in $(\tilde{\mathcal{B}}_{\mathcal{V},\mathcal{E}})_{v'}$. This contradicts that $(\tilde{\mathcal{B}}_{\mathcal{V},\mathcal{E}})_{u'} \cong_c (\tilde{\mathcal{B}}_{\mathcal{V},\mathcal{E}})_{v'}$.

We have thus shown that all nodes in $\mathcal{V}_c$ have isomorphic induced neighborhoods. By the Observation 4, this is equivalent to saying that $\mathcal{R}_{c,i}$ is GWL-1 symmetric and neighborhood regular. $\square$

**Definition B.11.** *A star graph $N_x$ is defined as a tree rooted at $x$ of depth $1$. The root $x$ is the only node that can have degree more than $1$.*

**Lemma B.11.** *If $L \in \mathbb{N}^+$ is small enough so that after running Algorithm 1 on $L$, for any $L$-GWL-1 node class $c_L$ on $\mathcal{V}$ none of the discovered $\mathcal{V}_{\mathcal{R}_{c_L,i}}$ are within $L$ hyperedges away from any $\mathcal{V}_{\mathcal{R}_{c_L,j}}$ for all $i, j \in 1...|\mathcal{C}_{c_L}|, i \neq j$, then after forming $\hat{\mathcal{H}}_L$, the new $L$-GWL-1 node classes of $\mathcal{V}_{\mathcal{R}_{c_L,i}}$ for $i = 1...|\mathcal{C}_{c_L}|$ in $\hat{\mathcal{H}}_L$ are all the same class $c'_L$ but are distinguishable from $c_L$ depending on $|\mathcal{V}_{\mathcal{R}_{c_L,i}}|$.*

*Proof.* After running Algorithm 1 on $\mathcal{H} = (\mathcal{V}, \mathcal{E})$, let $\hat{\mathcal{H}}_L = (\hat{\mathcal{V}}_L, \hat{\mathcal{E}}_L \triangleq \mathcal{E} \cup \bigcup_{c_L,i}\{\mathcal{V}_{\mathcal{R}_{c_L,i}}\})$ be the hypergraph formed by attaching a hyperedge to each $\mathcal{V}_{\mathcal{R}_{c_L,i}}$.

For any $c_L$, a $L$-GWL-1 node class, let $\mathcal{R}_{c_L,i}, i = 1...|\mathcal{C}_{c_L}|$ be a connected component subhypergraph of $\mathcal{H}_{c_L}$. Over all $(c_L, i)$ pairs, all the $\mathcal{R}_{c_L,i}$ are disconnected from each other and for each $c_L$ each $\mathcal{R}_{c_L,i}$ is maximally connected on $\mathcal{H}_{c_L}$.

Upon covering all the nodes $\mathcal{V}_{\mathcal{R}_{c_L,i}}$ of each induced connected component subhypergraph $\mathcal{R}_{c_L,i}$ with a single hyperedge $e = \mathcal{V}_{\mathcal{R}_{c_L,i}}$ of size $s = |\mathcal{V}_{\mathcal{R}_{c_L,i}}|$, we claim that every node of class $c_L$ becomes $c_{L,s}$, a $L$-GWL-1 node class depending on the original $L$-GWL-1 node class $c_L$ and the size of the hyperedge $s$.

Consider for each $v \in \mathcal{V}_{\mathcal{R}_{c_L,i}}$ the $2L$-hop rooted tree $(\tilde{\mathcal{B}}^{2L}_{\mathcal{V},\mathcal{E}})_{\tilde{v}}$ for $p_{\mathcal{B}_{\mathcal{V},\mathcal{E}}}(\tilde{v}) = v$. Also, for each $v \in \mathcal{V}_{\mathcal{R}_{c_L,i}}$, define the tree

$$T_e \triangleq (\tilde{\mathcal{B}}^{2L-1}_{\hat{\mathcal{V}}\setminus\{v\},\mathcal{E}})_{\tilde{e}} \tag{41}$$

We do not index the tree $T_e$ by $v$ since it does not depend on $v \in \mathcal{V}_{\mathcal{R}_{c_L,i}}$. We prove this in the following.

**proof for: $T_e$ does not depend on $v \in \mathcal{V}_{\mathcal{R}_{c_L,i}}$:**

Let node $\tilde{e}$ be the lift of $e$ to $(\tilde{\mathcal{B}}^{2L-1}_{\hat{\mathcal{V}},\hat{\mathcal{E}}})_{\tilde{e}}$. Define the star graph $(N(\tilde{e}))_{\tilde{e}}$ as the 1-hop neighborhood of $\tilde{e}$ in $(\tilde{\mathcal{B}}^{2L-1}_{\hat{\mathcal{V}},\hat{\mathcal{E}}})_{\tilde{e}}$. We must have:

$$(\tilde{\mathcal{B}}^{2L-1}_{\hat{\mathcal{V}},\hat{\mathcal{E}}})_{\tilde{e}} \cong_c ((N(\tilde{e}))_{\tilde{e}} \cup \bigcup_{\tilde{u} \in \mathcal{V}_{N(\tilde{e})}\setminus\{\tilde{e}\}} (\tilde{\mathcal{B}}^{2L-2}_{\mathcal{V},\mathcal{E}})_{\tilde{u}})_{\tilde{e}} \tag{42}$$

Define for each node $v \in e$ with lift $\tilde{v}$:

$$(N(\tilde{e}, \tilde{v}))_{\tilde{e}} \triangleq (\mathcal{V}_{(N(\tilde{e}))_{\tilde{e}}} \setminus \{\tilde{v}\}, \mathcal{E}_{(N(\tilde{e}))_{\tilde{e}}} \setminus \{(\tilde{e}, \tilde{v})\})_{\tilde{e}} \tag{43}$$

The tree $(N(\tilde{e}, \tilde{v}))_{\tilde{e}}$ is a star graph with the node $\tilde{v}$ deleted from $(N(\tilde{e}))_{\tilde{e}}$. The star graphs $(N(\tilde{e}, \tilde{v}))_{\tilde{e}} \subset (N(\tilde{e}))_{\tilde{e}}$ do not depend on $\tilde{v}$ as long as $\tilde{v} \in \mathcal{V}_{(N(\tilde{e}))_{\tilde{e}}}$. In other words,

$$(N(\tilde{e}, \tilde{v}))_{\tilde{e}} \cong_c (N(\tilde{e}, \tilde{v}'))_{\tilde{e}}, \forall \tilde{v}, \tilde{v}' \in \mathcal{V}_{(N(\tilde{e}))_{\tilde{e}}} \setminus \{\tilde{e}\} \tag{44}$$

Since the rooted tree $(\tilde{\mathcal{B}}^{2L-1}_{\hat{\mathcal{V}}, \hat{\mathcal{E}}})_{\tilde{e}}$, where $\tilde{e}$ is the lift of $e$ by universal covering map $p_{\mathcal{B}_{\mathcal{V}, \mathcal{E}}}$, has all pairs of nodes $\tilde{u}, \tilde{u}' \in \tilde{e}$ in it with $(\tilde{\mathcal{B}}^{2L}_{\mathcal{V}, \mathcal{E}})_{\tilde{u}} \cong_c (\tilde{\mathcal{B}}^{2L}_{\mathcal{V}, \mathcal{E}})_{\tilde{u}'}$, which implies

$$(\tilde{\mathcal{B}}^{2L-2}_{\mathcal{V}, \mathcal{E}})_{\tilde{u}} \cong_c (\tilde{\mathcal{B}}^{2L-2}_{\mathcal{V}, \mathcal{E}})_{\tilde{u}'}, \forall \tilde{u}, \tilde{u}' \in \tilde{e} \tag{45}$$

By Equations 45, 44, we thus have:

$$(\tilde{\mathcal{B}}^{2L-1}_{\hat{\mathcal{V}} \setminus \{v\}, \hat{\mathcal{E}}})_{\tilde{e}} \cong_c ((N(\tilde{e}, \tilde{v}))_{\tilde{e}} \cup \bigcup_{\tilde{u} \in \mathcal{V}_{(N(\tilde{e}, \tilde{v}))_{\tilde{e}}} \setminus \{\tilde{e}\}} (\tilde{\mathcal{B}}^{2L-2}_{\mathcal{V}, \mathcal{E}})_{\tilde{u}})_{\tilde{e}} \tag{46}$$

This proves that $T_e$ does not need to be indexed by $v \in \mathcal{V}_{\mathcal{R}_{c_L, i}}$.

We continue with the proof that all nodes in $\mathcal{V}_{\mathcal{R}_{c_L, i}}$ become the $L$-GWL-1 node class $c_{L,s}$ for $s = |\mathcal{V}_{\mathcal{R}_{c_L, i}}|$.

Since every $v \in \mathcal{V}_{\mathcal{R}_{c_L, i}}$ becomes connected to a hyperedge $e = \mathcal{V}_{\mathcal{R}_{c_L, i}}$ in $\hat{\mathcal{H}}$, we must have:

$$(\tilde{\mathcal{B}}^{2L}_{\hat{\mathcal{V}}, \hat{\mathcal{E}}})_{\tilde{v}} \cong_c ((\tilde{\mathcal{B}}^{2L}_{\mathcal{V}, \mathcal{E}})_{\tilde{v}} \cup_{(\tilde{v}, \tilde{e})} T_e)_{\tilde{v}}, \forall v \in \mathcal{V}_{\mathcal{R}_{c_L, i}} \tag{47}$$

The notation $((\tilde{\mathcal{B}}^{2L}_{\mathcal{V}, \mathcal{E}})_{\tilde{v}} \cup_{(\tilde{v}, \tilde{e})} T_e)_{\tilde{v}}$ denotes a tree rooted at $\tilde{v}$ that is the attachment of the tree $T_e$ rooted at $e$ to the node $\tilde{v}$ by the edge $(\tilde{v}, \tilde{e})$. As is usual, we assume $\tilde{v}, \tilde{e}$ are the lifts of $v \in \mathcal{V}, e \in \mathcal{E}$ respectively. We only need to consider the single $e$ since $L$ was chosen small enough so that the $2L$-hop tree $(\tilde{\mathcal{B}}^{2L}_{\hat{\mathcal{V}}, \hat{\mathcal{E}}})_{\tilde{v}}$ does not contain a node $\tilde{u}$ satisfying $p_{\mathcal{B}_{\mathcal{V}, \mathcal{E}}}(\tilde{u}) = u$ with $u \in \mathcal{V}_{\mathcal{R}_{c_L, j}}$ for all $j = 1...|\mathcal{C}_{c_L}|, j \neq i$.

Since $T_e$ does not depend on $v \in \mathcal{V}_{\mathcal{R}_{c_L, i}}$,

$$(\tilde{\mathcal{B}}^{2L}_{\hat{\mathcal{V}}, \hat{\mathcal{E}}})_{\tilde{u}} \cong_c (\tilde{\mathcal{B}}^{2L}_{\hat{\mathcal{V}}, \hat{\mathcal{E}}})_{\tilde{v}}, \forall u, v \in \mathcal{V}_{\mathcal{R}_{c_L, i}} \tag{48}$$

This shows that $h^L_u(\hat{H}) = h^L_v(\hat{H}), \forall u, v \in \mathcal{V}_{\mathcal{R}_{c_L, i}}$ by Theorem B.8. Furthermore, since each $v \in \mathcal{V}_{\mathcal{R}_{c_L, i}} \subset \hat{\mathcal{V}}$ in $\hat{\mathcal{H}}$ is now incident to a new hyperedge $e = \mathcal{V}_{\mathcal{R}_{c_L, i}}$, we must have that the $L$-GWL-1 class $c_L$ of $\mathcal{V}_{\mathcal{R}_{c_L, i}}$ on $\mathcal{H}$ is now distinguishable by $|\mathcal{V}_{\mathcal{R}_{c_L, i}}|$.

$\square$

We will need the following definition to prove the next lemma.

**Definition B.12.** *A partial universal cover of hypergraph $\mathcal{H} = (\mathcal{V}, \mathcal{E})$ with an unexpanded induced subhypergraph $\mathcal{R}$, denoted $U(\mathcal{H}, \mathcal{R})_{\mathcal{V}, \mathcal{E}}$ is a graph cover of $\mathcal{B}_{\mathcal{V}, \mathcal{E}}$ where we freeze $\mathcal{B}_{\mathcal{V}_{\mathcal{R}}, \mathcal{E}_{\mathcal{R}}} \subset \tilde{\mathcal{B}}_{\mathcal{V}, \mathcal{E}}$ as an induced subgraph.*

*A $l$-hop rooted partial universal cover of hypergraph $\mathcal{H} = (\mathcal{V}, \mathcal{E})$ with an unexpanded induced subhypergraph $\mathcal{R}$, denoted $(U^l(\mathcal{H}, \mathcal{R})_{\mathcal{V}, \mathcal{E}})_{\tilde{u}}$ for $u \in \mathcal{V}$ or $(U^l(\mathcal{H}, \mathcal{R})_{\mathcal{V}, \mathcal{E}})_{\tilde{e}}$ for $e \in \mathcal{E}$, where $\tilde{v}, \tilde{e}$ are lifts of $v, e$, is a rooted graph cover of $\mathcal{B}_{\mathcal{V}, \mathcal{E}}$ where we freeze $\mathcal{B}_{\mathcal{V}_{\mathcal{R}}, \mathcal{E}_{\mathcal{R}}} \subset \tilde{\mathcal{B}}_{\mathcal{V}, \mathcal{E}}$ as an induced subgraph.*

**Lemma B.12.** *Assuming the same conditions as Lemma B.11, where $\mathcal{H} = (\mathcal{V}, \mathcal{E})$ is a hypergraph and for all $L$-GWL-1 node classes $c_L$ with connected components $\mathcal{R}_{c_L, i}$, as discovered by Algorithm 1, so that $L \geq diam(\mathcal{R}_{c_L, i})$. Instead of only adding the hyperedges $\{\mathcal{V}_{\mathcal{R}_{c_L, i}}\}_{c_L, i}$ to $\mathcal{E}$ as stated in the main paper, let $\hat{\mathcal{H}}_\dagger \triangleq (\mathcal{V}, (\mathcal{E} \setminus \mathcal{R}_E) \cup \mathcal{R}_V)$, meaning $\mathcal{H}$ with each $\mathcal{R}_{c_L, i}$ for $i = 1...|\mathcal{C}_{c_L}|$ having all of its hyperedges dropped and with a single hyperedge that covers $\mathcal{V}_{\mathcal{R}_{c_L, i}}$ and let $\hat{\mathcal{H}} = (\mathcal{V}, \mathcal{E} \cup \mathcal{R}_V)$ then:*

*The GWL-1 node classes of $\mathcal{V}_{\mathcal{R}_{c_L, i}}$ for $i = 1...|\mathcal{C}_{c_L}|$ in $\hat{\mathcal{H}}$ are all the same class $c'_L$ but are distinguishable from $c_L$ depending on $|\mathcal{V}_{\mathcal{R}_{c_L, i}}|$.*

*Proof.* For any $c_L$, a $L$-GWL-1 node class, let $\mathcal{R}_{c_L,i}, i = 1...|\mathcal{C}_{c_L}|$ be a connected component subhypergraph of $\mathcal{H}_{c_L}$. These connected components are discovered by the algorithm. Over all $(c_L, i)$ pairs, all the $\mathcal{R}_{c_L,i}$ are disconnected from each other. Upon arbitrarily deleting all hyperedges in each such induced connected component subhypergraph $\mathcal{R}_{c_L,i}$ and adding a single hyperedge of size $s = |\mathcal{V}_{\mathcal{R}_{c_L,i}}|$, we claim that every node of class $c_L$ becomes $c_{L,s}$, a $L$-GWL-1 node class depending on the original $L$-GWL-1 node class $c_L$ and the size of the hyperedge $s$.

Define the subhypergraph made up of the disconnected components $\mathcal{R}_{c_L,i}$ as:

$$\mathcal{R} := \bigcup_{c,i} \mathcal{R}_{c_L,i} \tag{49}$$

Since $L \geq diam(\mathcal{R}_{c_L,i})$, we can construct the $2L$-hop rooted partial universal cover with unexpanded induced subhypergraph $\mathcal{R}$, denoted by $(U^{2L}(\mathcal{H},\mathcal{R})_{\mathcal{V},\mathcal{E}})_{\tilde{v}}, \forall v \in \mathcal{V}$ of $\mathcal{H}$ as given in Definition B.12.

Denote the hyperedge nodes, or right hand nodes of the bipartite graph by $\mathcal{B}(\mathcal{V}_{\mathcal{R}}, \mathcal{E}_{\mathcal{R}})$ by $R(\mathcal{B}(\mathcal{V}_{\mathcal{R}}, \mathcal{E}_{\mathcal{R}}))$. Their corresponding hyperedges are $\mathcal{E}_{\mathcal{R}} \subset \mathcal{E}(U(\mathcal{H},\mathcal{R})) \subset \mathcal{E}$. Since each $\mathcal{R}_{c_L,i}$ is maximally connected, for any nodes $u, v \in \mathcal{V}_{\mathcal{R}}$ we have:

$$(U^{2L}(\mathcal{H},\mathcal{R})_{\tilde{u}} \setminus R(\mathcal{B}(\mathcal{V}_{\mathcal{R}}, \mathcal{E}_{\mathcal{R}})))_{\tilde{u}} \cong_c (U^{2L}(\mathcal{H},\mathcal{R})_{\tilde{v}} \setminus R(\mathcal{B}(\mathcal{V}_{\mathcal{R}}, \mathcal{E}_{\mathcal{R}})))_{\tilde{v}} \tag{50}$$

by Proposition B.10, where $U^{2L}(\mathcal{H},\mathcal{R})_{\tilde{v}} \setminus R(\mathcal{B}(\mathcal{V}_{\mathcal{R}}, \mathcal{E}_{\mathcal{R}}))$ denotes removing the nodes $R(\mathcal{B}(\mathcal{V}_{\mathcal{R}}, \mathcal{E}_{\mathcal{R}}))$ from $U^{2L}(\mathcal{H},\mathcal{R})_{\tilde{v}}$. This follows since removing $R(\mathcal{B}(\mathcal{V}_{\mathcal{R}}, \mathcal{E}_{\mathcal{R}}))$ removes an isomorphic neighborhood of hyperedges from each node in $\mathcal{V}_{\mathcal{R}}$. This requires assuming maximal connectedness of each $\mathcal{R}_{c_L,i}$. Upon adding the hyperedge

$$e_{c_L,i} \triangleq \mathcal{V}_{\mathcal{R}_{c_L,i}} \tag{51}$$

covering all of $\mathcal{V}_{\mathcal{R}_{c_L,i}}$ after the deletion of $\mathcal{E}_{\mathcal{R}_{c_L,i}}$ for every $(c_L,i)$ pair, we see that any node $u \in \mathcal{V}_{\mathcal{R}_{c_L,i}}$ is connected to any other node $v \in \mathcal{V}_{\mathcal{R}_{c_L,i}}$ through $e_{c_L,i}$ in the same way for all nodes $u, v \in \mathcal{V}_{\mathcal{R}_{c_L,i}}$. In fact, we claim that all the nodes in $\mathcal{V}_{\mathcal{R}_{c_L,i}}$ still have the same GWL-1 class.

We can write $\hat{\mathcal{H}}_{\dagger}$ equivalently as $(\mathcal{V}, \bigcup_{c_L,i}(\mathcal{E} \setminus \mathcal{E}(\mathcal{R}_{c_L,i}) \cup \{e_{c_L,i}\}))$, which is the hypergraph formed by the algorithm. The replacement operation on $\mathcal{H}$ can be viewed in the universal covering space $\tilde{\mathcal{B}}_{\mathcal{V},\mathcal{E}}$ as taking $U(\mathcal{H},\mathcal{R})$ and replacing the frozen subgraph $\mathcal{B}_{\mathcal{V}_{\mathcal{R}},\mathcal{E}_{\mathcal{R}}}$ with the star graphs $(N_{\hat{\mathcal{H}}_{\dagger}}(\tilde{e}_{c_L,i}))_{\tilde{e}_{c_L,i}}$ of root node $\tilde{e}_{c_L,i}$ determined by hyperedge $e_{c_L,i}$ for each connected component indexed by $(c_L,i)$. Since the star graphs $(N_{\hat{\mathcal{H}}_{\dagger}}(\tilde{e}_{c_L,i}))_{\tilde{e}_{c_L,i}}$ are cycle-less, we have that:

$$(U(\mathcal{H},\mathcal{R}) \setminus R(\mathcal{B}(\mathcal{V}_{\mathcal{R}}, \mathcal{E}_{\mathcal{R}}))) \cup \bigcup_{c_L,i} (N_{\hat{\mathcal{H}}_{\dagger}}(\tilde{e}_{c_L,i}))_{\tilde{e}_{c_L,i}} \cong_c \tilde{\mathcal{B}}_{\mathcal{V}_{\hat{\mathcal{H}}_{\dagger}}, \mathcal{E}_{\hat{\mathcal{H}}_{\dagger}}} \tag{52}$$

Viewing Equation 52 locally, by our assumptions on $L$, for any $v \in \mathcal{V}_{\mathcal{R}_{c_L,i}}$, we must also have:

$$(U^{2L}(\mathcal{H},\mathcal{R})_{\tilde{v}} \setminus R(\mathcal{B}(\mathcal{V}_{\mathcal{R}}, \mathcal{E}_{\mathcal{R}}))) \bigcup (N_{\hat{\mathcal{H}}_{\dagger}}(\tilde{e}_{c,i}))_{\tilde{e}_{c,i}} \cong_c \tilde{\mathcal{B}}_{\mathcal{V}_{\hat{\mathcal{H}}_{\dagger}}, \mathcal{E}_{\hat{\mathcal{H}}_{\dagger}}} \tag{53}$$

We thus have $(\tilde{\mathcal{B}}^{2L}_{\mathcal{V}_{\hat{\mathcal{H}}_{\dagger}}, \mathcal{E}_{\hat{\mathcal{H}}_{\dagger}}})_{\tilde{u}} \cong_c (\tilde{\mathcal{B}}^{2L}_{\mathcal{V}_{\hat{\mathcal{H}}_{\dagger}}, \mathcal{E}_{\hat{\mathcal{H}}_{\dagger}}})_{\tilde{v}}$ for every $u, v \in \mathcal{V}_{\mathcal{R}_{c_L,i}}$ with $\tilde{u}, \tilde{v}$ being the lifts of $u, v$ by $p_{\mathcal{B}_{\mathcal{V},\mathcal{E}}}$, since $(U^{2L}(\mathcal{H},\mathcal{R})_{\tilde{u}} \setminus R(\mathcal{B}(\mathcal{V}_{\mathcal{R}}, \mathcal{E}_{\mathcal{R}})))_{\tilde{u}} \cong_c (U^{2L}(\mathcal{H},\mathcal{R})_{\tilde{v}} \setminus R(\mathcal{B}(\mathcal{V}_{\mathcal{R}}, \mathcal{E}_{\mathcal{R}})))_{\tilde{v}}$ for every $u, v \in \mathcal{V}_{\mathcal{R}_{c_L,i}}$ as in Equation 50. These rooted universal covers now depend on a new hyperedge $e_{c_L,i}$ and thus depend on its size $s$.

This proves the claim that all the nodes in $\mathcal{V}_{\mathcal{R}_{c_L,i}}$ retain the same $L$-GWL-1 node class by changing $\mathcal{H}$ to $\hat{\mathcal{H}}_{\dagger}$ and that this new class is distinguishable by $s = |\mathcal{V}_{\mathcal{R}_{c_L,i}}|$. In otherwords, the new class can be determined by $c_s$. Furthermore, $c_{L,s}$ on the hyperedge $e_{c_L,i}$ cannot become the same class as an existing class due to the algorithm. $\square$

**Theorem B.13.** *Let $|\mathcal{V}| = n, L \in \mathbb{N}^{+}$. If $vol(v) \triangleq \sum_{e \in \mathcal{E}: e \ni v} |e| = O(\log^{\frac{1-\epsilon}{4L}} n), \forall v \in \mathcal{V}$ for any constant $\epsilon > 0$; $|\mathcal{S}_{c_L}| \leq S, \forall c_L \in \mathcal{G}_L$, $S$ constant, and $|V_{c_L,s}| = O(\frac{n^{\epsilon}}{\log^{\frac{1}{2k}}(n)}), \forall s \in \mathcal{C}_{c_L}$ , then for $k \in \mathbb{N}^{+}$ and $k$-tuple $C = (c_{L,1}...c_{L,k}), c_{L,i} \in \mathcal{G}_L, i = 1..k$ there exists $\omega(n^{2k\epsilon})$ many pairs of $k$-node sets $S_1 \not\simeq S_2$ such that $(h_u^L(H))_{u \in S_1} = (h_{v \in S_2}^L(H)) = C$, as ordered $k$-tuples, while $h(S_1, \hat{H}_L) \neq h(S_2, \hat{H}_L)$ also by $L$ steps of GWL-1.*

*Proof.*

**1. Constructing forests from the rooted universal cover trees** :

The first part of the proof is similar to the first part of the proof of Theorem 2 of Zhang et al. (2021).

Consider an arbitrary node $v \in \mathcal{V}$ and denote the $2L$-tree rooted at $v$ from the universal cover as $(\tilde{\mathcal{B}}^{2L}_{\mathcal{V},\mathcal{E}})_v$ as in Theorem B.8. As each node $v \in \mathcal{V}$ has volume $vol(v) = \sum_{v \in e} |e| = O(\log^{\frac{1-\epsilon}{4L}} n)$, then every edge $e \in \mathcal{E}$ has $|e| = O(\log^{\frac{1-\epsilon}{4L}} n)$ and for all $v \in \mathcal{V}$ we have that $deg(v) = O(\log^{\frac{1-\epsilon}{4L}} n)$, we can say that every node in $(\tilde{\mathcal{B}}^{2L}_{\mathcal{V},\mathcal{E}})_{\tilde{v}}$ has degree $d = O(\log^{\frac{1-\epsilon}{4L}} n)$. Thus, the number of nodes in $(\tilde{\mathcal{B}}^{2L}_{\mathcal{V},\mathcal{E}})_{\tilde{v}}$, denoted by $|\mathcal{V}((\tilde{\mathcal{B}}^{2L}_{\mathcal{V},\mathcal{E}})_{\tilde{v}}|$, satisfies $|\mathcal{V}((\tilde{\mathcal{B}}^{2L}_{\mathcal{V},\mathcal{E}})_{\tilde{v}}| \leq \sum_{i=0}^{2L} d^i = O(d^{2L}) = O(\log^{\frac{1-\epsilon}{2}} n)$. We set $K \triangleq \max_{v \in V} |\mathcal{V}((\tilde{\mathcal{B}}^{2L}_{\mathcal{V},\mathcal{E}})_{\tilde{v}}|$ as the maximum number of nodes of $(\tilde{\mathcal{B}}^{2L}_{\mathcal{V},\mathcal{E}})_{\tilde{v}}$ and thus $K = O(\log^{\frac{1-\epsilon}{2}} n)$. For all $v \in \mathcal{V}$, expand trees $(\tilde{\mathcal{B}}^{2L}_{\mathcal{V},\mathcal{E}})_{\tilde{v}}$ to $\overline{(\tilde{\mathcal{B}}^{2L}_{\mathcal{V},\mathcal{E}})_{\tilde{v}}}$ by adding $K - |\mathcal{V}((\tilde{\mathcal{B}}^{2L}_{\mathcal{V},\mathcal{E}})_{\tilde{v}}|$ independent nodes. Then, all $\overline{(\tilde{\mathcal{B}}^{L}_{\mathcal{V},\mathcal{E}})_{\tilde{v}}}$ have the same number of nodes, which is $K$, becoming forests instead of trees.

**2. Counting $|\mathcal{G}_L|$:**

Next, we consider the number of non-isomorphic forests over $K$ nodes. Actually, the number of non-isomorphic graph over K nodes is bounded by $2^{\binom{K}{2}} = exp(O(\log^{\frac{1-\epsilon}{2}} n)) = o(n^{1-\epsilon})$. Therefore, due to the pigeonhole principle, there exist $\frac{n}{o(n^{1-\epsilon})} = \omega(n^\epsilon)$ many nodes $v$ whose $\overline{(\tilde{\mathcal{B}}^{L}_{\mathcal{V},\mathcal{E}})_{\tilde{v}}}$ are isomorphic to each other. Denote $\mathcal{G}_L$ as the set of all $L$-GWL-1 values. Denote the set of these nodes as $\mathcal{V}_{c_L}$, which consist of nodes whose $L$-GWL-1 values are all the same value $c_L \in \mathcal{G}_L$ after $L$ iterations of GWL-1 by Theorem B.8. For a fixed $L$, the sets $\mathcal{V}_{c_L}$ form a partition of $\mathcal{V}$, in other words, $\bigsqcup_{c_L \in \mathcal{G}_L} \mathcal{V}_{c_L} = \mathcal{V}$. Next, we focus on looking at $k$-sets of nodes that are not equivalent by GWL-1.

For any $c_L \in \mathcal{G}_L$, there is a partition $\mathcal{V}_{c_L} = \bigsqcup_s V_{c_L,s}$ where $V_{c_L,s}$ is the set of nodes all of which have $L$-GWL-1 class $c_L$ and that belong to a connected component of size $s$ in $\mathcal{H}_{c_L}$. Let $\mathcal{S}_{c_L} \triangleq \{|\mathcal{V}_{\mathcal{R}_{c_L,i}}| : \mathcal{R}_{c_L,i} \in \mathcal{C}_{c_L}\}$ denote the set of sizes $s \geq 1$ of connected component node sets of $\mathcal{H}_{c_L}$. We know that $|\mathcal{S}_{c_L}| \leq S$ where $S$ is independent of $n$.

**3. Computing the lower bound:**

Let $Y$ denote the number of pairs of $k$-node sets $S_1 \ncong S_2$ such that $(h^L_u(H))_{u \in S_1} = (h^L_v(H))_{v \in S_2} = C = (c_{L,1}...c_{L,k})$, as ordered tuples, from $L$-steps of GWL-1. Since if any pair of nodes $u, v$ have same $L$-GWL-1 values $c_L$, then they become distinguishable by the size of the connected component in $\mathcal{H}_{c_L}$ that they belong to. We can lower bound $Y$ by counting over all pairs of $k$ tuples of nodes $((u_1...u_k), (v_1...v_k)) \in (\prod_{i=1}^k \mathcal{V}_{c_{L,i}}) \times (\prod_{i=1}^k \mathcal{V}_{c_{L,i}})$ that both have $L$-GWL-1 values $(c_{L,1}...c_{L,k})$ where there is atleast one $i \in \{1..k\}$ where $u_i$ and $v_i$ belong to different sized connected components $s_i, s_i' \in \mathcal{S}_{c_{L,i}}$ with $s_i \neq s_i'$. We have:

$$Y \geq \frac{1}{k!} \Big[ \sum_{\substack{((s_i)_{i=1}^k, (s_i')_{i=1}^k) \in [(\prod_{i=1}^k \mathcal{S}^L_{c_{L,i}})]^2 \\ :(s_i)_{i=1}^k \neq (s_i')_{i=1}^k}} \prod_{i=1}^k |V^L_{(c_{L,i}),s_i}||V^L_{(c_{L,i}),s_i'}| \Big] \tag{54a}$$

$$= \frac{1}{k!} \Big[ \prod_{i=1}^k \big( \sum_{s_i \in \mathcal{S}^L_{c_i}} |V^L_{(c_{L,i}),s_i}| \big)^2 - \sum_{(s_i)_{i=1}^k \in \prod_{i=1}^k \mathcal{S}^L_{(c_{L,i})}} \big( \prod_{i=1}^k |V^L_{(c_{L,i}),s_i}|^2 \big) \Big] \tag{54b}$$

Using the fact that for each $i \in \{1...k\}$, $|\mathcal{V}_{c_{L,i}}| = \sum_{s_i \in \mathcal{S}_{c_{L,i}}} |V_{(c_{L,i}),s_i}|$ and by assumption $|V_{(c_{L,i}),s_i}| = O(\frac{n^\epsilon}{\log^{\frac{1}{2k}} n})$ for any $s_i \in \mathcal{S}_{c_{L,i}}$, thus we have:

$$Y \geq \omega(n^{2k\epsilon}) - O(|S|^k \frac{n^{2k\epsilon}}{\log n})] = \omega(n^{2k\epsilon}) \tag{55}$$

$\square$

For the following proof, we will denote $\cong_{\mathcal{H}}$ as a node or hypergraph isomorphism with respect to a hypergraph $\mathcal{H}$.

**Theorem B.14** (Invariance and Expressivity). *If $L = \infty$, GWL-1 enhanced by Algorithm 1 is still invariant to node isomorphism classes of $\mathcal{H}$ and can be strictly more expressive than GWL-1 to determine node isomorphism classes.*

*Proof.*

**1. Expressivity**:

Let $L \in \mathbb{N}^+$ be arbitrary. We first prove that $L$-GWL-1 enhanced by Algorithm 1 is strictly more expressive for node distinguishing than $L$-GWL-1 on some hypergraph(s). Let $C_4^3$ and $C_5^3$ be two 3-regular hypergraphs from Figure 2. Let $\mathcal{H} = C_4^3 \bigsqcup C_5^3$ be the disjoint union of the two regular hypergraphs. $L$ iterations of GWL-1 will assign the same node class to all of $\mathcal{V}_{\mathcal{H}}$. These two subhypergraphs can be distinguished by $L$-GWL-1 for $L \geq 1$ after editing the hypergraph $\mathcal{H}$ from the output of Algorithm 1 and becoming $\hat{\mathcal{H}} = \hat{C}_4^3 \cup \hat{C}_5^3$. This is all shown in Figure 2. Since $L$ was arbitrary, this is true for $L = \infty$.

**2. Invariance**:

For any hypergraph $\mathcal{H}$, let $\hat{\mathcal{H}} = (\hat{\mathcal{V}}, \hat{\mathcal{E}})$ be $\mathcal{H}$ modified by the output of Algorithm 1 by adding hyperedges to $\mathcal{V}_{\mathcal{R}_{c,i}}$. GWL-1 remains invariant to node isomorphism classes of $\mathcal{H}$ on $\hat{\mathcal{H}}$.

**a. Case 1**:

Let $L \in \mathbb{N}^+$ be arbitrary. For any node $u$ with $L$-GWL-1 class $c$ changed to $c_s$ in $\hat{\mathcal{H}}$, if $u \cong_{\mathcal{H}} v$ for any $v \in \mathcal{V}$, then the GWL-1 class of $v$ must also be $c_s$. In otherwords, both $u$ and $v$ belong to $s$-sized connected components in $\mathcal{H}_c$ We prove this by contradiction.

Say $u$ belong to a $L$-GWL-1 symmetric induced subhypergraph $S$ with $|\mathcal{V}_S| = s$. Say $v$ is originally of $L$-GWL-1 class $c$ and changes to $L$-GWL-1 $c_{s'}$ for $s' < s$ on $\hat{\mathcal{H}}$, WLOG. If this is the case then $v$ belongs to a $L$-GWL-1 symmetric induced subhypergraph $S'$ with $|\mathcal{V}_{S'}| = s'$. Since there is a $\pi \in Aut(\mathcal{H})$ with $\pi(u) = v$ and since $s' < s$, by the pigeonhole principle some node $w \in \mathcal{V}_S$ must have $\pi(w) \notin \mathcal{V}_{S'}$. Since $S$ and $S'$ are maximally connected, $\widetilde{\pi(w)}$ cannot share the same $L$-GWL-1 class as $w$. Thus, it must be that $(\tilde{B}_{\mathcal{V},\mathcal{E}}^{2L})_{\widetilde{\pi(w)}} \not\cong_c (\tilde{B}_{\mathcal{V},\mathcal{E}}^{2L})_{\tilde{w}}$ where $\tilde{w}, \widetilde{\pi(w)}$ are the lifts of $w, \pi(w)$ by universal covering map $p_{\mathcal{B}_{\mathcal{V},\mathcal{E}}}$. However by the contrapositive of Theorem B.8, this means $\pi(w) \neq w$. However $w$ and $\pi(w)$ both belong to $L$-GWL-1 class $c$ in $\mathcal{H}$, meaning $(\tilde{B}_{\mathcal{V},\mathcal{E}}^{2L})_{\widetilde{\pi(w)}} \cong_c (\tilde{B}_{\mathcal{V},\mathcal{E}}^{2L})_{\tilde{w}}$, contradiction. The argument for when $v$ does not change its class $c$ after the algorithm, follows by noticing that $c \neq c_s$ as GWL-1 node classes of $u$ and $v$ implies $u \not\cong_{\mathcal{H}} v$ once again by the contrapositive of Theorem B.8. Since $L$ was arbitrary, this is true for $L = \infty$

**b. Case 2**:

Now assume $L = \infty$. Let $p_{\mathcal{B}_{\mathcal{V},\mathcal{E}}}$ be the universal covering map of $\mathcal{B}_{\mathcal{V},\mathcal{E}}$. For all other nodes $u' \cong_{\mathcal{H}} v'$ for $u', v' \in \mathcal{V}$ unaffected by the replacement, meaning they do not belong to any $\mathcal{R}_{c,i}$ discovered by the algorithm, if the rooted universal covering tree rooted at node $\tilde{u}'$ connects to any node $\tilde{w}$ in $l$ hops in $(\tilde{\mathcal{B}}_{\mathcal{V},\mathcal{E}}^l)_{\tilde{u}'}$ where $p_{\mathcal{B}_{\mathcal{V},\mathcal{E}}}(\tilde{u}') = u', p_{\mathcal{B}_{\mathcal{V},\mathcal{E}}}(\tilde{w}) = w$ and where $w$ has any class $c$ in $\mathcal{H}$, then $\tilde{v}'$ must also connect to a node $\tilde{z}$ in $l$ hops in $(\tilde{\mathcal{B}}_{\mathcal{V},\mathcal{E}}^l)_{\tilde{u}'}$ where $p_{\mathcal{B}_{\mathcal{V},\mathcal{E}}}(\tilde{z}) = z$ and $w \cong_{\mathcal{H}} z$. Furthermore, if $w$ becomes class $c_s$ in $\mathcal{H}$ due to the algorithm, then $z$ also becomes class $c_s$ in $\hat{\mathcal{H}}$. This will follow by the previous result on isomorphic $w$ and $z$ both of class $c$ with $w$ becoming class $c_s$ in $\hat{\mathcal{H}}$.

Since $L = \infty$: For any $w \in \mathcal{V}$ connected by some path of hyperedges to $u' \in \mathcal{V}$, consider the smallest $l$ for which $(\tilde{B}_{\mathcal{V},\mathcal{E}}^l)_{\tilde{u}'}$, the $l$-hop universal covering tree of $\mathcal{H}$ rooted at $\tilde{u}'$, the lift of $u'$, contains the lifted $\tilde{w}$ of $w \in \mathcal{V}$ with GWL-1 node class $c$ at layer $l$. Since $u' \cong_{\mathcal{H}} v'$ by $\pi$. We can use $\pi$ to find some $z = \pi(w)$.

We claim that $\tilde{z}$ is $l$ hops away from $\tilde{v}'$. Since $u' \cong_{\mathcal{H}} v'$ due to some $\pi \in Aut(\mathcal{H})$ with $\pi(u') = v'$, using Proposition B.2 for singleton nodes and by Theorem B.8 we must have $(\tilde{\mathcal{B}}_{\mathcal{V},\mathcal{E}}^l)_{\tilde{u}'} \cong_c (\tilde{\mathcal{B}}_{\mathcal{V},\mathcal{E}}^l)_{\tilde{v}'}$ as isomorphic rooted universal covering trees due to an induced isomorphism $\tilde{\pi}$ of $\pi$ where we define an induced isomorphism $\tilde{\pi} : (\tilde{\mathcal{B}}_{\mathcal{V},\mathcal{E}})_{\tilde{u}'} \to (\tilde{\mathcal{B}}_{\mathcal{V},\mathcal{E}})_{\tilde{v}'}$ between rooted universal covers $(\tilde{\mathcal{B}}_{\mathcal{V},\mathcal{E}})_{\tilde{u}'}$ and $(\tilde{\mathcal{B}}_{\mathcal{V},\mathcal{E}})_{\tilde{v}'}$ for $\tilde{u}', \tilde{v}' \in \mathcal{V}(\tilde{\mathcal{B}}_{\mathcal{V},\mathcal{E}})$ as $\tilde{\pi}(\tilde{a}) = \tilde{b}$ if $\pi(a) = b$ $\forall a, b \in \mathcal{V}(\mathcal{B}_{\mathcal{V},\mathcal{E}})$ connected to $u'$ and $v'$ respectively and $p_{\mathcal{B}_{\mathcal{V},\mathcal{E}}}(\tilde{a}) = a, p_{\mathcal{B}_{\mathcal{V},\mathcal{E}}}(\tilde{b}) = b$. Since $l$ is the shortest path distance from $\tilde{u}'$ to $\tilde{w}$, there must exist some shortest (as defined by the path length in $\mathcal{B}_{\mathcal{V},\mathcal{E}}$) path $P$ of hyperedges from $u'$ to $w$ with no cycles. Using $\pi$, we must map $P$ to another acyclic shortest path of the same length from $v'$ to $z$. This path correponds to a $l$ length shortest path from $\tilde{v}'$ to $\tilde{z}$ in $(\tilde{\mathcal{B}}_{\mathcal{V},\mathcal{E}})_{\tilde{v}'}$.

If $w$ has GWL-1 class $c$ in $\mathcal{H}$ that doesn't become affected by the algorithm, then $z$ also has GWL-1 class $c$ in $\mathcal{H}$ since $w \cong_{\mathcal{H}} z$.

If $w$ has class $c$ and becomes $c_s$ in $\hat{\mathcal{H}}$, by the previous result, since $w \cong_{\mathcal{H}} z$ we must have the GWL-1 classes $c'$ and $c''$ of $w$ and $z$ in $\hat{\mathcal{H}}$ be both equal to $c_s$.

The node $w$ connected to $u'$ was arbitrary and so both $\tilde{w}$ and the isomorphism induced $\tilde{z}$ are $l$ hops away from $\tilde{u}'$ and $\tilde{v}'$ respectively, with the same GWL-1 class $c'$ in $\hat{\mathcal{H}}$, thus $(\tilde{\mathcal{B}}_{\hat{\mathcal{V}}, \hat{\mathcal{E}}})_{\tilde{u}'} \cong_c (\tilde{\mathcal{B}}_{\hat{\mathcal{V}}, \hat{\mathcal{E}}})_{\tilde{v}'}$.

We have thus shown, if $u \cong_{\mathcal{H}} v$ for $u, v \in \mathcal{H}$, then in $\hat{\mathcal{H}}$ we have $h_u^L(\hat{H}) = h_v^L(\hat{H})$ using the duality between universal covers and GWL-1 from Theorem B.8.

$\square$

**Proposition B.15** (Complexity). *Let $H$ be the star expansion matrix of $\mathcal{H}$. Algorithm 1 runs in time $O(L \cdot nnz(H) + (n + m))$, the size of the input hypergraph when viewing $L$ as constant, where $n$ is the number of nodes, $nnz(H) = vol(\mathcal{V}) \triangleq \sum_{v \in \mathcal{V}} deg(v)$ and $m$ is the number of hyperedges.*

*Proof.* Computing $E_{deg}$, which requires computing the degrees of all the nodes in each hyperedge takes time $O(nnz(H))$. The set $E_{deg}$ can be stored as a hashset datastructure. Constructing this takes $O(nnz(H))$. Computing GWL-1 takes $O(L \cdot nnz(H))$ time assuming a constant $L$ number of iterations. Constructing the bipartite graphs for $H$ takes time $O(nnz(H) + n + m)$ since it is an information preserving data structure change. Define for each $c \in C$, $n_c := |\mathcal{V}_c|, m_c := |\mathcal{E}_c|$. Since the classes partition $\mathcal{V}$, we must have:

$$n = \sum_{c \in C} n_c; m = \sum_{c \in C} m_c; nnz(H) = \sum_{c \in C} nnz(H_c) \tag{56}$$

where $H_c$ is the star expansion matrix of $\mathcal{H}_c$. Extracting the subgraphs can be implemented as a masking operation on the nodes taking time $O(n_c)$ to form $\mathcal{V}_c$ followed by searching over the neighbors of $\mathcal{V}_c$ in time $O(m_c)$ to construct $\mathcal{E}_c$. Computing the connected components for $\mathcal{H}_c$ for a predicted node class $c$ takes time $O(n_c + m_c + nnz(H_c))$. Iterating over each connected component for a given $c$ and extracting their nodes and hyperedges takes time $O(n_{c_i} + m_{c_i})$ where $n_c = \sum_i n_{c_i}, m_c = \sum_i m_{c_i}$. Checking that a connected component has size at least 3 takes $O(1)$ time. Computing the degree on $\mathcal{H}$ for all nodes in the connected component takes time $O(n_{c_i})$ since computing degree takes $O(1)$ time. Checking that the set of node degrees of the connected component doesn't belong to $E_{deg}$ can be implemented as a check that the hash of the set of degrees is not in the hashset datastructure for $E_{deg}$.

Adding up all the time complexities, we get the total complexity is:

$$O(nnz(H)) + O(nnz(H) + n + m) + \sum_{c \in C} (O(n_c + m_c + nnz(H_c)) + \sum_{\text{conn. comp. } i \text{ of } \mathcal{H}_c} O(n_{c_i} + m_{c_i})) \tag{57a}$$

$$= O(nnz(H) + n + m) + \sum_{c \in C} (O(n_c + m_c + nnz(H_c)) + O(n_c + m_c)) \tag{57b}$$

$$= O(nnz(H) + n + m) \tag{57c}$$

$\square$

**Proposition B.16.** *For a connected hypergraph $\mathcal{H}$, let $(\mathcal{R}_V, \mathcal{R}_E)$ be the output of Algorithm 1 on $\mathcal{H}$. Then there are Bernoulli probabilities $p, q_i$ for $i = 1...|\mathcal{R}_V|$ for attaching a covering hyperedge so that $\hat{\pi}$ is an unbiased estimator of $\pi$.*

*Proof.* Let $\mathcal{C}_{c_L} = \{\mathcal{R}_{c_L, i}\}_i$ be the maximally connected components induced by the vertices with $L$-GWL-1 values $c_L$. The set of vertex sets $\{\mathcal{V}(\mathcal{R}_{c_L, i})\}$ and the set of all hyperedges $\cup_i \{\mathcal{E}(\mathcal{R}_{c_L, i})\}$ over all the connected components $\mathcal{R}_{c_L, i}$ for $i = 1...\mathcal{C}_{c_L}$ form the pair $(\mathcal{R}_V, \mathcal{R}_E)$.

For a hypergraph random walk on connected $\mathcal{H} = (\mathcal{V}, \mathcal{E})$, its stationary distribution $\pi$ on $\mathcal{V}$ is given by the closed form:

$$\pi(v) = \frac{deg(v)}{\sum_{u \in \mathcal{V}} deg(u)} \tag{58}$$

for $v \in \mathcal{V}$.

Let $\hat{\mathcal{H}} = (\mathcal{V}, \hat{\mathcal{E}})$ be the random hypergraph as determined by $p$ and $q_i$ for $i = 1...|\mathcal{R}_V|$. These probabilities determine $\hat{\mathcal{H}}$ by the following operations on the hypergraph $\mathcal{H}$:

- Attaching a single hyperedge that covers $\mathcal{V}_{\mathcal{R}_{c_L,i}}$ with probability $q_i$ and not attaching with probability $1 - q_i$.

- All the hyperedges in $\mathcal{R}_{c_L,i}$ are dropped/kept with probability $p$ and $1 - p$ respectively.

**1. Setup:**

Let $deg(v) \triangleq |\{e : e \ni v\}|$ for $v \in \mathcal{V}(\mathcal{H})$ and $deg(\mathcal{S}) \triangleq \sum_{u \in \mathcal{V}(\mathcal{S})} deg(u)$ for $\mathcal{S} \subset \mathcal{H}$ a subhypergraph.

Let

$$Bernoulli(p) \triangleq \begin{cases} 1 & \text{prob. } p \\ 0 & \text{prob. } 1 - p \end{cases} \tag{59}$$

and

$$Binom(n, p) \triangleq \sum_{i=1}^{n} Bernoulli(p) \tag{60}$$

Define for each $v \in \mathcal{V}$, $C(v)$ to be the unique $\mathcal{R}_{c_L,i}$ where $v \in \mathcal{R}_{c_L,i}$ This means that we have the following independent random variables:

$$X_e \triangleq Bernoulli(1 - p), \forall e \in \mathcal{E} \text{ (i.i.d. across all } e \in \mathcal{E}) \tag{61a}$$

$$X_{C(v)} \triangleq Bernoulli(q_i) \tag{61b}$$

As well as the following constant, depending only on $C(v)$:

$$m_{C(v)} \triangleq \sum_{u \in \mathcal{V} \backslash C(v)} deg(u) \tag{62}$$

where $C(v) \subset \mathcal{V}, \forall v \in \mathcal{V}$

Let $\hat{\pi}$ be the stationary distribution of $\hat{H}$. Its expectation $\mathbb{E}[\hat{\pi}]$ can be written as:

$$\hat{\pi}(v) \triangleq \frac{\sum_{e \ni v} X_e + X_{C(v)}}{m_{C(v)} + \sum_{e \ni u : u \in C(v)} X_e + X_{C(v)}} \tag{63}$$

Letting

$$N_v \triangleq \sum_{e \ni v} X_e = Binom(deg(v), 1 - p) \tag{64a}$$

$$N \triangleq N_v + X_{C(v)} \tag{64b}$$

$$D \triangleq m_{C(v)} + \sum_{e \ni u : u \in C(v)} X_e + X_{C(v)} \tag{64c}$$

$$C \triangleq D - (\sum_{e \ni v} |e|) N_v - m_{C(v)} = \sum_{e \ni u : v \notin e, u \in C(v)} X_e - m_{C(v)} = Binom(F_v, 1 - p) - m_{c(v)} \tag{64d}$$

$$\text{where } F_v \triangleq |\{e \ni u : v \notin e, u \in C(v)\}|$$

and so we have: $\hat{\pi}(v) = \frac{N}{D}$

We have the following joint independence $N_v \perp X_{C(v)} \perp C$ due to the fact that each random variable describes disjoint hyperedge sets.

**2. Computing the Expectation:**

Writing out the expectation with conditioning on the joint distribution $P(D, N_v, X_{C(v)})$, we have:

$$\mathbb{E}[\hat{\pi}(v)] = \sum_{i=0}^{1} \sum_{j=0}^{deg(v)} \sum_{k=m_{C(v)}}^{deg(C(v))} \mathbb{E}[\hat{\pi}(v)|D=k, N=j]P(D=k, N_v=j, X_{C(v)}=i) \tag{65a}$$

$$= \sum_{i=0}^{1} \sum_{j=0}^{deg(v)} \sum_{k=m_{C(v)}}^{deg(C(v))} \frac{1}{k}\mathbb{E}[N|D=k, N_v=j, X_{C(v)}=i]P(D=k, N_v=j, X_{C(v)}=i) \tag{65b}$$

$$= \sum_{i=0}^{1} \sum_{j=0}^{deg(v)} \sum_{k=m_{C(v)}}^{deg(C(v))} \frac{j+i}{k}P(D=k, N_v=j, X_{C(v)}=i) \tag{65c}$$

$$= \sum_{i=0}^{1} \sum_{j=0}^{deg(v)} \sum_{k=m_{C(v)}}^{deg(C(v))} \frac{j+i}{k}P(D=k)P(N_v=j)P(X_{C(v)}=i) \tag{65d}$$

$$= \sum_{i=0}^{1} \sum_{j=0}^{deg(v)} \sum_{k=m_{C(v)}}^{deg(C(v))} \frac{j+i}{k}P(C=k-deg(v)j-m_{C(v)})P(N_v=j)P(X_{C(v)}=i) \tag{65e}$$

$$= \sum_{i=0}^{1} \sum_{j=0}^{deg(v)} \sum_{k=m_{C(v)}}^{deg(C(v))} \frac{j+i}{k}P(Binom(F_v, 1-p), 1-p) = k-deg(v)j-m_{C(v)})$$
$$\cdot P(Binom(deg(v), 1-p) = j) \cdot P(Bernoulli(q) = i) \tag{65f}$$

$$= \sum_{i=0}^{1} \sum_{j=0}^{deg(v)} \sum_{k=m_{C(v)}}^{deg(C(v))} \frac{j+i}{k}\binom{F_v}{k-deg(v)j-m_{C(v)}}(\frac{1}{2})^{F_v} \cdot \binom{deg(v)}{j}(\frac{1}{2})^{deg(v)} \cdot P(Bernoulli(q) = i) \tag{65g}$$

$$= \sum_{i=0}^{1} \sum_{j=0}^{deg(v)} \sum_{k=m_{C(v)}}^{deg(C(v))} \frac{j+i}{k}\binom{F_v}{k-deg(v)j-m_{C(v)}}(\frac{1}{2})^{F_v}$$
$$\cdot \binom{deg(v)}{j}(\frac{1}{2})^{deg(v)} \cdot P(Bernoulli(q) = i) \tag{65h}$$

$$= \sum_{j=0}^{deg(v)} \sum_{k=m_{C(v)}}^{deg(C(v))} \binom{F_v}{k-deg(v)j-m_{C(v)}}(\frac{1}{2})^{F_v} \cdot \binom{deg(v)}{j}(\frac{1}{2})^{deg(v)}[(1-q)\frac{j}{k}+q\frac{j+1}{k}] \tag{65i}$$

$$= \sum_{j=0}^{deg(v)} \sum_{k=m_{C(v)}}^{deg(C(v))} \binom{F_v}{k-deg(v)j-m_{C(v)}}(1-p)^{F_v-(k-deg(v))}p^{k-deg(v)} \cdot \binom{deg(v)}{j}(1-p)^j p^{deg(v)-j}[\frac{j}{k}+q\frac{1}{k}] \tag{65j}$$

$$= C_1(p) + qC_2(p) \tag{65k}$$

**3. Pick $p$ and $q$:**

We want to find $p$ and $q$ so that $\mathbb{E}[\hat{\pi}(v)] = C_1(p) + qC_2(p) = \pi(v)$

We know that for a given $p \in [0, 1]$, we must have:

$$q = \frac{\pi(v) - C_1(p)}{C_2(p)} \tag{66}$$

In order for $q \in [0, 1]$, must have $\pi(v) \geq C_1(p)$ and $\pi(v) - C_1(p) \leq C_2(p)$.

**a. Pick $p$ sufficiently large:**

Notice that

$$0 \le C_1(p) \le O(\mathbb{E}[\frac{1}{Binom(F_v, 1-p) + m_{C(v)}}] \cdot \mathbb{E}[Binom(deg(v), 1-p)]) = O(\frac{1}{m_{C(v)}} deg(v)(1-p)) \quad (67)$$

and that

$$0 \le C_1(p) \le C_2(p) \quad (68)$$

for $p \in [0, 1]$ sufficiently large. This is because

$$C_1(p) \le O(\frac{1}{m_{C(v)}} deg(v)(1-p)) \quad (69)$$

and

$$\Omega(\frac{1}{m_{C(v)}} deg(v)(1-p)) \le C_2(p) \quad (70)$$

Piecing these two inequalities together gets the desired inequality 68.

We can then pick a $p \in [0, 1]$ even larger than the previous $p$ so that for the $C' > 0$ which gives $C_1(p) \le \frac{C'}{m_{C(v)}} deg(v)(1-p)$, we achieve

$$C_1(p) \le \frac{C'}{m_{C(v)}} deg(v)(1-p) < \pi(v) = \frac{deg(v)}{m_{C(v)} + \sum_{u \in C(v)} deg(u)} \quad (71)$$

We then have that there exists a $s > 1$ so that

$$sC_1(p) = \pi(v) \quad (72)$$

Using this relationship, we can then prove that for a sufficiently large $p \in [0, 1]$, we must have a $q \in [0, 1]$

**b. $p \in [0, 1]$ sufficiently large implies $q \ge 0$:**

We thus have $q \ge 0$ since its numerator is nonnegative:

$$\pi(v) - C_1(p) = (s - 1)C_1(p) \ge 0 \Rightarrow q \ge 0 \quad (73)$$

**c. $p \in [0, 1]$ sufficiently large implies $q \le 1$:**

$$\pi(v) - C_1(p) = sC_1(p) - C_1(p) = (s - 1)C_1(p) \le C_2(p) \Rightarrow q \le 1 \quad (74)$$

$\square$

## C  Additional Experiments

We show in Figure 5 the distributions for the component sizes over all connected components for samples from the Hy-MMSBM model Ruggeri et al. (2023). This is a model to sample hypergraphs with community structure. In Figure 5a we sample hypergraphs with 3 isolated communities, meaning that there is 0 chance of any interconnections between any two communities. In Figure 5b we sample hypergraphs with 3 communities where every node in a community has a weight of 1 to stay in its community and a weight of 0.2 to move out to any other community. We plot the boxplots as a function of increasing number of nodes. We notice that the more communication there is between communities for more nodes there is more spread in possible connected component sizes. Isolated communities should make for predictable clusters/connected components.

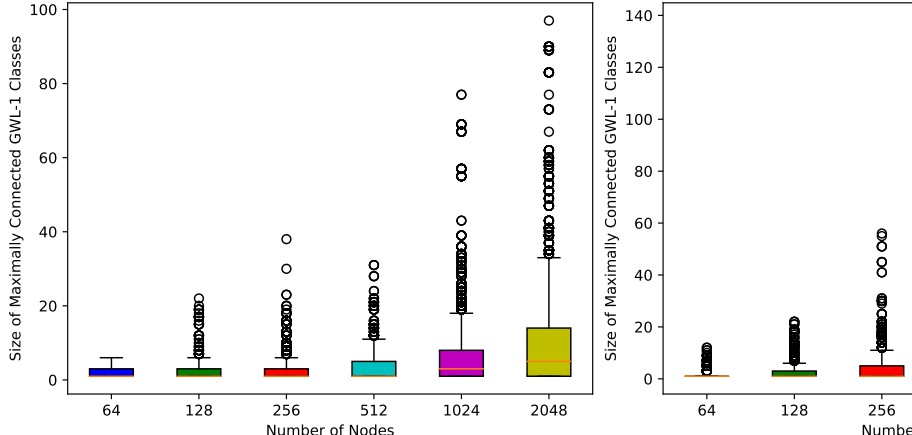

(a) Boxplot of the sizes of the connected components with equal GWL-1 node values from the hy-MMSBM sampling algorithm where there are three independent communities.

(b) Boxplot of the sizes of the connected components with equal GWL-1 node values from the hy-MMSBM sampling algorithm where any two of the three communities can communicate.

Figure 5

## C.1 Additional Experiments on Hypergraphs

In Table 3, we show the PR-AUC scores for four additional hypergraph datasets, CAT-EDGE-BRAIN, CAT-EDGE-VEGAS-BAR-REVIEWS, WIKIPEOPLE-0BI, and JF17K for predicting size 3 hyperedges.

| PR-AUC ↑ | Baseline | Ours | Baseln.+edrop |
|---|---|---|---|
| HGNN | $0.75 \pm 0.01$ | $0.79 \pm 0.11$ | $0.74 \pm 0.09$ |
| HGNNP | $0.75 \pm 0.05$ | $0.78 \pm 0.10$ | $0.74 \pm 0.12$ |
| HNHN | $0.74 \pm 0.04$ | $0.74 \pm 0.02$ | $0.74 \pm 0.05$ |
| HyperGCN | $0.74 \pm 0.09$ | $0.50 \pm 0.07$ | $0.50 \pm 0.12$ |
| UniGAT | $0.73 \pm 0.07$ | $0.81 \pm 0.10$ | $0.81 \pm 0.09$ |
| UniGCN | $0.78 \pm 0.04$ | $0.81 \pm 0.09$ | $0.71 \pm 0.08$ |
| UniGIN | $0.74 \pm 0.09$ | $0.74 \pm 0.03$ | $0.74 \pm 0.07$ |
| UniSAGE | $0.74 \pm 0.03$ | $0.74 \pm 0.12$ | $0.74 \pm 0.01$ |

(a) CAT-EDGE-BRAIN

| PR-AUC ↑ | Baseline | Ours | Baseln.+edrop |
|---|---|---|---|
| HGNN | $0.95 \pm 0.10$ | $0.99 \pm 0.04$ | $0.96 \pm 0.09$ |
| HGNNP | $0.95 \pm 0.06$ | $0.96 \pm 0.09$ | $0.96 \pm 0.08$ |
| HNHN | $1.00 \pm 0.08$ | $0.99 \pm 0.09$ | $0.95 \pm 0.10$ |
| HyperGCN | $0.76 \pm 0.03$ | $0.67 \pm 0.14$ | $0.68 \pm 0.09$ |
| UniGAT | $0.87 \pm 0.07$ | $1.00 \pm 0.09$ | $0.99 \pm 0.08$ |
| UniGCN | $0.99 \pm 0.07$ | $0.96 \pm 0.09$ | $0.92 \pm 0.05$ |
| UniGIN | $0.98 \pm 0.06$ | $0.96 \pm 0.08$ | $0.95 \pm 0.06$ |
| UniSAGE | $0.94 \pm 0.05$ | $0.98 \pm 0.07$ | $0.97 \pm 0.07$ |

(b) CAT-EDGE-VEGAS-BAR-REVIEWS

| PR-AUC ↑ | Baseline | Ours | Baseln.+edrop |
|---|---|---|---|
| HGNN | $0.52 \pm 0.01$ | $0.57 \pm 0.08$ | $0.54 \pm 0.10$ |
| HGNNP | $0.52 \pm 0.03$ | $0.54 \pm 0.07$ | $0.54 \pm 0.06$ |
| HNHN | $0.73 \pm 0.03$ | $0.73 \pm 0.07$ | $0.73 \pm 0.00$ |
| HyperGCN | $0.54 \pm 0.05$ | $0.55 \pm 0.02$ | $0.49 \pm 0.10$ |
| UniGAT | $0.49 \pm 0.09$ | $0.54 \pm 0.04$ | $0.53 \pm 0.04$ |
| UniGCN | $0.46 \pm 0.08$ | $0.68 \pm 0.08$ | $0.51 \pm 0.08$ |
| UniGIN | $0.73 \pm 0.09$ | $0.73 \pm 0.01$ | $0.73 \pm 0.02$ |
| UniSAGE | $0.73 \pm 0.06$ | $0.73 \pm 0.02$ | $0.73 \pm 0.08$ |

(c) WIKIPEOPLE-0BI

| PR-AUC ↑ | Baseline | Ours | Baseln.+edrop |
|---|---|---|---|
| HGNN | $0.59 \pm 0.04$ | $0.63 \pm 0.04$ | $0.45 \pm 0.09$ |
| HGNNP | $0.71 \pm 0.07$ | $0.63 \pm 0.07$ | $0.57 \pm 0.04$ |
| HNHN | $0.73 \pm 0.04$ | $0.73 \pm 0.03$ | $0.73 \pm 0.04$ |
| HyperGCN | $0.59 \pm 0.05$ | $0.58 \pm 0.09$ | $0.48 \pm 0.01$ |
| UniGAT | $0.61 \pm 0.07$ | $0.61 \pm 0.04$ | $0.51 \pm 0.08$ |
| UniGCN | $0.58 \pm 0.00$ | $0.60 \pm 0.03$ | $0.59 \pm 0.02$ |
| UniGIN | $0.80 \pm 0.04$ | $0.77 \pm 0.08$ | $0.75 \pm 0.05$ |
| UniSAGE | $0.79 \pm 0.02$ | $0.77 \pm 0.08$ | $0.74 \pm 0.01$ |

(d) JF17K

Table 3: PR-AUC on four other hypergraph datasets. The top average scores for each hyperGNN method, or row, is colored. Red scores denote the top scores in a row. Orange scores denote a two way tie and brown scores denote a threeway tie.

## C.2 Experiments on Graph Data

We show in Tables 4, 5, 6 the PR-AUC test scores for link prediction on some nonattributed graph datasets. The train-val-test splits are predefined for FB15K-237 and for the other graph datasets a single graph is deterministically split into 80/5/15 for train/val/test. We remove 10% of the edges in training and let them be positive examples $P_{tr}$ to predict. For validation and test, we remove 50% of the edges from both validation and test to set as the positive examples $P_{val}, P_{te}$ to predict. For train, validation, and test, we sample $1.2|P_{tr}|, 1.2|P_{val}|, 1.2|P_{te}|$ negative link samples from the links of train, validation and test. Along with HyperGNN architectures we use for the hypergraph experiments, we also compare with standard GNN architectures: APPNP Gasteiger et al. (2018), GAT Veličković et al. (2017), GCN2 Chen et al. (2020), GCN Kipf & Welling (2016a), GIN Xu et al. (2018), and GraphSAGE Hamilton et al. (2017). For every HyperGNN/GNN architecture, we also apply drop-edge Rong et al. (2019) to the input graph and use this also as baseline. The number of layers of each GNN is set to 5 and the hidden dimension at 1024. For APPNP and GCN2, one MLP is used on the initial node positional encodings. Since graphs do not have any hyperedges beyond size 2, graph neural networks fit the inductive bias of the graph data more easily and thus may perform better than hypergraph neural network baselines more often than expected.

| PR-AUC ↑ | HGNN | HGNNP | HNHN | HyperGCN | UniGAT | UniGCN | UniGIN | UniSAGE |
|---|---|---|---|---|---|---|---|---|
| Ours | 0.71 ± 0.04 | 0.71 ± 0.09 | 0.69 ± 0.09 | 0.75 ± 0.14 | 0.75 ± 0.09 | 0.74 ± 0.09 | 0.65 ± 0.08 | 0.65 ± 0.07 |
| HyperGNN Baseline | 0.68 ± 0.00 | 0.69 ± 0.06 | 0.67 ± 0.02 | 0.75 ± 0.04 | 0.74 ± 0.02 | 0.74 ± 0.00 | 0.65 ± 0.05 | 0.64 ± 0.08 |
| HyperGNN Baseln.+edrop | 0.67 ± 0.02 | 0.70 ± 0.07 | 0.66 ± 0.00 | 0.75 ± 0.03 | 0.73 ± 0.08 | 0.74 ± 0.05 | 0.63 ± 0.01 | 0.64 ± 0.03 |
| APPNP | 0.40 ± 0.03 | 0.40 ± 0.03 | 0.40 ± 0.03 | 0.40 ± 0.03 | 0.40 ± 0.03 | 0.40 ± 0.03 | 0.40 ± 0.03 | 0.40 ± 0.03 |
| APPNP+edrop | 0.40 ± 0.13 | 0.40 ± 0.13 | 0.40 ± 0.13 | 0.40 ± 0.13 | 0.40 ± 0.13 | 0.40 ± 0.13 | 0.40 ± 0.13 | 0.40 ± 0.13 |
| GAT | 0.49 ± 0.03 | 0.49 ± 0.03 | 0.49 ± 0.03 | 0.49 ± 0.03 | 0.49 ± 0.03 | 0.49 ± 0.03 | 0.49 ± 0.03 | 0.49 ± 0.03 |
| GAT+edrop | 0.51 ± 0.05 | 0.51 ± 0.05 | 0.51 ± 0.05 | 0.51 ± 0.05 | 0.51 ± 0.05 | 0.51 ± 0.05 | 0.51 ± 0.05 | 0.51 ± 0.05 |
| GCN2 | 0.50 ± 0.09 | 0.50 ± 0.09 | 0.50 ± 0.09 | 0.50 ± 0.09 | 0.50 ± 0.09 | 0.50 ± 0.09 | 0.50 ± 0.09 | 0.50 ± 0.09 |
| GCN2+edrop | 0.56 ± 0.07 | 0.56 ± 0.07 | 0.56 ± 0.07 | 0.56 ± 0.07 | 0.56 ± 0.07 | 0.56 ± 0.07 | 0.56 ± 0.07 | 0.56 ± 0.07 |
| GCN | 0.73 ± 0.02 | 0.73 ± 0.02 | 0.73 ± 0.02 | 0.73 ± 0.02 | 0.73 ± 0.02 | 0.73 ± 0.02 | 0.73 ± 0.02 | 0.73 ± 0.02 |
| GCN+edrop | 0.73 ± 0.01 | 0.73 ± 0.01 | 0.73 ± 0.01 | 0.73 ± 0.01 | 0.73 ± 0.01 | 0.73 ± 0.01 | 0.73 ± 0.01 | 0.73 ± 0.01 |
| GIN | 0.73 ± 0.06 | 0.73 ± 0.06 | 0.73 ± 0.06 | 0.73 ± 0.06 | 0.73 ± 0.06 | 0.73 ± 0.06 | 0.73 ± 0.06 | 0.73 ± 0.06 |
| GIN+edrop | 0.73 ± 0.01 | 0.73 ± 0.01 | 0.73 ± 0.01 | 0.73 ± 0.01 | 0.73 ± 0.01 | 0.73 ± 0.01 | 0.73 ± 0.01 | 0.73 ± 0.01 |
| GraphSAGE | 0.73 ± 0.08 | 0.73 ± 0.08 | 0.73 ± 0.08 | 0.73 ± 0.08 | 0.73 ± 0.08 | 0.73 ± 0.08 | 0.73 ± 0.08 | 0.73 ± 0.08 |
| GraphSAGE+edrop | 0.73 ± 0.09 | 0.73 ± 0.09 | 0.73 ± 0.09 | 0.73 ± 0.09 | 0.73 ± 0.09 | 0.73 ± 0.09 | 0.73 ± 0.09 | 0.73 ± 0.09 |

Table 4: PR-AUC on graph dataset JOHNSHOPKINS55. Each column is a comparison of the baseline PR-AUC scores against the PR-AUC score for our method (first row) applied to a standard HyperGNN architecture. Red color denotes the highest average score in the column. Orange color denotes a two-way tie in the column, and brown color denotes a three-or-more-way tie in the column.

| PR-AUC ↑ | HGNN | HGNNP | HNHN | HyperGCN | UniGAT | UniGCN | UniGIN | UniSAGE |
|---|---|---|---|---|---|---|---|---|
| Ours | 0.66 ± 0.06 | 0.78 ± 0.02 | 0.63 ± 0.07 | 0.82 ± 0.10 | 0.75 ± 0.05 | 0.74 ± 0.03 | 0.75 ± 0.03 | 0.75 ± 0.06 |
| HyperGNN Baseline | 0.65 ± 0.06 | 0.65 ± 0.06 | 0.65 ± 0.04 | 0.82 ± 0.09 | 0.74 ± 0.04 | 0.74 ± 0.05 | 0.75 ± 0.03 | 0.77 ± 0.01 |
| HyperGNN Baseln.+edrop | 0.65 ± 0.09 | 0.65 ± 0.00 | 0.64 ± 0.05 | 0.82 ± 0.00 | 0.72 ± 0.00 | 0.74 ± 0.07 | 0.73 ± 0.03 | 0.72 ± 0.07 |
| APPNP | 0.72 ± 0.10 | 0.72 ± 0.10 | 0.72 ± 0.10 | 0.72 ± 0.10 | 0.72 ± 0.10 | 0.72 ± 0.10 | 0.72 ± 0.10 | 0.72 ± 0.10 |
| APPNP+edrop | 0.71 ± 0.05 | 0.71 ± 0.05 | 0.71 ± 0.05 | 0.71 ± 0.05 | 0.71 ± 0.05 | 0.71 ± 0.05 | 0.71 ± 0.05 | 0.71 ± 0.05 |
| GAT | 0.64 ± 0.06 | 0.64 ± 0.06 | 0.64 ± 0.06 | 0.64 ± 0.06 | 0.64 ± 0.06 | 0.64 ± 0.06 | 0.64 ± 0.06 | 0.64 ± 0.06 |
| GAT+edrop | 0.61 ± 0.09 | 0.61 ± 0.09 | 0.61 ± 0.09 | 0.61 ± 0.09 | 0.61 ± 0.09 | 0.61 ± 0.09 | 0.61 ± 0.09 | 0.61 ± 0.09 |
| GCN2 | 0.66 ± 0.03 | 0.66 ± 0.03 | 0.66 ± 0.03 | 0.66 ± 0.03 | 0.66 ± 0.03 | 0.66 ± 0.03 | 0.66 ± 0.03 | 0.66 ± 0.03 |
| GCN2+edrop | 0.65 ± 0.10 | 0.65 ± 0.10 | 0.65 ± 0.10 | 0.65 ± 0.10 | 0.65 ± 0.10 | 0.65 ± 0.10 | 0.65 ± 0.10 | 0.65 ± 0.10 |
| GCN | 0.69 ± 0.03 | 0.69 ± 0.03 | 0.69 ± 0.03 | 0.69 ± 0.03 | 0.69 ± 0.03 | 0.69 ± 0.03 | 0.69 ± 0.03 | 0.69 ± 0.03 |
| GCN+edrop | 0.71 ± 0.06 | 0.71 ± 0.06 | 0.71 ± 0.06 | 0.71 ± 0.06 | 0.71 ± 0.06 | 0.71 ± 0.06 | 0.71 ± 0.06 | 0.71 ± 0.06 |
| GIN | 0.73 ± 0.03 | 0.73 ± 0.03 | 0.73 ± 0.03 | 0.73 ± 0.03 | 0.73 ± 0.03 | 0.73 ± 0.03 | 0.73 ± 0.03 | 0.73 ± 0.03 |
| GIN+edrop | 0.56 ± 0.07 | 0.56 ± 0.07 | 0.56 ± 0.07 | 0.56 ± 0.07 | 0.56 ± 0.07 | 0.56 ± 0.07 | 0.56 ± 0.07 | 0.56 ± 0.07 |
| GraphSAGE | 0.46 ± 0.15 | 0.46 ± 0.15 | 0.46 ± 0.15 | 0.46 ± 0.15 | 0.46 ± 0.15 | 0.46 ± 0.15 | 0.46 ± 0.15 | 0.46 ± 0.15 |
| GraphSAGE+edrop | 0.47 ± 0.01 | 0.47 ± 0.01 | 0.47 ± 0.01 | 0.47 ± 0.01 | 0.47 ± 0.01 | 0.47 ± 0.01 | 0.47 ± 0.01 | 0.47 ± 0.01 |

Table 5: PR-AUC on graph dataset FB15K-237. Each column is a comparison of the baseline PR-AUC scores against the PR-AUC score for our method (first row) applied to a standard HyperGNN architecture. Red color denotes the highest average score in the column. Orange color denotes a two-way tie in the column, and brown color denotes a three-or-more-way tie in the column.

| PR-AUC ↑ | HGNN | HGNNP | HNHN | HyperGCN | UniGAT | UniGCN | UniGIN | UniSAGE |
|---|---|---|---|---|---|---|---|---|
| Ours | 0.79 ± 0.11 | 0.73 ± 0.10 | 0.73 ± 0.02 | 0.85 ± 0.07 | 0.75 ± 0.10 | 0.84 ± 0.09 | 0.72 ± 0.03 | 0.72 ± 0.12 |
| HyperGNN Baseline | 0.72 ± 0.07 | 0.72 ± 0.07 | 0.72 ± 0.06 | 0.85 ± 0.05 | 0.75 ± 0.09 | 0.84 ± 0.05 | 0.72 ± 0.07 | 0.72 ± 0.06 |
| HyperGNN Baseln.+edrop | 0.72 ± 0.05 | 0.72 ± 0.08 | 0.72 ± 0.06 | 0.85 ± 0.07 | 0.73 ± 0.09 | 0.84 ± 0.06 | 0.72 ± 0.03 | 0.72 ± 0.07 |
| APPNP | 0.81 ± 0.12 | 0.81 ± 0.12 | 0.81 ± 0.12 | 0.81 ± 0.12 | 0.81 ± 0.12 | 0.81 ± 0.12 | 0.81 ± 0.12 | 0.81 ± 0.12 |
| APPNP+edrop | 0.80 ± 0.05 | 0.80 ± 0.05 | 0.80 ± 0.05 | 0.80 ± 0.05 | 0.80 ± 0.05 | 0.80 ± 0.05 | 0.80 ± 0.05 | 0.80 ± 0.05 |
| GAT | 0.50 ± 0.02 | 0.50 ± 0.02 | 0.50 ± 0.02 | 0.50 ± 0.02 | 0.50 ± 0.02 | 0.50 ± 0.02 | 0.50 ± 0.02 | 0.50 ± 0.02 |
| GAT+edrop | 0.33 ± 0.02 | 0.33 ± 0.02 | 0.33 ± 0.02 | 0.33 ± 0.02 | 0.33 ± 0.02 | 0.33 ± 0.02 | 0.33 ± 0.02 | 0.33 ± 0.02 |
| GCN2 | 0.83 ± 0.05 | 0.83 ± 0.05 | 0.83 ± 0.05 | 0.83 ± 0.05 | 0.83 ± 0.05 | 0.83 ± 0.05 | 0.83 ± 0.05 | 0.83 ± 0.05 |
| GCN2+edrop | 0.78 ± 0.04 | 0.78 ± 0.04 | 0.78 ± 0.04 | 0.78 ± 0.04 | 0.78 ± 0.04 | 0.78 ± 0.04 | 0.78 ± 0.04 | 0.78 ± 0.04 |
| GCN | 0.73 ± 0.14 | 0.73 ± 0.14 | 0.73 ± 0.14 | 0.73 ± 0.14 | 0.73 ± 0.14 | 0.73 ± 0.14 | 0.73 ± 0.14 | 0.73 ± 0.14 |
| GCN+edrop | 0.75 ± 0.08 | 0.75 ± 0.08 | 0.75 ± 0.08 | 0.75 ± 0.08 | 0.75 ± 0.08 | 0.75 ± 0.08 | 0.75 ± 0.08 | 0.75 ± 0.08 |
| GIN | 0.73 ± 0.00 | 0.73 ± 0.00 | 0.73 ± 0.00 | 0.73 ± 0.00 | 0.73 ± 0.00 | 0.73 ± 0.00 | 0.73 ± 0.00 | 0.73 ± 0.00 |
| GIN+edrop | 0.73 ± 0.10 | 0.73 ± 0.10 | 0.73 ± 0.10 | 0.73 ± 0.10 | 0.73 ± 0.10 | 0.73 ± 0.10 | 0.73 ± 0.10 | 0.73 ± 0.10 |
| GraphSAGE | 0.46 ± 0.15 | 0.46 ± 0.15 | 0.46 ± 0.15 | 0.46 ± 0.15 | 0.46 ± 0.15 | 0.46 ± 0.15 | 0.46 ± 0.15 | 0.46 ± 0.15 |
| GraphSAGE+edrop | 0.47 ± 0.01 | 0.47 ± 0.01 | 0.47 ± 0.01 | 0.47 ± 0.01 | 0.47 ± 0.01 | 0.47 ± 0.01 | 0.47 ± 0.01 | 0.47 ± 0.01 |

Table 6: PR-AUC on graph dataset AIFB. Each column is a comparison of the baseline PR-AUC scores against the PR-AUC score for our method (first row) applied to a standard HyperGNN architecture. Red color denotes the highest average score in the column. Orange color denotes a two-way tie in the column, and brown color denotes a three-or-more-way tie in the column.

## D   Dataset and Hyperparameters

Table 7 lists the datasets and hyperparameters used in our experiments. All datasets are originally from Benson et al. (2018b) or are general hypergraph datasets provided in Sinha et al. (2015); Amburg et al. (2020a). We list the total number of hyperedges $|\mathcal{E}|$, the total number of vertices $|\mathcal{V}|$, the positive to negative label ratios for train/val/test, and the percentage of the connected components searched over by our algorithm that are size atleast 3. A node isomorphism class is determined by our isomorphism testing algorithm. By Proposition B.2 we can guarantee that if two nodes are in separate isomorphism classes by our isomorphism tester, then they are actually nonisomorphic.

We use 1024 dimensions for all HyperGNN/GNN layer latent spaces, 5 layers for all hypergraph/graph neural networks, and a common learning rate of 0.01. Exactly 2000 epochs are used for training.

The HyperGNN architecture baselines are described in the follwoing:

- HGNN Feng et al. (2019) A neural network that generalizes the graph convolution to hypergraphs where there are hyperedge weights. Its architecture can be described by the following update step for the $l + 1$-layer from the $l$th layer:

$$X^{(l+1)} = \sigma(D_v^{-\frac{1}{2}} H W D_e^{-1} H^T D_v^{-\frac{1}{2}} X^{(l)} W^{(l)}) \tag{75}$$

  where $D_v \in \mathbb{R}^{n \times n}$ is the diagonal node degree matrix, $D_e \in \mathbb{R}^{m \times m}$ is the diagonal hyperedge degree matrix, $H \in \mathbb{R}^{n \times m}$ is the star incidence matrix, $X^{(l)} \in \mathbb{R}^{n \times d}$ is a node signal matrix, $W^{(l)} \in \mathbb{R}^{d \times d}$ is a weight matrix, and $\sigma$ is a nonlinear activation. Following the matrix products, as a message passing neural network, HGNN is GWL-1 based since the nodes pass to the hyperedges and back.

- HGNNP Feng et al. (2023) is an improved version of HGNN where asymmetry is introduced into the message passing weightings to distinguish the vertices from the hyperedges. This is also a GWL-1 based message passing neural network. It is described by the following node signal update equation:

$$X^{(l+1)} = \sigma(D_v^{-1} H W D_e^{-1} H^T X^{(l)} W^{(l)}) \tag{76}$$

  where the matrices are exactly the same as from HGNN.

- HyperGCN Yadati et al. (2019) computes GCN on a clique expansion of a hypergraph. This has an updateable adjacency matrix defined as follows:

$$A_{i,j}^{(l)} = \begin{cases} 1 & (i,j) \in E^{(l)} \\ 0 & (i,j) \notin E^{(l)} \end{cases} \tag{77}$$

  where

$$E^{(l)} = \{(i_e, j_e) = argmax_{i,j \in e} |X_i^{(l)} - X_j^{(l)}| : e \in \mathcal{E}\} \tag{78}$$

$$X_v^{(l+1)} = \sigma(\sum_{u \in N(v)} ([A^{(l)}]_{v,u} X_u^{(l)} W^{(l)})) \tag{79}$$

  The $X^{(l)} \in \mathbb{R}^{n \times d}$ is the node signal matrix at layer $l$, the $W^{(l)} \in \mathbb{R}^{d \times d}$ is the weight matrix at layer $l$, and $\sigma$ is some nonlinear activation. This architecture has less expressive power than GWL-1.

- HNHN Dong et al. (2020) This is like HGNN but where the message passing is explicitly broken up into two hyperedge to node and node to hyperedge layers.

$$X_E^{(l)} = \sigma(H^T X_V^{(l)} W_E^{(l)} + b_E^{(l)}) \tag{80a}$$

  and

$$X_V^{(l+1)} = \sigma(H X_E^{(l)} W_V^{(l)} + b_V^{(l)}) \tag{80b}$$

  where $H \in \mathbb{R}^{n \times m}$ is the star expansion incidence matrix, $W_E^{(l)}, W_V^{(l)} \in \mathbb{R}^{d \times d}, b_E^{(l)} \in \mathbb{R}^m, b_V^{(l)} \in \mathbb{R}^n$ are weights and biases, $X_E^{(l)}, X_V^{(l)}$ are the hyperedge and node signal matrices at layer $l$, and $\sigma$ is a nonlinear activation function. The bias vectors prevent HNHN from being permutation equivariant.

- UniGNN Huang & Yang (2021) The idea is directly related to generalizing WL-1 GNNs to Hypergraphs. Define the following hyperedge representation for hyperedge $e \in \mathcal{E}$:

$$h_e^{(l)} = \frac{1}{|e|} \sum_{u \in e} X_u^{(l)} \tag{81}$$

  - UniGCN: a generalization of GCN to hypergraphs

$$X_v^{(l)} = \frac{1}{\sqrt{d_v}} \sum_{e \ni v} \frac{1}{\sqrt{d_e}} W^{(l)} h_e^{(l)} \tag{82}$$

  - UniGAT: a generalization of GAT to hypergraphs

$$\alpha_{ue} = \sigma(a^T [X_i^{(l)} W^{(l)}; X_j^{(l)} W^{(l)}]) \tag{83a}$$

$$\tilde{\alpha}_{ue} = \frac{e^{\alpha_{ue}}}{\sum_{v \in e} e^{\alpha_{ve}}} \tag{83b}$$

$$X_v^{(l+1)} = \sum_{e \ni v} \alpha_{ve} h_e W^{(l)} \tag{83c}$$

  - UniGIN: a generalization of GIN to hypergraphs

$$X_v^{(l+1)} = ((1+\epsilon) X_v^{(l)} + \sum_{e \ni v} h_e) \tag{84}$$

  - UniSAGE: a generalization of GraphSAGE to hypergraphs

$$X_v^{(l+1)} = (X_v^{(l)} + \sum_{e \ni v} (h_e)) \tag{85}$$

All positional encodings are computed from the training hyperedges before data augmentation. The loss we use for higher order link prediction is the Binary Cross Entropy Loss for all the positive and negatives samples. Hypergraph neural network implementations were mostly taken from `https://github.com/iMoonLab/DeepHypergraph`, which uses the Apache License 2.0.

| Dataset Information | | | | | | |
|---|---|---|---|---|---|---|
| Dataset | $\|\mathcal{E}\|$ | $\|\mathcal{V}\|$ | $\frac{\Delta_{+,tr}}{\Delta_{-,tr}}$ | $\frac{\Delta_{+,val}}{\Delta_{-,val}}$ | $\frac{\Delta_{+,te}}{\Delta_{-,te}}$ | % of Conn. Comps. Selected |
| CAT-EDGE-DAWN | 87,104 | 2,109 | 8,802/10,547 | 1,915/2,296 | 1,867/2,237 | 0.05% |
| EMAIL-EU | 234,760 | 998 | 1,803/2,159 | 570/681 | 626/749 | 0.6% |
| CONTACT-PRIMARY-SCHOOL | 106,879 | 242 | 1,620/1,921 | 461/545 | 350/415 | 9.3% |
| CAT-EDGE-MUSIC-BLUES-REVIEWS | 694 | 1,106 | 16/19 | 7/6 | 3/3 | 0.14% |
| CAT-EDGE-VEGAS-BARS-REVIEWS | 1,194 | 1,234 | 72/86 | 12/14 | 11/13 | 0.7% |
| CONTACT-HIGH-SCHOOL | 7,818 | 327 | 2,646/3,143 | 176/208 | 175/205 | 5.6% |
| CAT-EDGE-BRAIN | 21,180 | 638 | 13,037/13,817 | 2,793/3,135 | 2,794/3,020 | 9.9% |
| JOHNSHOPKINS55 | 298,537 | 5,163 | 29,853/35,634 | 9,329/11,120 | 27,988/29,853 | 2.0% |
| AIFB | 46,468 | 8,083 | 4,646/5,575 | 1,452/1,739 | 4,356/5,222 | 0.02% |
| AMHERST41 | 145,526 | 2,234 | 14,552/17,211 | 4,547/5,379 | 16,125/13,643 | 4.4% |
| FB15K-237 | 272,115 | 14,505 | 27,211/32,630 | 8,767/10,509 | 10,233/12,271 | 2.1% |
| WIKIPEOPLE-0BI | 18,828 | 43,388 | 27,211/32,630 | 10,254/12,301 | 1,164/1,396 | 0.05% |
| JF17K | 76,379 | 28,645 | 11,907/14,287 | 1,341/1,608 | 1,341/1,608 | 0.6% |

Table 7: Dataset statistics and training hyperparameters used for all datasets in scoring all experiments.

We describe here some more information about each dataset we use in our experiments as provided by Benson et al. (2018b): Here is some information about the hypergraph datasets:

- Amburg et al. (2020a) CAT-EDGE-DAWN: Here nodes are drugs, hyperedges are combinations of drugs taken by a patient prior to an emergency room visit and edge categories indicate the patient disposition (e.g., "sent home", "surgery", "released to detox").

- Benson et al. (2018a); Yin et al. (2017); Leskovec et al. (2007)EMAIL-EU: This is a temporal higher-order network dataset, which here means a sequence of timestamped simplices, or hyperedges with all its node subsets existing as hyperedges, where each simplex is a set of nodes. In email communication, messages can be sent to multiple recipients. In this dataset, nodes are email addresses at a European research institution. The original data source only contains (sender, receiver, timestamp) tuples, where timestamps are recorded at 1-second resolution. Simplices consist of a sender and all receivers such that the email between the two has the same timestamp. We restricted to simplices that consist of at most 25 nodes.

- Stehlé et al. (2011) CONTACT-PRIMARY-SCHOOL: This is a temporal higher-order network dataset, which here means a sequence of timestamped simplices where each simplex is a set of nodes. The dataset is constructed from interactions recorded by wearable sensors by people at a primary school. The sensors record interactions at a resolution of 20 seconds (recording all interactions from the previous 20 seconds). Nodes are the people and simplices are maximal cliques of interacting individuals from an interval.

- Amburg et al. (2020b)CAT-EDGE-VEGAS-BARS-REVIEWS: Hypergraph where nodes are Yelp users and hyperedges are users who reviewed an establishment of a particular category (different types of bars in Las Vegas, NV) within a month timeframe.

- Benson et al. (2018a); Mastrandrea et al. (2015) CONTACT-HIGH-SCHOOL: This is a temporal higher-order network dataset, which here means a sequence of timestamped simplices where each simplex is a set of nodes. The dataset is constructed from interactions recorded by wearable sensors by people at a high school. The sensors record interactions at a resolution of 20 seconds (recording all interactions from the previous 20 seconds). Nodes are the people and simplices are maximal cliques of interacting individuals from an interval.

- Crossley et al. (2013) CAT-EDGE-BRAIN: This is a graph whose edges have categorical edge labels. Nodes represent brain regions from an MRI scan. There are two edge categories: one for connecting regions with high fMRI correlation and one for connecting regions with similar activation patterns.

- Lim et al. (2021)JOHNSHOPKINS55: Non-homophilous graph datasets from the facebook100 dataset.

- Ristoski & Paulheim (2016)AIFB: The AIFB dataset describes the AIFB research institute in terms of its staff, research groups, and publications. The dataset was first used to predict the affiliation (i.e., research group) for people in the dataset. The dataset contains 178 members of five research groups, however, the smallest group contains only four people, which is removed from the dataset, leaving four classes.

- Lim et al. (2021)AMHERST41: Non-homophilous graph datasets from the facebook100 dataset.

- Bordes et al. (2013)FB15K-237: A subset of entities that are also present in the Wikilinks database Singh et al. (2012) and that also have at least 100 mentions in Freebase (for both entities and relationships). Relationships like '!/people/person/nationality' which just reverses the head and tail compared to the relationship '/people/person/nationality' are removed. This resulted in 592,213 triplets with 14,951 entities and 1,345 relationships which were randomly split.

- Guan et al. (2019)WIKIPEOPLE-0BI: The Wikidata dump was downloaded and the facts concerning entities of type human were extracted. These facts are denoised. Subsequently, the subsets of elements which have at least 30 mentions were selected. And the facts related to these elements were kept. Further, each fact was parsed into a set of its role-value pairs. The remaining facts were randomly split into training set, validation set and test set by a percentage of 80%:10%:10%. All binary relations are removed for simplicity. This modifies WikiPeople to WikiPeople-0bi.

- Wen et al. (2016)JF17K: The full Freebase data in RDF format was downloaded. Entities involved in very few triples and the triples involving String, Enumeration Type and Numbers were removed. A fact representation was recovered from the remaining triples. Facts from meta-relations having only a single role were removed. From each meta-relation containing more than 10,000 facts, 10,000 facts were randomly selected.

### D.1 Timings

We perform experiments on a cluster of machines equipped with AMD MI100s GPUs and 112 shared AMD EPYC 7453 28-Core Processors with 2.6 PB shared RAM. We show here the times for computing each method. The timings may vary heavily for different machines as the memory we used is shared and during peak usage there is a lot of paging. We notice that although our data preprocessing algorithm involves seemingly costly steps such as GWL-1, connected connected components etc. The complexity of the entire preprocessing algorithm is linear in the size of the input as shown in Proposition B.15. Thus these operations are actually very efficient in practice as shown by Tables 9 and 10 for the hypergraph and graph datasets respectively. The preprocessing algorithm is run on CPU while the training is run on GPU for 2000 epochs.

| Timings (hh:mm) $\pm$ (s) | | |
|---|---|---|
| Method | Preprocessing Time | Training Time |
| HGNN | 2m:45s±108s | 35m:9s±13s |
| HGNNP | 1m:52s±0s | 35m:16s±0s |
| HNHN | 1m:55s±0s | 35m:0s±1s |
| HyperGCN | 1m:50s±0s | 58m:17s±79s |
| UniGAT | 1m:54s±0s | 1h:19m:34s±0s |
| UniGCN | 1m:50s±2s | 35m:19s±2s |
| UniGIN | 1m:50s±1s | 35m:12s±1288s |
| UniSAGE | 1m:51s±0s | 35m:16s±0s |

(a) CAT-EDGE-DAWN

| Timings (hh:mm) $\pm$ (s) | | |
|---|---|---|
| Method | Preprocessing Time | Training Time |
| HGNN | 1.72s±5s | 2m:11s±11s |
| HGNNP | 1.42s±0s | 2m:10s±0s |
| HNHN | 1.99s±0s | 3m:43s±2s |
| HyperGCN | 1.47s±2s | 4m:12s±3s |
| UniGAT | 1.85s±0s | 3m:54s±287s |
| UniGCN | 2.93s±0s | 3m:15s±19s |
| UniGIN | 2.24s±0s | 3m:17s±18s |
| UniSAGE | 2.04s±0s | 3m:13s±3s |

(b) CAT-EDGE-MUSIC-BLUES-REVIEWS

| Timings (hh:mm) $\pm$ (s) | | |
|---|---|---|
| Method | Preprocessing Time | Training Time |
| HGNN | 4.17s±0s | 2m:34s±1954s |
| HGNNP | 4.54s±0s | 2m:41s±53s |
| HNHN | 3.06s±0s | 2m:27s±15s |
| HyperGCN | 1.81s±1s | 2m:27s±0s |
| UniGAT | 1.91s±0s | 2m:27s±306s |
| UniGCN | 2.84s±0s | 2m:30s±72s |
| UniGIN | 3.20s±0s | 2m:27s±1189s |
| UniSAGE | 1.65s±0s | 2m:27s±0s |

(c) CAT-EDGE-VEGAS-BARS-REVIEWS

| Timings (hh:mm) $\pm$ (s) | | |
|---|---|---|
| Method | Preprocessing Time | Training Time |
| HGNN | 5.84s±1s | 6m:49s±8s |
| HGNNP | 5.82s±0s | 9m:8s±19s |
| HNHN | 5.95s±0s | 8m:21s±19s |
| HyperGCN | 5.74s±0s | 10m:16s±1s |
| UniGAT | 8.80s±0s | 2m:31s±282s |
| UniGCN | 6.35s±0s | 6m:9s±957s |
| UniGIN | 5.99s±0s | 10m:41s±43s |
| UniSAGE | 5.97s±0s | 9m:50s±0s |

(d) CONTACT-PRIMARY-SCHOOL

| Timings (hh:mm) $\pm$ (s) | | |
|---|---|---|
| Method | Preprocessing Time | Training Time |
| HGNN | 23.25s±1s | 25m:41s±17s |
| HGNNP | 23.25s±0s | 19m:52s±49s |
| HNHN | 24.27s±1s | 5m:12s±63s |
| HyperGCN | 24.00s±0s | 21m:16s±0s |
| UniGAT | 14.27s±0s | 5m:13s±243s |
| UniGCN | 25.44s±0s | 5m:51s±1019s |
| UniGIN | 13.71s±1s | 19m:10s±3972s |
| UniSAGE | 14.08s±2s | 36m:29s±5s |

(e) EMAIL-EU

| Timings (hh:mm) $\pm$ (s) | | |
|---|---|---|
| Method | Preprocessing Time | Training Time |
| HGNN | 4.89s±6s | 1m:27s±8s |
| HGNNP | 2.12s±0s | 2m:42s±30s |
| HNHN | 2.12s±0s | 2m:39s±42s |
| HyperGCN | 2.11s±0s | 40.11s±3s |
| UniGAT | 2.13s±0s | 3m:18s±8s |
| UniGCN | 2.11s±0s | 3m:21s±2s |
| UniGIN | 2.11s±0s | 2m:24s±70s |
| UniSAGE | 2.11s±0s | 2m:8s±49s |

(f) CAT-EDGE-MADISON-RESTAURANT-REVIEWS

| Timings (hh:mm) ± (s) | | | Timings (hh:mm) ± (s) | | |
|---|---|---|---|---|---|
| Method | Preprocessing Time | Training Time | Method | Preprocessing Time | Training Time |
| HGNN | 15.11s±4s | 4m:59s±1s | HGNN | 11.34s±10s | 4m:24s±6s |
| HGNNP | 12.72s±0s | 2m:29s±0s | HGNNP | 6.02s±0s | 4m:13s±2s |
| HNHN | 12.17s±0s | 3m:6s±0s | HNHN | 6.01s±0s | 5m:31s±1s |
| HyperGCN | 12.47s±0s | 49.25s±0s | HyperGCN | 6.32s±0s | 1m:33s±0s |
| UniGAT | 12.74s±0s | 2m:1s±1s | UniGAT | 6.04s±0s | 4m:11s±0s |
| UniGCN | 12.50s±0s | 2m:29s±3s | UniGCN | 5.79s±0s | 4m:12s±0s |
| UniGIN | 12.57s±0s | 2m:16s±3s | UniGIN | 6.64s±1s | 3m:4s±1s |
| UniSAGE | 12.67s±0s | 1m:50s±29s | UniSAGE | 5.79s±0s | 3m:2s±0s |

(a) CONTACT-HIGH-SCHOOL           (b) CAT-EDGE-BRAIN

| Timings (hh:mm) ± (s) | | | Timings (hh:mm) ± (s) | | |
|---|---|---|---|---|---|
| Method | Preprocessing Time | Training Time | Method | Preprocessing Time | Training Time |
| HGNN | 3m:30s±5s | 1h:29m:33s±6s | HGNN | 8m:11s±52s | 37m:18s±9s |
| HGNNP | 3m:34s±1s | 1h:48m:57s±1s | HGNNP | 7m:34s±1s | 47m:56s±1s |
| HNHN | 3m:41s±1s | 2h:9m:34s±1s | HNHN | 6m:21s±1s | 49m:33s±1s |
| HyperGCN | 3m:24s±1s | 58m:27s±1s | HyperGCN | 8m:20s±1s | 28m:25s±1s |
| UniGAT | 3m:50s±1s | 4h:21m:24s±1s | UniGAT | 10m:40s±1s | 1h:54m:36s±1s |
| UniGCN | 3m:38s±1s | 29m:14s±1s | UniGCN | 7m:25s±1s | 2h:40m:20s±1s |
| UniGIN | 3m:50s±1s | 27m:50s±1s | UniGIN | 10m:37s±1s | 2h:48m:35s±1s |
| UniSAGE | 3m:41s±1s | 27m:22s±1s | UniSAGE | 6m:58s±1s | 2h:35m:4s±1s |

(c) WIKIPEOPLE-0BI           (d) JF17K

Table 9: Timings for our method broken up into the preprocessing phase and training phases (2000 epochs) for the hypergraph datasets.

| Timings (hh:mm) ± (s) | | |
|---|---|---|
| Method | Preprocessing Time | Training Time |
| HGNN | 11m:14s±75s | 53m:21s±2845s |
| HGNNP | 11m:10s±21s | 1h:34m:25s±35s |
| HNHN | 5m:15s±395s | 1h:35m:15s±419s |
| HyperGCN | 33.98s±0s | 5m:8s±0s |
| UniGAT | 1m:59s±120s | 2h:2m:47s±25s |
| UniGCN | 34.37s±0s | 1h:17m:38s±2s |
| UniGIN | 34.05s±0s | 1h:16m:38s±7s |
| UniSAGE | 34.36s±0s | 1h:16m:34s±3s |

(a) JOHNSHOPKINS55

| Timings (hh:mm) ± (s) | | |
|---|---|---|
| Method | Preprocessing Time | Training Time |
| HGNN | 17m:9s±164s | 12m:38s±549s |
| HGNNP | 15m:34s±61s | 20m:26s±124s |
| HNHN | 15m:31s±83s | 18m:11s±30s |
| HyperGCN | 15m:46s±32s | 4m:17s±80s |
| UniGAT | 1m:27s±6s | 16m:30s±0s |
| UniGCN | 15m:57s±24s | 18m:42s±170s |
| UniGIN | 16m:14s±73s | 16m:22s±39s |
| UniSAGE | 8m:42s±610s | 8m:49s±324s |

(b) AIFB

| Timings (hh:mm) ± (s) | | |
|---|---|---|
| Method | Preprocessing Time | Training Time |
| HGNN | 4m:1s±11s | 22m:30s±1177s |
| HGNNP | 3m:53s±4s | 39m:30s±3s |
| HNHN | 3m:16s±22s | 44m:7s±71s |
| HyperGCN | 3m:35s±23s | 5m:22s±25s |
| UniGAT | 11.92s±0s | 1h:51m:53s±123s |
| UniGCN | 3m:20s±6s | 39m:18s±51s |
| UniGIN | 3m:21s±8s | 38m:3s±0s |
| UniSAGE | 11.27s±0s | 58m:48s±956s |

(c) AMHERST41

| Timings (hh:mm) ± (s) | | |
|---|---|---|
| Method | Preprocessing Time | Training Time |
| HGNN | 3m:32s±9s | 1h:19m:5s±4684s |
| HGNNP | 3m:26s±10s | 2h:19m:44s±3586s |
| HNHN | 3m:27s±0s | 1h:55m:48s±22s |
| HyperGCN | 3m:28s±0s | 10m:31s±18s |
| UniGAT | 3m:24s±5s | 3h:50m:24s±91s |
| UniGCN | 3m:19s±4s | 1h:39m:46s±13s |
| UniGIN | 3m:17s±0s | 1h:36m:47s±35s |
| UniSAGE | 3m:25s±13s | 1h:37m:16s±102s |

(d) FB15K-237

Table 10: Timings for our method broken up into the preprocessing phase and training phases (2000 epochs) for the graph datasets.

