# OpenReview forum: "Expressive Higher-Order Link Prediction through Hypergraph Symmetry Breaking"
_TMLR — Rejected by TMLR_

### Review · Reviewer_RErF · 2024-03-07

**Summary Of Contributions:**

The present work studies the problem of predicting the hyperedges of a hypergraph by using some hypergraph embedding techniques based on the Weisfeiler-Lehman algorithm. The work presents both theoretical and experimental results.

**Audience:**

Yes

**Claims And Evidence:**

No

**Requested Changes:**

Rewrite the manuscript so that it is possible to follow it.

**Strengths And Weaknesses:**

The main strength is that Weisfeiler-Lehman techniques are actively being investigated and therefore any new piece of work on them is welcome. The main weakness is that the work is, at least to me, almost unintelligible. I read it carefully through the first 4-5 pages, but then I had to give up. Several technical notions are used without being defined, complex mathematical statements are made without any introduction or commentary, and so on.

Details:

1. Section 1 assumes familiarity with GWL, hypergraph prediction, and/or message-passing algorithms. If one is not an expert then it will be very difficult to figure out what the section is saying.

2. What does it mean for two neighborhoods (i.e., two sets of vertices) to be isomorphic? That the subgraphs they induce are isomorphic?

3. The "star expansion incidence matrix" is simply the incidence matrix of H.

4. Proposition 2.1. It says "Aut ≈ Stab are equivalent as isomorphic groups". I do not understand what "equivalent" means, and moreover I guess "Aut ≈ Stab" should be "Aut and Stab", or alternatively the proposition should just say "Aut ≈ Stab."

5. Above and below Proposition 2.1 it is said "[the elements of Stab] are equivalent to automorphisms on the original hypergraph H" and "Intuitively, hypergraph automorphisms and stablizer permutations on star expansion adjacancy matrices are equivalent". This is just repeating what the proposition says; there's no further intuition or information.

6. Section 2 uses non-elementary mathematics without properly introducing it. In particular it uses group actions, orbits, and orbit stabilizers (see e.g. Equation (2)), without introducing any of them.

7. 2.2 says "These two conditions mean that the representation does not lose any information when doing prediction for missing hyperedges on k nodes". If that is true then one should be able to retrieve from the representation the subgraph induced by the k nodes. I doubt this holds here.

8. In 2.3 there is no intuition or introduction for the GWL-1 method, and in particular for Equation (3), which is therefore very hard to understand. It is said that X is a vector of attributes, but the paper was only talking about unattributed hypergraphs so far. Moreove  I do not see the relationship with the k-node representations of the previous section; the set of equations defined here seem to yield a sequence of nested tuples and not a vector in R^d.

9. In 2.3, the discussion in the paragraph after Eq (3) is totally cryptic to me. What is the goal now? In particular, I was totally lost at "Define the operator AGG as a permutation invariant map on a set of vectors to representation space R^d". What does that mean? What is the purpose of this?

10. The statement of the problem (Problem 1) is sloppy. What does "Predict Egt \ E" mean? One needs to define a cost, or a loss, and possibly a distribution. In any case there must be a way to measure the error of the algorithm, as well as some assumption on the relationship between Egt and E. Here there is none of that.

---

> ### Author Response · Authors · 2024-03-26
>
> We would like to thank the reviewer for taking the time to provide detailed comments, which are very helpful for us to further improve the quality and readability of the paper. We have addressed all your comments and questions one by one in the following:
>
> 1. The introduction presents a hypergraph and how current approaches learn on them. We have explained in the introduction what higher order link prediction is, what GWL-1 and WL-1 are as local message passing algorithms within node neighborhoods and what the issues are with them, namely that they view graphs/hypergraphs as forests.
>
> 2. After definition 2.2, it is stated: “A neighborhood $N(v) \triangleq (\cup_{v \in e} e, \\{e : e \ni v \\})$ of a node $v \in \mathcal{V}$ of a hypergraph $\mathcal{H} = (\mathcal{V}, \mathcal{E})$ is the subhypergraph of $\mathcal{H}$ induced by the set of all hyperedges incident to $v$. Thus isomorphic neighborhoods $N(u),N(v) \subset \mathcal{H}$ are isomorphic as subhypergraphs of $\mathcal{H}$.
>
> 3. There are many possible definitions of incidence matrices. Taking this liberty, we call $H$ the star expansion incidence matrix which defines the connectivity between nodes and hyperedges. Other possible incidence matrices could be the node tuple incidence matrix: $A$ where $A_{i_1…i_n}=1$ if $e=(i_1...i_n)$ forms a hyperedge in the hypergraph, $0$ otherwise.
>
> 4. The stabilizer subgroup is defined by the set of permutations whose action on the star expansion incidence matrix $H$ is invariant. The automorphism group is defined by isomorphisms (see Defn. 2.2) from a hypergraph $\mathcal{H}$ to itself. Since one is defined on integer indices of a matrix and the other is defined on the nodes of a hypergraph, we have two different groups. They are isomorphic as groups, however. In group theory, an isomorphism is a bijective homomorphism between groups. The two groups are thus “equivalent” up to isomorphism.
>
> 5. See response to (4) above.
>
> 6. There is no prior knowledge assumed in Section 2 except for elementary knowledge of what a group is. We have added a citation to Dummit and Foote, a standard textbook in abstract algebra. We already define the action of a permutation on a star expansion incidence matrix and how it defines a stabilizer subgroup. Based on the writing, there is actually no need to have prior knowledge of these terms since they are defined explicitly for the reader. We never define or use the word orbit anywhere.
>
> 7. The statement was intended to mean predicting a single hyperedge on $k$ nodes instead of any hyperedge on $k$ nodes. This follows since $h(S,H)$ is an injective learner on set $S$ up to automorphism of $\mathcal{H}$. We have rewritten this to say: “when doing prediction for missing $k$-sized hyperedges on a set of $k$ nodes.”
>
> 8. We have removed the initial attributes in the presentation of GWL-1. We have updated the notation in Section 2.3: GWL-1 returns multisets on each node. After $L$ passes GWL-1, a set of GWL-1 multiset representations of a set of nodes can then be encoded as a vector by a set learner (AGG) or enumerated into a vector for any order.
>
> 9. We have updated the manuscript for clarity accordingly. The idea here is to show two properties of GWL-1: (1). permutation equivariance over all the nodes and (2). $k$-node invariance of a set learner on sets of GWL-1 representations of $k$ nodes.
>
> 10. We have clarified the problem statement to say: “Given a hypergraph $\mathcal{H} = (\mathcal{V}, \mathcal{E})$ and ground truth hypergraph $\mathcal{H_{gt}} = (\mathcal{V}, \mathcal{E_{gt}}), \mathcal{E} \subset \mathcal{E_{gt}}$, where $\mathcal{E}$ is observable: Predict the existence of the unobserved hyperedges $\mathcal{E_{gt}} \setminus \mathcal{E}$.” The problem statement only asks for a prediction of a set. This does not require specifying any tools for approximation such as losses, or definitions of error.

---

### Review · Reviewer_QGYa · 2024-03-09

**Summary Of Contributions:**

The paper addresses limitations in existing hypergraph representation learning by introducing a preprocessing algorithm to identify and break symmetry in regular subhypergraphs. Through extensive experiments, they demonstrate the effectiveness of their approach for higher-order link prediction on both graph and hypergraph datasets with minimal increase in computation.

**Audience:**

Yes

**Broader Impact Concerns:**

N/A.

**Claims And Evidence:**

Yes

**Requested Changes:**

- Hypergraph Symmetry Breaking does not always enhance performance. If symmetry is a crucial pattern for graph learning tasks, disrupting it will hinder the network's ability to acquire information.
- Theoretical contribution is limited, and all theorems are not hard to be extended from the common graphs. The definition of GWL is also obtained with minor modifications to existing work.
- Lack of experiments on WikiPeople and JF17k.

**Strengths And Weaknesses:**

- The idea of breaking symmetry is novel.
- The proposed method has good experimental results.

---

> ### Author Response · Authors · 2024-03-26
>
> We are grateful to your constructive review on our paper, and your encouraging comments on the novelty of our symmetry breaking idea and experimental results. We have addressed your questions and suggestions one by one as follows.
>
> 1. Our work focuses only on addressing the symmetry problem. Our intention is not to build a one size fits all solution for all datasets. Since hypergraph representation learners based on GWL-1 only see the hypergraph as a forest, it is reasonable to say that there is a necessity for some symmetry breaking. If the application at hand, however, only requires this forest view of the hypergraph, then symmetry breaking may not be needed. This is hard to know in practice, however, but should not degrade baseline performance.
>
> 2. Since hypergraphs are equivalent to bipartite graphs, our proofs naturally reduce to proofs on graphs. GWL is a message passing algorithm that encompasses the way many existing hyperGNN architectures compute on a hypergraph so it is not  a completely new definition.
>
> 3. Thank you for suggesting the two datasets. We have added PR-AUC and timing tables for WikiPeople and JF17K in the Appendix. (see Table 3, Table 9). The new results further confirm the effectiveness of our method to address the issues in the symmetry problem.

---

### Review · Reviewer_3KN8 · 2024-03-14

**Summary Of Contributions:**

The paper studies link prediction on hypergraphs, which are representations of higher-order relationships on a graph. The paper devises a generalized Weisfeiler-Lehman test, which is a condition to tell if two hypergraphs are equivalent or not under this test.

The paper then moves on to propose a method to predict higher-order links.

The experiments focus on six different hypergraph datasets, comparing the proposed method to a couple of hyper-order graph neural networks.

**Audience:**

Yes

**Claims And Evidence:**

No

**Requested Changes:**

On Page 9, the authors write Algorithm 14---should it be Algorithm 1?

Other requested changes include addressing the weaknesses mentioned above.

**Strengths And Weaknesses:**

S1) The paper is detailed in detail, especially the theoretical conditions of the Weisfeiler-Lehman test.

S2) The appendix (which I did not check in detail) provides great details about the theoretical statements.

W1) The experimental comparison is weak---from looking at Table 1 and Table 2, I could not see a significant advantage brought by the proposed method.

W2) There is no description of the implementation of the network architectures, and the baselines are not described in detail for me to appreciate the experimental results.

W3) There are no results concerning the convergence of the training dynamics---even though it is stated in Proposition 5.5 that the algorithm runs in linear time to the input, the number of iterations required to train this method is not clear.

W4) I didn't get a great sense of the intuition behind the benefit of adding this symmetry breaking to the method---to better explain this, a simple working example would help. The fact that the empirical gains are marginal added to my concern regarding the benefit of this idea.

---

> ### Author Response · Authors · 2024-03-26
>
> Thank you for the comments on our paper. We have addressed the weaknesses, questions and suggestions one by one as follows.
>
> * In this paper, we focus on addressing the symmetry problem. There is no guarantee that every dataset requires addressing the symmetry problem. We have shown that our method would not worsen the baselines since if there are any false positive symmetries then they would be addressed. We have also added experimental results on two additional data sets, which are PR-AUC and timing tables for WikiPeople and JF17K in the Appendix. (see Table 3, Table 9).
>
> * Thank you for the suggestions. We have added a new section (Section D) that gives full descriptions of all the baseline architectures and their relation to GWL-1 used in the experiments. This will help readers to have a comprehensive understanding of the implementation of all the network architectures.
>
> * The preprocessing algorithm is nonparametric so no learning is necessary. This is intentional since there is no need to overfit the data augmentations since they were never observed. Learning these data augmentations would bias the training data to an incorrect hyperedge distribution. We hope this answer addresses your concern.
>
> * We hope Figure 2 provides a simple illustration of symmetry breaking that addresses your question.
>
> * Thank you for catching the typo. Page 9 now has Algorithm 1 linked.

---

### Decision · Action_Editor_mzLZ · 2024-04-13

**Recommendation:** Reject

**Comment:**

I struggled with this decision since I can see value in the approach.  But I defer to the expert reviewers who are concerned with the technically dense explanation for the claims, paired with minor empirical performance.  I hope the feedback is helpful to the authors.

**Audience:**

The reviewers found the topic appropriate for the audience, but as mentioned above, struggled with the technically dense language used to describe the methods.

**Claims And Evidence:**

The expert reviewers were not convinced of the contribution of the paper.  This seems to arise from a combination of issues:
 (1) the mathematical language makes it hard for machine learning experts (who work in link prediction and other learning on hypergraphs) to understand how the algorithm work, and the full meaning of the claims about the algorithm.  A more gentle introduction providing intuition for the abstract algebra could have made this paper more accessible for the audience of this venue.
 (2) the proofs are deferred to the appendix, and there is not a lot of intuition in the main text provided for why those claims are true.
 (3) I found Section 4.1 and Figure 2 very helpful.  But this came well after the introduction, and before I reached this, I was somewhat lost as to what the main thrust was.
 (4) The empirical results are small, and for the most part, within the margin of error.

I hope that points (1,2,3) provide feedback on how this paper could be re-organized so that it would become more appropriate for this venue.



Regarding (2):  The paper was submitted as not a 12-page paper, so reviewers should be able to judge the content, for the most part, in the first 12 pages.  This ensures a faster review process.  TMLR also takes longer submissions, and then gives reviewers more time to review.   Perhaps the longer format would have a better choice so you could have included more explanation outside of the appendix for why the claims are true in the main part meant to be carefully evaluated.

**Resubmission Of Major Revision:**

The authors may consider submitting a major revision at a later time.